# A PRIMAL-DUAL APPROACH TO SOLVING VARIATIONAL INEQUALITIES WITH GENERAL CONSTRAINTS

**Tatjana Chavdarova**[*]
University of California, Berkeley
tatjana.chavdarova@berkeley.edu

**Tong Yang**[*]
Carnegie Mellon University
tongyang@andrew.cmu.edu

**Matteo Pagliardini**
University of California, Berkeley & EPFL
matteo.pagliardini@epfl.ch

**Michael I. Jordan**
University of California, Berkeley
jordan@cs.berkeley.edu

## ABSTRACT

Yang et al. (2023) recently showed how to use first-order gradient methods to solve general variational inequalities (VIs) under a limiting assumption that analytic solutions of specific subproblems are available. In this paper, we circumvent this assumption via a warm-starting technique where we solve subproblems approximately and initialize variables with the approximate solution found at the previous iteration. We prove the convergence of this method and show that the gap function of the last iterate of the method decreases at a rate of $\mathcal{O}(\frac{1}{\sqrt{K}})$ when the operator is $L$-Lipschitz and monotone. In numerical experiments, we show that this technique can converge much faster than its exact counterpart. Furthermore, for the cases when the inequality constraints are simple, we introduce an alternative variant of ACVI and establish its convergence under the same conditions. Finally, we relax the smoothness assumptions in Yang et al., yielding, to our knowledge, the first convergence result for VIs with general constraints that does not rely on the assumption that the operator is $L$-Lipschitz.

## 1 INTRODUCTION

We study variational inequalities (VIs), a general class of problems that encompasses both equilibria and optima. The general (constrained) VI problem involves finding a point $\boldsymbol{x}^\star \in \mathcal{X}$ such that:

$$\langle \boldsymbol{x} - \boldsymbol{x}^\star, F(\boldsymbol{x}^\star) \rangle \geq 0, \quad \forall \boldsymbol{x} \in \mathcal{X}, \tag{cVI}$$

where $\mathcal{X}$ is a subset of the Euclidean $n$-dimensional space $\mathbb{R}^n$, and where $F \colon \mathcal{X} \mapsto \mathbb{R}^n$ is a continuous map. VIs generalize standard constrained minimization problems, where $F$ is a gradient field $F \equiv \nabla f$, and, by allowing $F$ to be a general vector field, they also include problems such as finding equilibria in zero-sum games and general-sum games (Cottle & Dantzig, 1968; Rockafellar, 1970). This increased expressivity underlies their practical relevance to a wide range of emerging applications in machine learning, such as *(i)* multi-agent games (Goodfellow et al., 2014; Vinyals et al., 2017), *(ii)* robustification of single-objective problems, which yields min-max formulations (Szegedy et al., 2014; Mazuelas et al., 2020; Christiansen et al., 2020; Rothenhäusler et al., 2018), and *(iii)* statistical approaches to modeling complex multi-agent dynamics in stochastic and adversarial environments. We refer the reader to (Facchinei & Pang, 2003; Yang et al., 2023) for further examples.

Such generality comes, however, at a price in that solving for equilibria is notably more challenging than solving for optima. In particular, as the Jacobian of $F$ is not necessarily symmetric, we may have rotational trajectories or *limit cycles* (Korpelevich, 1976; Hsieh et al., 2021). Moreover, in sharp contrast to standard minimization, the last iterate can be quite far from the solution even though the average iterate converges to the solution (Chavdarova et al., 2019). This has motivated recent efforts

---

[*]Equal contribution. Source code: https://github.com/Chavdarova/I-ACVI.

to study specifically the convergence of the *last iterate* produced by gradient-based methods. Thus, herein, our focus and discussions refer to the last iterate.

Recent work has focused primarily on solving VIs in two cases of the domain $\mathcal{X}$: *(i)* the unconstrained setting where $\mathcal{X} \equiv \mathbb{R}^n$ (Golowich et al., 2020b; Chavdarova et al., 2023; Gorbunov et al., 2022a; Bot et al., 2022) and for *(ii)* the constrained setting with *projection-based* methods (Tseng, 1995; Daskalakis et al., 2018; Diakonikolas, 2020; Nemirovski, 2004; Mertikopoulos et al., 2019; Cai et al., 2022). The latter approach assumes that the projection is "simple," in the sense that this step does not require gradient computation. This holds, for example, for inequality constraints of the form $\boldsymbol{x} \leq \tau$ where $\tau$ is some constant, in which case fast operations such as clipping suffice. However, as is the case in constrained minimization, the constraint set—denoted herein with $\mathcal{C} \subseteq \mathcal{X}$—is, in the general case, an intersection of finitely many inequalities and linear equalities:

$$\mathcal{C} = \left\{ \boldsymbol{x} \in \mathbb{R}^n | \varphi_i(\boldsymbol{x}) \leq 0, i \in [m], \ \boldsymbol{C}\boldsymbol{x} = \boldsymbol{d} \right\}, \tag{CS}$$

where each $\varphi_i \colon \mathbb{R}^n \mapsto \mathbb{R}$, $\boldsymbol{C} \in \mathbb{R}^{p \times n}$, and $\boldsymbol{d} \in \mathbb{R}^p$. Given a general CS (without assuming additional structure), implementing the projection requires second-order methods, which quickly become computationally prohibitive as the dimension $n$ increases. If the second-order derivative computation is approximated, the derived convergence rates will yet be multiplied with an additional factor; thus, the resulting rate of convergence may not match the known lower bound (Golowich et al., 2020a; Cai et al., 2022). This motivates a third thread of research, focusing on *projection-free* methods for the constrained VI problem, where the update rule does not rely on the projection operator. This is the case we focus on in this paper.

There has been significant work on developing second-order projection-free methods for the formulation in cVI; we refer the interested reader to (Chapter 7, Nesterov & Nemirovski, 1994) and (Chapter 11, Facchinei & Pang, 2003, vol. 2) for example. We remark that the seminal mirror-descent and mirror-prox methods (Nemirovski & Yudin, 1983; Beck & Teboulle, 2003; Nemirovski, 2004) (see App. A.5) exploit a certain structure of the domain and avoid the projection operator, but cannot be applied for general CS.

In recent work, Yang et al. (2023) presented a first-order method, referred to as the ***ADMM-based Interior Point Method for Constrained VIs*** (ACVI), for solving the cVI problem with general constraints. ACVI combines path-following interior point (IP) methods and primal-dual methods. Regarding the latter, it generalizes the *alternating direction method of multipliers* (ADMM) method (Glowinski & Marroco, 1975; Gabay & Mercier, 1976), an algorithmic paradigm that is central to large-scale optimization (Boyd et al., 2011; Tibshirani, 2017)–see (Yang et al., 2023) and App. A.1; but which has been little explored in the cVI context. On a high level, ACVI has two nested loops: *(i)* the outer loop smoothly decreases the weight $\mu_i$ of the inequality constraints as in IP methods, whereas *(ii)* the inner loop performs a primal-dual update (for a fixed $\mu_i$) as follows:

- solve a subproblem whose main (primal) variable $\boldsymbol{x}_i^j$ aims to satisfy the equality constraints,
- solve a subproblem whose main (primal) variable $\boldsymbol{y}_i^j$ aims to satisfy the inequality constraints,
- update the dual variable $\boldsymbol{\lambda}_i^j$.

The first two steps solve the subproblems exactly using an analytical expression of the solution, and the variables converge to the same value, thus eventually satisfying both the inequality and equality constraints. See Algorithm 3 for a full description, and see Fig. 2 for illustrative examples. The authors documented that projection-based methods may extensively zig-zag when hitting a constraint when there is a rotational component in the vector field, an observation that further motivates projection-free approaches even when the projection is simple.

Yang et al. showed that the gap function of the last iterate of ACVI decreases at a rate of $\mathcal{O}(\frac{1}{\sqrt{K}})$ when the operator is $L$-Lipschitz, monotone, and at least one constraint is active. It is, however, an open problem to determine if the same rate on the gap function applies while assuming only that the operator is monotone (where monotonicity for VIs is analogous to convexity for standard minimization, see Def. 2.1). Moreover, in some cases, the subproblems of ACVI may be cumbersome to solve analytically. Hence, a natural question is whether we can show convergence approximately when the subproblems are solved. As a result, we raise the following questions:

- *Does the last iterate of ACVI converge when the operator is monotone without requiring it to be L-Lipschitz?*

- *Does ACVI converge when the subproblems are solved approximately*?

In this paper, we answer the former question affirmatively. Specifically, we prove that the last iterate of ACVI converges at a rate of $\mathcal{O}(\frac{1}{\sqrt{K}})$ in terms of the gap function (Def. 2.2) even when assuming only the monotonicity of the operator. The core of our analysis lies in identifying a relationship between the reference point of the gap function and a KKT point that ACVI targets implicitly (i.e., it does not appear explicitly in the ACVI algorithm). This shows that ACVI explicitly works to decrease the gap function at each iteration. The argument further allows us to determine a convergence rate by making it possible to upper bound the gap function. This is in contrast to the approach of Yang et al. (2023), who upper bound the iterate distance and then the gap function, an approach that requires a Lipschitz assumption. This is the first convergence rate for the last iterate for monotone VIs with constraints that does not rely on an $L$-Lipschitz assumption on the operator.

To address the latter question, we leverage a fundamental property of the ACVI algorithm—namely, its homotopic structure as it smoothly transitions to the original problem, a homotopy that inherently arises from its origin as an interior-point method (Boyd & Vandenberghe, 2004). Moreover, due to the alternating updates of the two sets of parameters of ACVI ($x$ and $y$; see Algorithm 3), the subproblems change negligibly, with the changes proportional to the step sizes. This motivates the standard *warm-start* technique where, at every iteration, instead of initializing at random, we initialize the corresponding optimization variable with the approximate solution found at the previous iteration. We refer to the resulting algorithm as *inexact ACVI*, described in Algorithm 1. Furthermore, inspired by the work of Schmidt et al. (2011), which focuses on the proximal gradient method for standard minimization, we prove that inexact ACVI converges with the same rate of $\mathcal{O}(\frac{1}{\sqrt{K}})$, under a condition on the rate of decrease of the approximation errors. We evaluate inexact ACVI empirically on 2D and high-dimensional games and show how multiple inexact yet computationally efficient iterations can lead to faster wall-clock convergence than fewer exact ones.

Finally, we provide a detailed study of a special case of the problem class that ACVI can solve. In particular, we focus on the case when the inequality constraints are simple, in the sense that projection on those inequalities is fast to compute. Such problems often arise in machine learning, e.g., whenever the constraint set is an $L_p$-ball, with $p \in \{1, 2, \infty\}$ as in adversarial training (Goodfellow et al., 2015). We show that the same convergence rate holds for this variant of ACVI. Moreover, we show empirically that when using this method to train a constrained GAN on the MNIST (Lecun & Cortes, 1998) dataset, it converges faster than the projected variants of the standard VI methods.

In summary, our main contributions are as follows:

- We show that the gap function of the last iterate of ACVI (Yang et al., 2023, Algorithm 1 therein) decreases at a rate of $\mathcal{O}(\frac{1}{\sqrt{K}})$ for monotone VIs, without relying on the assumption that the operator is $L$-Lipschitz.
- We combine a standard warm-start technique with ACVI and propose a precise variant with approximate solutions, named *inexact ACVI*—see Algorithm 1. We show that inexact ACVI recovers the same convergence rate as ACVI, provided that the errors decrease at appropriate rates.
- We propose a variant of ACVI designed for inequality constraints that are fast to project to—see Algorithm 2. We guarantee its convergence and provide the corresponding rate; in this case, we omit the central path, simplifying the convergence analysis.
- Empirically, we: *(i)* verify the benefits of warm-start of the inexact ACVI; *(ii)* observe that I-ACVI can be faster than other methods by taking advantage of cheaper approximate steps; *(iii)* train a constrained GAN on MNIST and show the projected version of ACVI is faster to converge than other methods; and *(iv)* provide visualizations contrasting the different ACVI variants.

## 1.1 RELATED WORKS

**Last-iterate convergence of first-order methods on VI-related problems.** When solving VIs, the last and average iterates can be far apart; see examples in (Chavdarova et al., 2019). Thus, an extensive line of work has aimed at obtaining last-iterate convergence for special cases of VIs that are important in applications, including bilinear or strongly monotone games (e.g., Tseng, 1995; Malitsky, 2015; Facchinei & Pang, 2003; Daskalakis et al., 2018; Liang & Stokes, 2019; Gidel et al., 2019b; Azizian et al., 2020; Thekumparampil et al., 2022), and VIs with cocoercive operators (Diakonikolas, 2020). Several papers exploit continuous-time analyses as these provide

direct insights on last-iterate convergence and simplify the derivation of the Lyapunov potential function (Ryu et al., 2019; Bot et al., 2020; Rosca et al., 2021; Chavdarova et al., 2023; Bot et al., 2022). For monotone VIs, *(i)* Golowich et al. (2020b;a) established that the lower bound of $\tilde{p}$-*stationary canonical linear iterative* ($\tilde{p}$-SCLI) first-order methods (Arjevani et al., 2016) is $\mathcal{O}(\frac{1}{\tilde{p}\sqrt{K}})$, *(ii)* Golowich et al. (2020b) obtained a rate in terms of the gap function, relying on first- and second-order smoothness of $F$, *(iii)* Gorbunov et al. (2022a) and Gorbunov et al. (2022b) obtained a rate of $\mathcal{O}(\frac{1}{K})$ for extragradient (Korpelevich, 1976) and optimistic GDA (Popov, 1980), respectively—in terms of reducing the squared norm of the operator, relying on first-order smoothness of $F$, and *(iv)* Golowich et al. (2020b) and Chavdarova et al. (2023) provided the best iterate rate for OGDA while assuming first-order smoothness of $F$. Daskalakis & Panageas (2019) focused on zero-sum convex-concave constrained problems and provided an asymptotic convergence guarantee for the last iterate of the *optimistic multiplicative weights update* (OMWU) method. For constrained and monotone VIs with $L$-Lipschitz operator, Cai et al. (2022) recently showed that the last iterate of extragradient and optimistic GDA have a rate of convergence that matches the lower bound. Gidel et al. (2017) consider strongly convex-concave zero-sum games with strongly convex constraint set to study the convergence of the Frank-Wolfe method (Lacoste-Julien & Jaggi, 2015).

**Interior point (IP) methods for VIs.** IP methods are a broad class of algorithms for solving problems constrained by general inequality and equality constraints. One of the widely adopted sub-classes within IP methods utilizes log-barrier terms to handle inequality constraints. They typically rely on Newton's method, which iteratively approaches the solution from the feasible region. Several works extend IP methods for constrained VI problems. Among these, Nesterov & Nemirovski (Chapter 7, 1994) study extensions to VI problems while relying on Newton's method. Further, an extensive line of work discusses specific settings (e.g., Chen et al., 1998; Qi & Sun, 2002; Qi et al., 2000; Fan & Yan, 2010). On the other hand, Goffin et al. (1997) described a second-order cutting-plane method for solving pseudomonotone VIs with linear inequalities. Although these methods enjoy fast convergence regarding the number of iterations, each iteration requires computing second-order derivatives, which becomes computationally prohibitive for large-scale problems. Recently, Yang et al. (2023) derived the aforementioned ACVI method which combines IP methods and the ADMM method, resulting in a *first*-order method that can handle general constraints.

## 2 PRELIMINARIES

**Notation.** Bold small and bold capital letters denote vectors and matrices, respectively, while curly capital letters denote sets. We let $[n]$ denote $\{1, \dots, n\}$ and let $e$ denote vector of all 1's. The Euclidean norm of $v$ is denoted by $\|v\|$, and the inner product in Euclidean space by $\langle \cdot, \cdot \rangle$. $\odot$ denotes element-wise product.

**Problem.** Let $rank(C) = p$ be the rank of $C$ as per (CS). With abuse of notation, let $\varphi$ be the concatenated $\varphi_i(\cdot), i \in [m]$. We assume that each of the inequality constraints is convex and $\varphi_i \in C^1(\mathbb{R}^n), i \in [m]$. We define the following sets:
$$\mathcal{C}_{\leq} \triangleq \{x \in \mathbb{R}^n \,|\, \varphi(x) \leq \mathbf{0}\}, \quad \mathcal{C}_{<} \triangleq \{x \in \mathbb{R}^n \,|\, \varphi(x) < \mathbf{0}\}, \quad \text{and} \quad \mathcal{C}_{=} \triangleq \{y \in \mathbb{R}^n \,|\, Cy = d\};$$
thus the relative interior of $\mathcal{C}$ is $int\,\mathcal{C} \triangleq \mathcal{C}_{<} \cap \mathcal{C}_{=}$. We assume $int\,\mathcal{C} \neq \emptyset$ and that $\mathcal{C}$ is compact.

In the following, we list the necessary definitions and assumptions; see App. A for additional background. We define these for a general domain set $\mathcal{S}$, and by setting $\mathcal{S} \equiv \mathbb{R}^n$ and $\mathcal{S} \equiv \mathcal{X}$, these refer to the unconstrained and constrained settings, respectively. We will use the standard *gap function* as a convergence measure, which requires $\mathcal{S}$ to be compact to define it.

**Definition 2.1** (monotone operators). An operator $F: \mathcal{X} \supseteq \mathcal{S} \to \mathbb{R}^n$ is monotone on $\mathcal{S}$ if and only if the following inequality holds for all $x, x' \in \mathcal{S}$: $\langle x - x', F(x) - F(x') \rangle \geq 0$.

**Definition 2.2** (gap function). Given a candidate point $x' \in \mathcal{X}$ and a map $F: \mathcal{X} \supseteq \mathcal{S} \to \mathbb{R}^n$ where $\mathcal{S}$ is compact, the gap function $\mathcal{G}: \mathbb{R}^n \to \mathbb{R}$ is defined as: $\mathcal{G}(x', \mathcal{S}) \triangleq \max_{x \in \mathcal{S}} \langle F(x'), x' - x \rangle$.

**Definition 2.3** ($\sigma$-approximate solution). Given a map $F: \mathcal{X} \to \mathbb{R}^n$ and a positive scalar $\sigma$, $x \in \mathcal{X}$ is said to be a $\sigma$-approximate solution of $F(x) = \mathbf{0}$ iff $\|F(x)\| \leq \sigma$.

**Definition 2.4** ($\varepsilon$-minimizer). Given a minimization problem $\min_{x} h(x)$, s.t. $x \in \mathcal{S}$, and a fixed positive scalar $\varepsilon$, a point $\hat{x} \in \mathcal{S}$ is said to be an $\varepsilon$-minimizer of this problem if and only if it holds that: $h(\hat{x}) \leq h(x) + \varepsilon, \quad \forall x \in \mathcal{S}$.

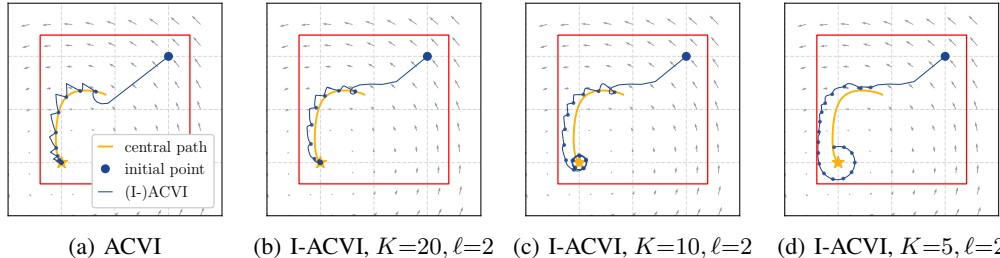

(a) ACVI     (b) I-ACVI, $K{=}20, \ell{=}2$   (c) I-ACVI, $K{=}10, \ell{=}2$   (d) I-ACVI, $K{=}5, \ell{=}2$

Figure 1: **Convergence of ACVI and I-ACVI on the (2D-BG) problem**. The central path is depicted in yellow. For all methods, we show the $\boldsymbol{y}$-iterates initialized at the same point (blue circle). Each subsequent point on the trajectory depicts the (exact or approximate) solution at the end of the inner loop. A yellow star represents the game's Nash equilibrium (NE), and the constraint set is the interior of the red square. **(a)**: As we decay $\mu_t$, the solutions of the inner loop of ACVI follow the central path. As $\mu_t \to 0$, the solution of the inner loop of ACVI converges to the NE. **(b, c, d)**: When the $\boldsymbol{x}$ and $\boldsymbol{y}$ subproblems are solved approximately with a finite $K$ and $\ell$, the iterates need not converge as the approximation error increases (and $K$ decreases). See § 5 for a discussion.

---

**Algorithm 1** Inexact ACVI (I-ACVI) pseudocode.

1: **Input:** operator $F \colon \mathcal{X} \to \mathbb{R}^n$, constraints $\boldsymbol{Cx} = \boldsymbol{d}$ and $\varphi_i(\boldsymbol{x}) \leq 0, i = [m]$, hyperparameters $\mu_{-1}, \beta > 0, \delta \in (0,1)$, barrier map $\wp$ ($\wp_1$ or $\wp_2$), inner optimizers $\mathcal{A}_{\boldsymbol{x}}$ (e.g. EG, GDA) and $\mathcal{A}_{\boldsymbol{y}}$ (GD) for the $\boldsymbol{x}$ and $\boldsymbol{y}$ subproblems, resp.; outer and inner loop iterations $T$ and $K$, resp.
2: **Initialize:** $\boldsymbol{x}_0^{(0)} \in \mathbb{R}^n, \boldsymbol{y}_0^{(0)} \in \mathbb{R}^n, \boldsymbol{\lambda}_0^{(0)} \in \mathbb{R}^n$
3: $\boldsymbol{P}_c \triangleq \boldsymbol{I} - \boldsymbol{C}^{\intercal}(\boldsymbol{CC}^{\intercal})^{-1}\boldsymbol{C}$     where $\boldsymbol{P}_c \in \mathbb{R}^{n \times n}$
4: $\boldsymbol{d}_c \triangleq \boldsymbol{C}^{\intercal}(\boldsymbol{CC}^{\intercal})^{-1}\boldsymbol{d}$     where $\boldsymbol{d}_c \in \mathbb{R}^n$
5: **for** $t = 0, \ldots, T-1$ **do**
6:     $\mu_t = \delta \mu_{t-1}$
7:     **for** $k = 0, \ldots, K-1$ **do**
8:         Set $\boldsymbol{x}_{k+1}^{(t)}$ to be a $\sigma_{k+1}$-approximate solution of: $\boldsymbol{x} + \frac{1}{\beta}\boldsymbol{P}_c F(\boldsymbol{x}) - \boldsymbol{P}_c \boldsymbol{y}_k^{(t)} + \frac{1}{\beta}\boldsymbol{P}_c \boldsymbol{\lambda}_k^{(t)} - \boldsymbol{d}_c = \boldsymbol{0}$
            (w.r.t. $\boldsymbol{x}$), by running $\ell_{\boldsymbol{x}}^{(t)}$ steps of $\mathcal{A}_{\boldsymbol{x}}$, with $\boldsymbol{x}$ initialized to the previous solution ($\boldsymbol{x}_k^{(t)}$ if $k > 0$, else $\boldsymbol{x}_K^{(t-1)}$)
9:         Set $\boldsymbol{y}_{k+1}^{(t)}$ to be an $\varepsilon_{k+1}$-minimizer of $\min_{\boldsymbol{y}} \sum_{i=1}^m \wp\big(\varphi_i(\boldsymbol{y}), \mu\big) + \frac{\beta}{2}\left\| \boldsymbol{y} - \boldsymbol{x}_{k+1}^{(t)} - \frac{1}{\beta}\boldsymbol{\lambda}_k^{(t)} \right\|^2$,
            by running $\ell_{\boldsymbol{y}}^{(t)}$ steps of $\mathcal{A}_{\boldsymbol{y}}$, with $\boldsymbol{y}$ initialized to $\boldsymbol{y}_k^{(t)}$ when $k > 0$, or $\boldsymbol{y}_K^{(t-1)}$ otherwise
10:         $\boldsymbol{\lambda}_{k+1}^{(t)} = \boldsymbol{\lambda}_k^{(t)} + \beta(\boldsymbol{x}_{k+1}^{(t)} - \boldsymbol{y}_{k+1}^{(t)})$
11:     **end for**
12:     $(\boldsymbol{y}_0^{(t+1)}, \boldsymbol{\lambda}_0^{(t+1)}) \triangleq (\boldsymbol{y}_K^{(t)}, \boldsymbol{\lambda}_K^{(t)})$
13: **end for**

---

## 3 CONVERGENCE OF THE EXACT AND INEXACT ACVI ALGORITHMS FOR MONOTONE VIs

In this section, we present our main theoretical findings: *(i)* the rate of convergence of the last iterate of ACVI (the exact ACVI algorithm is stated in App. A) while relying exclusively on the assumption that the operator $F$ is monotone; and *(ii)* the corresponding convergence when the subproblems are solved approximately—where the proposed algorithm is referred to as *inexact ACVI*—Algorithm 1 ($\wp_1$, $\wp_2$ are defined below). Note that we only assume $F$ is $L$-Lipschitz for the latter result, and if we run Algorithm 1 with extragradient for line 8, for example, the method only has a convergence guarantee if $F$ is $L$-Lipschitz (see Korpelevich, 1976, Theorem 1). For easier comparison with one loop algorithms, we will state both of these results for a fixed $\mu_{-1}$ (hence only have the $k \in [K]$ iteration count) as in (Yang et al., 2023); nonetheless, the same rates hold without knowing $\mu_{-1}$—see App. B.4 in Yang et al. (2023) and our App. B.3. Thus, both guarantees are parameter-free.

## 3.1 LAST ITERATE CONVERGENCE OF EXACT ACVI

**Theorem 3.1** (Last iterate convergence rate of ACVI—Algorithm 1 in (Yang et al., 2023)). *Given a continuous operator $F\colon \mathcal{X} \to \mathbb{R}^n$, assume: (i) F is monotone on $\mathcal{C}_=$, as per Def. 2.1; (ii) either F is strictly monotone on $\mathcal{C}$ or one of $\varphi_i$ is strictly convex. Let $(\boldsymbol{x}_K^{(t)}, \boldsymbol{y}_K^{(t)}, \boldsymbol{\lambda}_K^{(t)})$ denote the last iterate of ACVI. Given any fixed $K \in \mathbb{N}_+$, run with sufficiently small $\mu_{-1}$, then $\forall t \in [T]$, it holds that:*

$$\mathcal{G}(\boldsymbol{x}_K^{(t)}, \mathcal{C}) \leq \mathcal{O}(\frac{1}{\sqrt{K}}), \text{ and } \left\| \boldsymbol{x}_K^{(t)} - \boldsymbol{y}_K^{(t)} \right\| \leq \mathcal{O}(\frac{1}{\sqrt{K}}).$$

App. B gives the details on the constants that appear in the rates and the proof of Theorem 3.1.

## 3.2 LAST ITERATE CONVERGENCE RATE OF INEXACT ACVI

For some problems, the equation in line 8 or the convex optimization problem in line 9 of ACVI may not have an analytic solution, or the exact solution may be expensive to compute. Thus we consider solving these two problems approximately, using warm-starting. At each iteration, we set the initial variable $\boldsymbol{x}$ and $\boldsymbol{y}$ to be the solution at the previous step when solving the $\boldsymbol{x}$ and $\boldsymbol{y}$ sub-problems, respectively, as described in Algorithm 1. The following Theorem—inspired by (Schmidt et al., 2011)—establishes that when the errors in the calculation of the subproblems satisfy certain conditions, the last iterate convergence rate of inexact ACVI recovers that of (exact) ACVI. The theorem holds for the standard barrier function used for IP methods, as well as for a new barrier function ($\wp_2$) that we propose that is smooth and defined in the entire domain, as follows:

$$\wp_1(\boldsymbol{z}, \mu) \triangleq -\mu \, \log(-\boldsymbol{z}) \qquad (\wp_1) \qquad \wp_2(\boldsymbol{z}, \mu) \triangleq \begin{cases} -\mu \log(-\boldsymbol{z}), & \boldsymbol{z} \leq -e^{-\frac{c}{\mu}} \\ \mu e^{\frac{c}{\mu}} \boldsymbol{z} + \mu + c, & \text{otherwise} \end{cases} \qquad (\wp_2)$$

where $c$ in ($\wp_2$) is fixed constant. Choosing among these is denoted with $\wp(\cdot)$ in Algorithm 1.

**Theorem 3.2** (Last iterate convergence rate of Inexact ACVI—Algorithm 1 with $\wp_1$ or $\wp_2$). *Given a continuous operator $F\colon \mathcal{X} \to \mathbb{R}^n$, assume: (i) F is monotone on $\mathcal{C}_=$, as per Def. 2.1; (ii) either F is strictly monotone on $\mathcal{C}$ or one of $\varphi_i$ is strictly convex; and (iii) F is L-Lipschitz on $\mathcal{X}$, that is, $\|F(\boldsymbol{x}) - F(\boldsymbol{x}')\| \leq L \|\boldsymbol{x} - \boldsymbol{x}'\|$, for all $\boldsymbol{x}, \boldsymbol{x}' \in \mathcal{X}$ and some $L > 0$. Let $(\boldsymbol{x}_K^{(t)}, \boldsymbol{y}_K^{(t)}, \boldsymbol{\lambda}_K^{(t)})$ denote the last iterate of Algorithm 1; and let $\sigma_k$ and $\varepsilon_k$ denote the approximation errors at step k of lines 8 and 9 (as per Def. 2.3 and 2.4), respectively. Further, suppose: $\lim_{K \to \infty} \frac{1}{\sqrt{K}} \sum_{k=1}^{K+1} \left(k(\sqrt{\varepsilon_k} + \sigma_k)\right) < +\infty$. Given any fixed $K \in \mathbb{N}_+$, run with sufficiently small $\mu_{-1}$, then for all $t \in [T]$, it holds:*

$$\mathcal{G}(\boldsymbol{x}_K^{(t)}, \mathcal{C}) \leq \mathcal{O}(\frac{1}{\sqrt{K}}), \text{ and } \left\| \boldsymbol{x}_K^{(t)} - \boldsymbol{y}_K^{(t)} \right\| \leq \mathcal{O}(\frac{1}{\sqrt{K}}).$$

As is the case for Theorem 3.1, Theorem 3.2 gives a nonasymptotic convergence guarantee. While the condition involving the sequences $\{\epsilon_k\}_{k=1}^{K+1}$ and $\{\sigma_k\}_{k=1}^{K+1}$ requires the given expression to be summable, the convergence rate is nonasymptotic as it holds for any $K$. App. B gives details on the constants in the rates of Theorem 3.2, provides the proof, and also discusses the algorithms $\mathcal{A}_{\boldsymbol{x}}, \mathcal{A}_{\boldsymbol{y}}$ for the sub-problems that satisfy the conditions. App. C discusses further details of the implementation of Algorithm 1; and we will analyze the effect of warm-starting in § 5.

## 4 SPECIALIZATION OF ACVI FOR SIMPLE INEQUALITY CONSTRAINTS

We now consider that the inequality constraints are simple in that the projection is fast to compute. This scenario frequently occurs in machine learning, particularly when dealing with $L_\infty$-ball constraints, for instance. Projections onto the $L_2$ and $L_1$-balls can also be obtained efficiently through simple normalization for $L_2$ and a $\mathcal{O}(n \log(n))$ algorithm for $L_1$ (Duchi et al., 2008). In ACVI, we have the flexibility to substitute the $\boldsymbol{y}$-subproblem with a projection onto the set defined by the inequalities. The $\boldsymbol{x}$-subproblem still accounts for equality constraints, and if there are none, this simplifies the $\boldsymbol{x}$-subproblem further since $\boldsymbol{P}_c \equiv \boldsymbol{I}$, and $\boldsymbol{d}_c \equiv \boldsymbol{0}$. Projection-based methods cannot leverage this structural advantage of simple inequality constraints as the intersection with the equality constraints can be nontrivial.

**The P-ACVI Algorithm: omitting the** log **barrier**. Assume that the provided inequality constraints can be met efficiently through a projection $\Pi_\leq(\cdot)\colon \mathbb{R}^n \to \mathcal{C}_\leq$. In that case, we no longer need the log barrier, and we omit $\mu$ and the outer loop of ACVI over $t \in [T]$. Differentiating the remaining expression of the $\boldsymbol{y}$ subproblem with respect to $\boldsymbol{y}$ and setting it to zero gives:

---

**Algorithm 2** P-ACVI: ACVI with simple inequalities.

1: **Input:** operator $F: \mathcal{X} \to \mathbb{R}^n$, constraints $\boldsymbol{Cx} = \boldsymbol{d}$ and projection operator $\Pi_\leq$ for the inequality constraints, hyperparameter $\beta > 0$, and number of iterations $K$.
2: **Initialize:** $\boldsymbol{y}_0 \in \mathbb{R}^n, \boldsymbol{\lambda}_0 \in \mathbb{R}^n$
3: $\boldsymbol{P}_c \triangleq \boldsymbol{I} - \boldsymbol{C}^\mathsf{T}(\boldsymbol{CC}^\mathsf{T})^{-1}\boldsymbol{C}$          where $\boldsymbol{P}_c \in \mathbb{R}^{n \times n}$
4: $\boldsymbol{d}_c \triangleq \boldsymbol{C}^\mathsf{T}(\boldsymbol{CC}^\mathsf{T})^{-1}\boldsymbol{d}$             where $\boldsymbol{d}_c \in \mathbb{R}^n$
5: **for** $k = 0, \dots, K-1$ **do**
6:      Set $\boldsymbol{x}_{k+1}$ to be the solution of: $\boldsymbol{x} + \frac{1}{\beta}\boldsymbol{P}_c F(\boldsymbol{x}) - \boldsymbol{P}_c \boldsymbol{y}_k + \frac{1}{\beta}\boldsymbol{P}_c \boldsymbol{\lambda}_k - \boldsymbol{d}_c = \boldsymbol{0}$ (w.r.t. $\boldsymbol{x}$)
7:      $\boldsymbol{y}_{k+1} = \Pi_\leq(\boldsymbol{x}_{k+1} + \frac{1}{\beta}\boldsymbol{\lambda}_k)$
8:      $\boldsymbol{\lambda}_{k+1} = \boldsymbol{\lambda}_k + \beta(\boldsymbol{x}_{k+1} - \boldsymbol{y}_{k+1})$
9: **end for**

---

$$\underset{\boldsymbol{y}}{argmin} \; \frac{\beta}{2} \left\| \boldsymbol{y} - \boldsymbol{x}_{k+1} - \frac{1}{\beta}\boldsymbol{\lambda}_k \right\|^2 = \boldsymbol{x}_{k+1} + \frac{1}{\beta}\boldsymbol{\lambda}_k \,.$$

This implies that line 9 of the exact ACVI algorithm (given in App. A) can be replaced with the solution of the $\boldsymbol{y}$ problem *without* the inequality constraints, and we can cheaply project to satisfy the inequality constraints, as follows: $\boldsymbol{y}_{k+1} = \Pi_\leq(\boldsymbol{x}_{k+1} + \frac{1}{\beta}\boldsymbol{\lambda}_k)$, where the $\varphi_i(\cdot)$ are included in the projection. We describe the resulting procedure in Algorithm 2 and refer to it as *P-ACVI*. In this scenario with simple $\varphi_i$, the $\boldsymbol{y}$ problem is always solved exactly; nonetheless, when the $\boldsymbol{x}$-subproblem is also solved approximately, we refer to it as *PI-ACVI*.

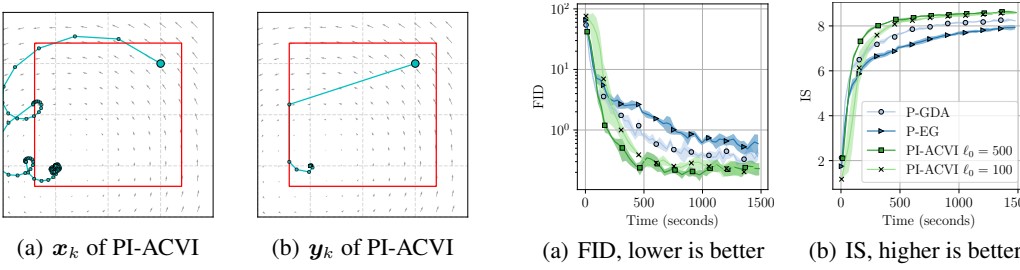

(a) $\boldsymbol{x}_k$ of PI-ACVI      (b) $\boldsymbol{y}_k$ of PI-ACVI      (a) FID, lower is better      (b) IS, higher is better

Figure 2: **Intermediate iterates of PI-ACVI (Algorithm 2) on the 2D minmax game** (2D-BG). The boundary of the constraint set is shown in red. **(b)** depicts the $\boldsymbol{y}_k$ (from line 7 in Algorithm 2) which we obtain through projections. In **(a)**, each spiral corresponds to iteratively solving the $\boldsymbol{x}_k$ subproblem for $\ell = 20$ steps (line 6 in Algorithm 2). Jointly, the trajectories of $\boldsymbol{x}$ and $\boldsymbol{y}$ illustrate the ACVI dynamics: $\boldsymbol{x}$ and the constrained $\boldsymbol{y}$ "collaborate" and converge to the same point.

Figure 3: **Experiments on the** (C-GAN) **game**, using GDA, EG, and PI-ACVI on MNIST. All curves are averaged over 4 seeds. **(a)**: Frechet Inception Distance (FID, lower is better) given CPU wall-clock time. **(b)**: Inception Score (IS, higher is better) given wall-clock time. We observe that PI-ACVI converges faster than EG and GDA for both metrics. Moreover, we see that using a large $\ell$ for the first iteration ($\ell_0$) can give a significant advantage. The two PI-ACVI curves use the same $\ell_+ = 20$.

**Last-iterate convergence of P-ACVI.** The following theorem shows that P-ACVI has the same last-iterate rate as ACVI. Its proof can be derived from that of Theorem 3.1, which focuses on a more general setting, see App. B. We state it as a separate theorem, as it cannot be deduced directly from the statement of the former.

**Theorem 4.1** (Last iterate convergence rate of P-ACVI—Algorithm 2)**.** *Given a continuous operator $F: \mathcal{X} \to \mathbb{R}^n$, assume $F$ is monotone on $\mathcal{C}_=$, as per Def. 2.1. Let $(\boldsymbol{x}_K, \boldsymbol{y}_K, \boldsymbol{\lambda}_K)$ denote the last iterate of Algorithm 2. Then for all $K \in \mathbb{N}_+$, it holds that:*

$$\mathcal{G}(\boldsymbol{x}_K, \mathcal{C}) \leq \mathcal{O}(\frac{1}{\sqrt{K}}), \; and \; \left\| \boldsymbol{x}^K - \boldsymbol{y}^K \right\| \leq \mathcal{O}(\frac{1}{\sqrt{K}}) \,.$$

*Remark* 4.2. Note that Theorem 4.1 relies on weaker assumptions than Theorem. 3.1. This is a ramification of removing the central path in the P-ACVI Algorithm. Thus, assumption *(ii)* in Theorem 3.1—used earlier to guarantee the existence of the central path (see App. A)—is not needed.

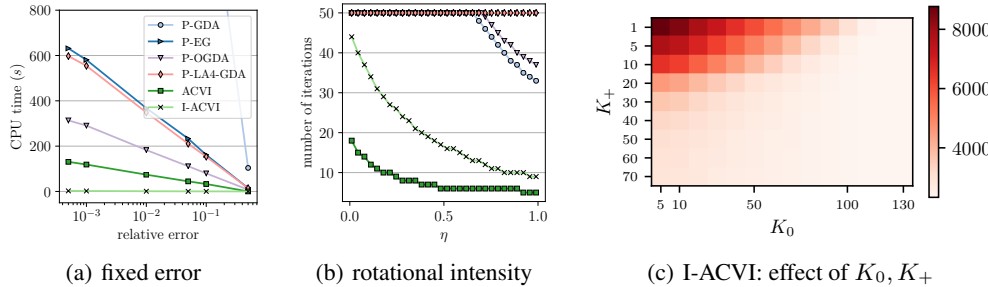

(a) fixed error      (b) rotational intensity      (c) I-ACVI: effect of $K_0, K_+$

Figure 4: **Comparison between I-ACVI, (exact) ACVI, and projection-based algorithms on the** (HBG) **problem. (a)**: CPU time (in seconds) to reach a given relative error ($x$-axis), where the rotational intensity is fixed to $\eta = 0.05$ in (HBG) for all methods. **(b)**: Number of iterations to reach a relative error of $0.02$ for varying values of the rotational intensity $\eta$. We fix the maximum number of iterations to $50$. **(c)**: joint impact of the number of inner-loop iterations $K_0$ at $t = 0$ and different choices of inner-loop iterations for $K_+$ at any $t > 0$ on the number of iterations needed to reach a fixed relative error of $10^{-4}$. We see that irrespective of the selection of $K_+$, I-ACVI converges fast if $K_0$ is large enough. For instance, ($K_0 = 130, K_+ = 1$) converges faster than ($K_0 = 20, K_+ = 20$). We fix $\ell = 10$ for all the experiments, in all of **(a)**, **(b)**, and **(c)**.

## 5 EXPERIMENTS

**Methods.** We compare ACVI, Inexact-ACVI (I-ACVI), and Projected-Inexact-ACVI (PI-ACVI) with the projected variants of Gradient Descent Ascent (P-GDA), Extragradient (Korpelevich, 1976) (P-EG), Optimistic-GDA (Popov, 1980) (P-OGDA), and Lookahead-Minmax (Zhang et al., 2019; Chavdarova et al., 2021) (P-LA). We always use GDA as an inner optimizer for I-ACVI, PI-ACVI, and P-ACVI. See App. D and C for comparison with additional methods and implementation.

**Problems.** We study the empirical performance of these methods on three different problems:

- *2D bilinear game*: a version of the bilinear game with $L_\infty$ constraints, as follows

$$\min_{x_1 \in \triangle} \max_{x_2 \in \triangle} x_1 x_2, \qquad \text{with} \quad \triangle = \{x \in \mathbb{R}| -0.4 \le x \le 2.4\}. \tag{2D-BG}$$

- *High-dimensional bilinear game*: where each player is a 500-dimensional vector. The iterates are constrained to the probability simplex. A parameter $\eta \in (0, 1)$ controls the rotational component of the game (when $\eta = 1$ the game is a potential, when $\eta = 0$ the game is Hamiltonian):

$$\min_{\boldsymbol{x}_1 \in \triangle} \max_{\boldsymbol{x}_2 \in \triangle} \eta \boldsymbol{x}_1^\intercal \boldsymbol{x}_1 + (1 - \eta) \boldsymbol{x}_1^\intercal \boldsymbol{x}_2 - \eta \boldsymbol{x}_2^\intercal \boldsymbol{x}_2, \text{ with } \triangle = \{\boldsymbol{x}_i \in \mathbb{R}^{500}|\boldsymbol{x}_i \ge \boldsymbol{0}, \text{ and } \boldsymbol{e}^\intercal \boldsymbol{x}_i = 1\}. \tag{HBG}$$

- *MNIST*. We train GANs on the MNIST (Lecun & Cortes, 1998) dataset. We use linear inequality constraints and no equality constraints, as follows:

$$\min_{G \in \triangle_{\boldsymbol{\theta}}} \max_{D \in \triangle_{\boldsymbol{\zeta}}} \mathbb{E}_{\boldsymbol{s} \sim p_d} \big[\log D(\boldsymbol{s})\big] + \mathbb{E}_{\boldsymbol{z} \sim p_z} \big[\log(1 - D(G(\boldsymbol{z})))\big] \tag{C-GAN}$$

$$\text{where } \triangle_{\boldsymbol{\theta}} = \{\boldsymbol{\theta}|\boldsymbol{A}_1 \boldsymbol{\theta} \le \boldsymbol{b}_1\}, \triangle_{\boldsymbol{\zeta}} = \{\boldsymbol{\zeta}|\boldsymbol{A}_2 \boldsymbol{\zeta} \le \boldsymbol{b}_2\},$$

with $p_z, p_d$ respectively, noise and data distributions; $\boldsymbol{\theta}$ and $\boldsymbol{\zeta}$ are the parameters of the generator and discriminator, resp. $D$ and $G$ are the Generator and Discriminator maps, parameterized with $\boldsymbol{\theta}$ and $\boldsymbol{\zeta}$, resp. $\boldsymbol{A}_i \in \mathbb{R}^{100 \times n_i}$ and $\boldsymbol{b}_i \in \mathbb{R}^{n_i}$, where $n_i$ is the number of parameters of $D$ or $G$.

### 5.1 INEXACT ACVI

**2D bilinear game.** In Fig. 1, we compare exact and inexact ACVI on the 2D-Bilinear game. Rather than solving the subproblems of I-ACVI until we reach appropriate accuracy of the solutions of the subproblems, herein, we fix the $K$ and $\ell$ number of iterations in I-ACVI. We observe how I-ACVI can converge following the central path when the inner loop of I-ACVI over $k \in [K]$ is solved with sufficient precision. The two parameters influencing the convergence of the iterates to the central path are $K$ and $\ell$, where the latter is the number of iterations to solve the two subproblems (line 8 and line 9 in Algorithm 1). Fig. 1 shows that small values such as $K = 20$ and $\ell = 2$ are sufficient for convergence for this purely rotational game. Nonetheless, as $K$ and $\ell$ decrease further, the iterates

of I-ACVI may not converge. This accords with Theorem 3.2, which indicates that the sum of errors is bounded only if $K$ is large. Hence, larger $K$ implies a smaller error.

**HD bilinear game.** In Fig. 4(a) and Fig. 4(b) we compare I-ACVI with ACVI and the projection-based algorithms on the (HBG) problem. We observe that both ACVI and I-ACVI outperform the remaining baselines significantly in terms of speed of convergence measured in both CPU time and the number of iterations. Moreover, while I-ACVI requires more iterations than ACVI to reach a given relative error, those iterations are computationally cheaper relative to solving exactly each subproblem; hence, I-ACVI converges much faster than any other method. Fig. 4(c) aims to demonstrate that the subproblems of I-ACVI are suitable for warm-starting. Interestingly, we notice that the choice of the number of iterations at the first step $t = 0$ plays a crucial role. Given that we initialize variables at each iteration with the previous solution, it aids the convergence to solve the subproblems as accurately as possible at $t = 0$. This initial accuracy reduces the initial error, subsequently decreasing the error at all subsequent iterations. We revisit this observation in § 5.3.

## 5.2 Projected-Inexact-ACVI

**2D bilinear game.** In Fig. 2 we show the dynamics of PI-ACVI on the 2D game defined by (2D-BG). Compared to ACVI in Fig. 1, the iterates converge to the solution without following the central path. A comparison with other optimizers is available in App. D.

**MNIST.** In Fig. 3 we compare PI-ACVI and baselines on the (C-GAN) game trained on the MNIST dataset. We employ the greedy projection algorithm (Beck, 2017) for the projections. Since ACVI was derived primarily for handling general constraints, a question that arises is how it (and its variants) performs when the projection is fast to compute. Although the projection is fast to compute for these experiments, PI-ACVI converges faster than the projection-based methods. Compared to the projected EG method, which only improves upon GDA when the rotational component of $F$ is high, it gives more consistent improvements over the GDA baseline.

## 5.3 Effect of Warm-up on I-ACVI and PI-ACVI

**I-ACVI.** The experiments in Fig. 1 motivate increasing the number of iterations $K$ only at the first iteration $t = 0$—denoted $K_0$, so that the early iterates are close to the central path. Recall that the $K$ steps (corresponding to line 7 in Algorithm 1) bring the iterates closer to the central path as $K \rightarrow \infty$ (see App. B). After those $K_0$ steps, $\mu$ is decayed, which moves the problem's solution along the central path. For I-ACVI, from Fig. 4(c)—where $\ell$ is fixed to 10—we observed that regardless of the selected value of $K_+$ for $t > 0$, it can be compensated by a large enough $K_0$.

**PI-ACVI.** We similarly study the impact of the warmup technique for the PI-ACVI method (Algorithm 2). Compared to I-ACVI, this method omits the outer loop over $t \in [T]$. Hence, instead of varying $K_0$, we experiment with increasing the first $\ell$ at iteration $k = 0$, denoted by $\ell_0$. In Fig. 3 we solve the constrained MNIST problem with PI-ACVI using either $\ell_0 = 500$ or $\ell_0 = 100$, $\ell_+$ is set to 20 in both cases. Increasing the $\ell_0$ value significantly improves the convergence speed.

**Conclusion.** We observe consistently that using a large $K_0$ or I-ACVI, or large $l_0$ for PI-ACVI aids the convergence. Conversely, factors such as $l$ and $K_+$ in I-ACVI, or $l_+$ in PI-ACVI, exert a comparatively lesser influence. Further experiments and discussions are available in App. D.

## 6 Discussion

We contributed to an emerging line of research on the ACVI method, showing that the last iterate of ACVI converges at a rate of order $\mathcal{O}(1/\sqrt{K})$ for monotone VIs. This result is significant because it does not rely on the first-order smoothness of the operator, resolving an open problem in the VI literature. To address subproblems that cannot always be solved in closed form, we introduced an inexact ACVI (I-ACVI) variant that uses warm-starting for its subproblems and proved last iterate convergence under certain weak assumptions. We also proposed P-ACVI for simple inequality constraints and showed that it converges with $\mathcal{O}(1/\sqrt{K})$ rate. Our experiments provided insights into I-ACVI's behavior when subproblems are solved approximately, emphasized the impact of warm-starting, and highlighted advantages over standard projection-based algorithms.

ACKNOWLEDGMENTS

We acknowledge support from the Swiss National Science Foundation (SNSF), grants P2ELP2_199740 and P500PT_214441 The work of T. Yang is supported in part by the NSF grant CCF-2007911 to Y. Chi.

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

## A  ADDITIONAL BACKGROUND

In this section, we give background in addition to that presented in the main part. This includes:

  (i) in A.1 we describe the ADMM method,
 (ii) in App. A.2 we list relevant definitions,
(iii) details of the ACVI method, including its derivation, required for the proofs of the theorems in this paper are explained in App. A.3, and
 (iv) the baseline methods used in § 5 of the main part are described in App. A.5.

### A.1  ALTERNATING DIRECTION METHOD OF MULTIPLIERS–ADMM

**The ADMM method.**  ADMM (Glowinski & Marroco, 1975; Gabay & Mercier, 1976; Lions & Mercier, 1979; Glowinski & Le Tallec, 1989) was proposed for objectives separable into two or more different functions whose arguments are nondisjoint, as follows:

$$\min_{\boldsymbol{x},\boldsymbol{y}} f(\boldsymbol{x}) + g(\boldsymbol{y}) \quad s.t. \quad \boldsymbol{A}\boldsymbol{x} + \boldsymbol{B}\boldsymbol{y} = \boldsymbol{b}\,, \tag{ADMM-Pr}$$

where $f, g : \mathbb{R}^n \to \mathbb{R}$ are often assumed convex, $\boldsymbol{x}, \boldsymbol{y} \in \mathbb{R}^n$, $\boldsymbol{A}, \boldsymbol{B} \in \mathbb{R}^{n' \times n}$, and $\boldsymbol{b} \in \mathbb{R}^{n'}$. ADMM relies on the augmented Lagrangian function $\mathcal{L}_\beta(\cdot)$:

$$\mathcal{L}_\beta(\boldsymbol{x}, \boldsymbol{y}, \boldsymbol{\lambda}) = f(\boldsymbol{x}) + g(\boldsymbol{y}) + \langle \boldsymbol{A}\boldsymbol{x} + \boldsymbol{B}\boldsymbol{y} - \boldsymbol{b}, \boldsymbol{\lambda} \rangle + \frac{\beta}{2}\|\boldsymbol{A}\boldsymbol{x} + \boldsymbol{B}\boldsymbol{y} - \boldsymbol{b}\|^2\,, \tag{AL-CVX}$$

where $\beta > 0$. If the augmented Lagrangian method is used to solve (AL-CVX), at each step $k$ we have:

$$\boldsymbol{x}_{k+1}, \boldsymbol{y}_{k+1} = \operatorname*{argmin}_{\boldsymbol{x},\boldsymbol{y}} \ \mathcal{L}_\beta(\boldsymbol{x}, \boldsymbol{y}, \boldsymbol{\lambda}_k) \quad \text{and}$$

$$\boldsymbol{\lambda}_{k+1} = \boldsymbol{\lambda}_k + \beta(\boldsymbol{A}\boldsymbol{x}_{k+1} + \boldsymbol{B}\boldsymbol{y}_{k+1} - \boldsymbol{b})\,,$$

where the latter step is gradient ascent on the dual. In contrast, ADMM updates $\boldsymbol{x}$ and $\boldsymbol{y}$ in an alternating way as follows:

$$\boldsymbol{x}_{k+1} = \operatorname*{argmin}_{\boldsymbol{x}} \ \mathcal{L}_\beta(\boldsymbol{x}, \boldsymbol{y}_k, \boldsymbol{\lambda}_k)\,,$$

$$\boldsymbol{y}_{k+1} = \operatorname*{argmin}_{\boldsymbol{y}} \ \mathcal{L}_\beta(\boldsymbol{x}_{k+1}, \boldsymbol{y}_k, \boldsymbol{\lambda}_k)\,, \tag{ADMM}$$

$$\boldsymbol{\lambda}_{k+1} = \boldsymbol{\lambda}_k + \beta(\boldsymbol{A}\boldsymbol{x}_{k+1} + \boldsymbol{B}\boldsymbol{y}_{k+1} - \boldsymbol{b})\,,$$

where the key difference is that for the $\boldsymbol{y}$ update the latest iterate of $\boldsymbol{x}$ is used.

ADMM's popularity stems largely from its computational efficiency for large-scale machine learning problems (Boyd et al., 2011) and its rapid convergence in certain settings (e.g., Nishihara et al., 2015). In particular, it achieves linear convergence when one of the objective terms is strongly convex (Nishihara et al., 2015), and it is known in the community that it can converge faster than the proximal point method in some regression examples. It can be viewed as equivalent to the Douglas-Rachford operator splitting technique (Douglas & Rachford, 1956) applied within the dual space (see e.g. Gabay, 1983; Eckstein, 1989; Lin et al., 2022).

### A.2  ADDITIONAL VI DEFINITIONS AND EQUIVALENT FORMULATIONS

Here we give the complete statement of the definition of an $L$-Lipschitz operator for completeness, which assumption was used in Theorem 3.2.

**Definition A.1** ($L$-Lipschitz operator)**.** Let $F \colon \mathcal{X} \supseteq \mathcal{S} \to \mathbb{R}^n$ be an operator, we say that $F$ satisfies *$L$-first-order smoothness* on $\mathcal{S}$ if $F$ is an $L$-Lipschitz map; that is, there exists $L > 0$ such that:

$$\|F(\boldsymbol{x}) - F(\boldsymbol{x}')\| \le L \|\boldsymbol{x} - \boldsymbol{x}'\|\,, \qquad \forall \boldsymbol{x}, \boldsymbol{x}' \in \mathcal{S}\,.$$

To define *cocoercive* operators—mentioned in the discussions of the related work, we will first introduce the inverse of an operator.

Seeing an operator $F \colon \mathcal{X} \to \mathbb{R}^n$ as the graph $GrF = \{(\boldsymbol{x}, \boldsymbol{y}) | \boldsymbol{x} \in \mathcal{X}, \boldsymbol{y} = F(\boldsymbol{x})\}$, its inverse $F^{-1}$ is defined as $GrF^{-1} \triangleq \{(\boldsymbol{y}, \boldsymbol{x}) | (\boldsymbol{x}, \boldsymbol{y}) \in GrF\}$ (see e.g. Ryu & Yin, 2022).

**Definition A.2** ($\frac{1}{\mu}$-cocoercive operator). An operator $F \colon \mathcal{X} \supseteq \mathcal{S} \to \mathbb{R}^n$ is $\frac{1}{\mu}$-*cocoercive* (or $\frac{1}{\mu}$-*inverse strongly monotone*) on $\mathcal{S}$ if its inverse (graph) $F^{-1}$ is $\mu$-strongly monotone on $\mathcal{S}$, that is,

$$\exists \mu > 0, \quad \text{s.t.} \quad \langle \boldsymbol{x} - \boldsymbol{x}', F(\boldsymbol{x}) - F(\boldsymbol{x}') \rangle \geq \mu \left\| F(\boldsymbol{x}) - F(\boldsymbol{x}') \right\|^2, \forall \boldsymbol{x}, \boldsymbol{x}' \in \mathcal{S} \,.$$

It is star $\frac{1}{\mu}$-*cocoercive* if the above holds when setting $\boldsymbol{x}' \equiv \boldsymbol{x}^\star$ where $\boldsymbol{x}^\star$ denotes a solution, that is:

$$\exists \mu > 0, \quad \text{s.t.} \quad \langle \boldsymbol{x} - \boldsymbol{x}^\star, F(\boldsymbol{x}) - F(\boldsymbol{x}^\star) \rangle \geq \mu \left\| F(\boldsymbol{x}) - F(\boldsymbol{x}^\star) \right\|^2, \forall \boldsymbol{x} \in \mathcal{S}, \boldsymbol{x}^\star \in \mathcal{S}_{\mathcal{X},F}^\star \,.$$

Notice that cocoercivity implies monotonicity, and is thus a stronger assumption.

In the following, we will make use of the natural and normal mappings of an operator $F \colon \mathcal{X} \to \mathbb{R}^n$, where $\mathcal{X} \subset \mathbb{R}^n$. We denote the projection to the set $\mathcal{X}$ with $\Pi_{\mathcal{X}}$. Following similar notation as in (Facchinei & Pang, 2003), the natural map $F_{\mathcal{X}}^{NAT} \colon \mathcal{X} \to \mathbb{R}^n$ is defined as:

$$F_{\mathcal{X}}^{NAT} \triangleq \boldsymbol{x} - \Pi_{\mathcal{X}}\big(\boldsymbol{x} - F(\boldsymbol{x})\big), \qquad \forall \boldsymbol{x} \in \mathcal{X} \,, \tag{F-NAT}$$

whereas the normal map $F_{\mathcal{X}}^{NOR} \colon \mathbb{R}^n \to \mathbb{R}^n$ is:

$$F_{\mathcal{X}}^{NOR} \triangleq F\big(\Pi_{\mathcal{X}}(\boldsymbol{x})\big) + \boldsymbol{x} - \Pi_{\mathcal{X}}(\boldsymbol{x}), \qquad \forall \boldsymbol{x} \in \mathbb{R}^n \,. \tag{F-NOR}$$

Moreover, we have the following solution characterizations:

(i) $\boldsymbol{x}^\star \in \mathcal{S}_{\mathcal{X},F}^\star$    iff    $F_{\mathcal{X}}^{NAT}(\boldsymbol{x}^\star) = \boldsymbol{0}$, and

(ii) $\boldsymbol{x}^\star \in \mathcal{S}_{\mathcal{X},F}^\star$    iff    $\exists \boldsymbol{x}' \in R^n$ s.t. $\boldsymbol{x}^\star = \Pi_{\mathcal{X}}(\boldsymbol{x}')$ and $F_{\mathcal{X}}^{NOR}(\boldsymbol{x}') = \boldsymbol{0}$.

### A.3    DETAILS ON ACVI

For completeness, herein we state the *ACVI* algorithm and show its derivation, see (Yang et al., 2023) for details. We will use these equations also for the proofs of our main results.

**Derivation of ACVI.** We first restate the cVI problem in a form that will allow us to derive an interior-point procedure. By the definition of cVI it follows (see §1.3 in Facchinei & Pang, 2003) that:

$$\boldsymbol{x} \in \mathcal{S}_{\mathcal{C},F}^\star \Leftrightarrow \begin{cases} \boldsymbol{w} = \boldsymbol{x} \\ \boldsymbol{x} = \underset{\boldsymbol{z}}{arg\,min} F(\boldsymbol{w})^{\mathsf{T}} \boldsymbol{z} \\ s.t. \quad \varphi(\boldsymbol{z}) \leq \boldsymbol{0} \\ \quad\quad \boldsymbol{C}\boldsymbol{z} = \boldsymbol{d} \end{cases} \Leftrightarrow \begin{cases} F(\boldsymbol{x}) + \nabla \varphi^{\mathsf{T}}(\boldsymbol{x})\boldsymbol{\lambda} + \boldsymbol{C}^{\mathsf{T}}\boldsymbol{\nu} = \boldsymbol{0} \\ \boldsymbol{C}\boldsymbol{x} = \boldsymbol{d} \\ \boldsymbol{0} \leq \boldsymbol{\lambda} \perp \varphi(\boldsymbol{x}) \leq \boldsymbol{0}, \end{cases} \tag{KKT}$$

where $\boldsymbol{\lambda} \in \mathbb{R}^m$ and $\boldsymbol{\nu} \in \mathbb{R}^p$ are dual variables. Recall that we assume that $int\,\mathcal{C} \neq \emptyset$, thus, by the Slater condition (using the fact that $\varphi_i(\boldsymbol{x}), i \in [m]$ are convex) and the KKT conditions, the second equivalence holds, yielding the KKT system of cVI. Note that the above equivalence also guarantees the two solutions coincide; see Facchinei & Pang (2003, Prop. 1.3.4 (b)).

Analogous to the method described in § 2, we add a log-barrier term to the objective to remove the inequality constraints and obtain the following modified version of (KKT):

$$\begin{cases} \boldsymbol{w} = \boldsymbol{x} \\ \boldsymbol{x} = \underset{\boldsymbol{z}}{arg\,min} \; F(\boldsymbol{w})^{\mathsf{T}} \boldsymbol{z} - \mu \sum_{i=1}^{m} \log\big(-\varphi_i(\boldsymbol{z})\big) \\ s.t. \quad \boldsymbol{C}\boldsymbol{z} = \boldsymbol{d} \end{cases} \Leftrightarrow \begin{cases} F(\boldsymbol{x}) + \nabla \varphi^{\mathsf{T}}(\boldsymbol{x})\boldsymbol{\lambda} + \boldsymbol{C}^{\mathsf{T}}\boldsymbol{\nu} = \boldsymbol{0} \\ \boldsymbol{\lambda} \odot \varphi(\boldsymbol{x}) + \mu \boldsymbol{e} = \boldsymbol{0} \\ \boldsymbol{C}\boldsymbol{x} - \boldsymbol{d} = \boldsymbol{0} \\ \varphi(\boldsymbol{x}) < \boldsymbol{0}, \boldsymbol{\lambda} > \boldsymbol{0}, \end{cases} \tag{KKT-2}$$

with $\mu > 0$, $\boldsymbol{e} \triangleq [1, \dots, 1]^{\mathsf{T}} \in \mathbb{R}^m$. The equivalence holds by the KKT and the Slater condition.

The update rule at step $k$ is derived by the following subproblem:

$$\min_{\boldsymbol{x}} \; F(\boldsymbol{w}_k)^{\mathsf{T}} \boldsymbol{x} - \mu \sum_{i=1}^{m} \log\big(-\varphi_i(\boldsymbol{x})\big),$$
$$s.t. \; \boldsymbol{C}\boldsymbol{x} = \boldsymbol{d} \,,$$

where we fix $\boldsymbol{w} = \boldsymbol{w}_k$. Notice that (i) $\boldsymbol{w}_k$ is a constant vector in this subproblem, and (ii) the objective is split, making ADMM a natural choice to solve the subproblem. To apply an ADMM-type method, we introduce a new variable $\boldsymbol{y} \in \mathbb{R}^n$ yielding:

$$\begin{cases} \min_{\boldsymbol{x},\boldsymbol{y}} \ F(\boldsymbol{w}_k)^\mathsf{T}\boldsymbol{x} + \mathbb{1}[\boldsymbol{C}\boldsymbol{x} = \boldsymbol{d}] - \mu \sum_{i=1}^m \log\big(-\varphi_i(\boldsymbol{y})\big), \\ s.t. \quad \boldsymbol{x} = \boldsymbol{y} \end{cases}$$ (ACVI:subproblem)

where:

$$\mathbb{1}[\boldsymbol{C}\boldsymbol{x} = \boldsymbol{d}] \triangleq \begin{cases} 0, & \text{if } \boldsymbol{C}\boldsymbol{x} = \boldsymbol{d} \\ +\infty, & \text{if } \boldsymbol{C}\boldsymbol{x} \neq \boldsymbol{d} \end{cases},$$

is a generalized real-valued convex function of $\boldsymbol{x}$.

As in Algorithm 1, for ACVI we also have the same projection matrix:

$$\boldsymbol{P}_c \triangleq \boldsymbol{I} - \boldsymbol{C}^\mathsf{T}(\boldsymbol{C}\boldsymbol{C}^\mathsf{T})^{-1}\boldsymbol{C},$$ ($\boldsymbol{P}_c$)

and:

$$\boldsymbol{d}_c \triangleq \boldsymbol{C}^\mathsf{T}(\boldsymbol{C}\boldsymbol{C}^\mathsf{T})^{-1}\boldsymbol{d},$$ ($d_c$-EQ)

where $\boldsymbol{P}_c \in \mathbb{R}^{n \times n}$ and $\boldsymbol{d}_c \in \mathbb{R}^n$.

The augmented Lagrangian of (ACVI:subproblem) is thus:

$$\mathcal{L}_\beta(\boldsymbol{x}, \boldsymbol{y}, \boldsymbol{\lambda}) = F(\boldsymbol{w}_k)^\mathsf{T}\boldsymbol{x} + \mathbb{1}(\boldsymbol{C}\boldsymbol{x} = \boldsymbol{d}) - \mu \sum_{i=1}^m \log(-\varphi_i(\boldsymbol{y})) + \langle \boldsymbol{\lambda}, \boldsymbol{x} - \boldsymbol{y} \rangle + \frac{\beta}{2} \|\boldsymbol{x} - \boldsymbol{y}\|^2,$$ (AL)

where $\beta > 0$ is the penalty parameter. Finally, using ADMM, we have the following update rule for $\boldsymbol{x}$ at step $k$:

$$\boldsymbol{x}_{k+1} = \arg\min_{\boldsymbol{x}} \mathcal{L}_\beta(\boldsymbol{x}, \boldsymbol{y}_k, \boldsymbol{\lambda}_k)$$ (Def-X)

$$= \arg\min_{\boldsymbol{x} \in \mathcal{C}_=} \frac{\beta}{2} \left\| \boldsymbol{x} - \boldsymbol{y}_k + \frac{1}{\beta}(F(\boldsymbol{w}_k) + \boldsymbol{\lambda}_k) \right\|^2.$$

This yields the following update for $\boldsymbol{x}$:

$$\boldsymbol{x}_{k+1} = \boldsymbol{P}_c\left(\boldsymbol{y}_k - \frac{1}{\beta}\big(F(\boldsymbol{w}_k) + \boldsymbol{\lambda}_k\big)\right) + \boldsymbol{d}_c.$$ (X-EQ)

For $\boldsymbol{y}$ and the dual variable $\boldsymbol{\lambda}$, we have:

$$\boldsymbol{y}_{k+1} = arg\min_{\boldsymbol{y}} \mathcal{L}_\beta(\boldsymbol{x}_{k+1}, \boldsymbol{y}, \boldsymbol{\lambda}_k)$$ (Def-Y)

$$= arg\min_{\boldsymbol{y}} \left( -\mu \sum_{i=1}^m \log\big(-\varphi_i(\boldsymbol{y})\big) + \frac{\beta}{2} \left\| \boldsymbol{y} - \boldsymbol{x}_{k+1} - \frac{1}{\beta}\boldsymbol{\lambda}_k \right\|^2 \right).$$ (Y-EQ)

To derive the update rule for $\boldsymbol{w}$, $\boldsymbol{w}_k$ is set to be the solution of the following equation:

$$\boldsymbol{w} + \frac{1}{\beta}\boldsymbol{P}_c F(\boldsymbol{w}) - \boldsymbol{P}_c \boldsymbol{y}_k + \frac{1}{\beta}\boldsymbol{P}_c \boldsymbol{\lambda}_k - \boldsymbol{d}_c = \boldsymbol{0}.$$ (W-EQ)

The following theorem ensures the solution of (W-EQ) exists and is unique, see App. B in (Yang et al., 2023) for proof.

**Theorem A.3** (W-EQ: solution uniqueness)**.** *If $F$ is monotone on $\mathcal{C}_=$, the following statements hold true for the solution of (W-EQ):*

1. *it always exists,*

2. *it is unique, and*

3. *it is contained in $\mathcal{C}_=$.*

Finally, notice that $\boldsymbol{w}$ as it is redundant to be considered in the algorithm, since $\boldsymbol{w}_k = \boldsymbol{x}_{k+1}$, and it is thus removed.

---

**Algorithm 3** (exact) ACVI pseudocode (Yang et al., 2023).

---

1: **Input:** operator $F\colon \mathcal{X} \to \mathbb{R}^n$, constraints $\boldsymbol{C}\boldsymbol{x} = \boldsymbol{d}$ and $\varphi_i(\boldsymbol{x}) \leq 0, i = [m]$, hyperparameters $\mu_{-1}, \beta > 0, \delta \in (0,1)$, number of outer and inner loop iterations $T$ and $K$, resp.

2: **Initialize:** $\boldsymbol{y}_0^{(0)} \in \mathbb{R}^n, \boldsymbol{\lambda}_0^{(0)} \in \mathbb{R}^n$

3: $\boldsymbol{P}_c \triangleq \boldsymbol{I} - \boldsymbol{C}^\intercal (\boldsymbol{C}\boldsymbol{C}^\intercal)^{-1}\boldsymbol{C}$                   where $\boldsymbol{P}_c \in \mathbb{R}^{n \times n}$

4: $\boldsymbol{d}_c \triangleq \boldsymbol{C}^\intercal (\boldsymbol{C}\boldsymbol{C}^\intercal)^{-1}\boldsymbol{d}$                        where $\boldsymbol{d}_c \in \mathbb{R}^n$

5: **for** $t = 0, \ldots, T-1$ **do**

6:     $\mu_t = \delta \mu_{t-1}$

7:     **for** $k = 0, \ldots, K-1$ **do**

8:         Set $\boldsymbol{x}_{k+1}^{(t)}$ to be the solution of: $\boldsymbol{x} + \frac{1}{\beta}\boldsymbol{P}_c F(\boldsymbol{x}) - \boldsymbol{P}_c \boldsymbol{y}_k^{(t)} + \frac{1}{\beta}\boldsymbol{P}_c \boldsymbol{\lambda}_k^{(t)} - \boldsymbol{d}_c = \boldsymbol{0}$ (w.r.t. $\boldsymbol{x}$)

9:         $\boldsymbol{y}_{k+1}^{(t)} = \underset{\boldsymbol{y}}{arg min} - \mu \sum_{i=1}^m \log\left(-\varphi_i(\boldsymbol{y})\right) + \frac{\beta}{2}\left\|\boldsymbol{y} - \boldsymbol{x}_{k+1}^{(t)} - \frac{1}{\beta}\boldsymbol{\lambda}_k^{(t)}\right\|^2$

10:         $\boldsymbol{\lambda}_{k+1}^{(t)} = \boldsymbol{\lambda}_k^{(t)} + \beta(\boldsymbol{x}_{k+1}^{(t)} - \boldsymbol{y}_{k+1}^{(t)})$

11:     **end for**

12:     $(\boldsymbol{y}_0^{(t+1)}, \boldsymbol{\lambda}_0^{(t+1)}) \triangleq (\boldsymbol{y}_K^{(t)}, \boldsymbol{\lambda}_K^{(t)})$

13: **end for**

---

**The ACVI algorithm.**     Algorithm 3 describes the (exact) ACVI algorithm (Yang et al., 2023).

## A.4    EXISTENCE OF THE CENTRAL PATH

In this section, we discuss the results that establish guarantees of the existence of the central path. Let:

$$L(\boldsymbol{x}, \boldsymbol{\lambda}, \boldsymbol{\nu}) \triangleq F(\boldsymbol{x}) + \nabla\varphi^\intercal(\boldsymbol{x})\boldsymbol{\lambda} + \boldsymbol{C}^\intercal \boldsymbol{\nu}, \qquad \text{and}$$

$$h(\boldsymbol{x}) = \boldsymbol{C}^\intercal \boldsymbol{x} - \boldsymbol{d}.$$

For $(\boldsymbol{\lambda}, \boldsymbol{w}, \boldsymbol{x}, \boldsymbol{\nu}) \in \mathbb{R}^{2m+n+p}$, let

$$G(\boldsymbol{\lambda}, \boldsymbol{w}, \boldsymbol{x}, \boldsymbol{\nu}) \triangleq \begin{pmatrix} \boldsymbol{w} \circ \boldsymbol{\lambda} \\ \boldsymbol{w} + \varphi(\boldsymbol{x}) \\ L(\boldsymbol{x}, \boldsymbol{\lambda}, \boldsymbol{\nu}) \\ h(\boldsymbol{x}) \end{pmatrix} \in \mathbb{R}^{2m+n+p},$$

and

$$H(\boldsymbol{\lambda}, \boldsymbol{w}, \boldsymbol{x}, \boldsymbol{\nu}) \triangleq \begin{pmatrix} \boldsymbol{w} + \varphi(\boldsymbol{x}) \\ L(\boldsymbol{x}, \boldsymbol{\lambda}, \boldsymbol{\nu}) \\ h(\boldsymbol{x}) \end{pmatrix} \in \mathbb{R}^{m+n+p}.$$

Let $H_{++} \triangleq H(\mathbb{R}_{++}^{2m} \times \mathbb{R}^n \times \mathbb{R}^p)$.

By (Corollary 11.4.24, Facchinei & Pang, 2003) we have the following proposition.

**Proposition A.4** (sufficient condition for the existence of the central path)**.** *If $F$ is monotone, either $F$ is strictly monotone or one of $\varphi_i$ is strictly convex, and $\mathcal{C}$ is bounded. The following four statements hold for the functions $G$ and $H$:*

    *1. $G$ maps $\mathbb{R}_{++}^{2m} \times \mathbb{R}^{n+p}$ homeomorphically onto $\mathbb{R}_{++}^m \times H_{++}$;*

    *2. $\mathbb{R}_{++}^m \times H_{++} \subseteq G(\mathbb{R}_+^{2m} \times \mathbb{R}^{n+p})$;*

    *3. for every vector $\boldsymbol{a} \in \mathbb{R}_+^m$, the system*

$$H(\boldsymbol{\lambda}, \boldsymbol{w}, \boldsymbol{x}, \boldsymbol{\nu}) = \boldsymbol{0}, \quad \boldsymbol{w} \circ \boldsymbol{\lambda} = \boldsymbol{a}$$

    *has a solution $(\boldsymbol{\lambda}, \boldsymbol{w}, \boldsymbol{x}, \boldsymbol{\nu}) \in \mathbb{R}_+^{2m} \times \mathbb{R}^{n+p}$; and*

    *4. the set $H_{++}$ is convex.*

## A.5 SADDLE-POINT OPTIMIZATION METHODS

In this section, we describe in detail the saddle point methods that we compare with in the main paper in § 5. We denote the projection to the set $\mathcal{X}$ with $\Pi_\mathcal{X}$, and when the method is applied in the unconstrained setting $\Pi_\mathcal{X} \equiv \boldsymbol{I}$.

For an example of the associated vector field and its Jacobian, consider the following constrained zero-sum game:

$$\min_{\boldsymbol{x}_1 \in \mathcal{X}_1} \max_{\boldsymbol{x}_2 \in \mathcal{X}_2} f(\boldsymbol{x}_1, \boldsymbol{x}_2), \tag{ZS-G}$$

where $f : \mathcal{X}_1 \times \mathcal{X}_2 \to \mathbb{R}$ is smooth and convex in $\boldsymbol{x}_1$ and concave in $\boldsymbol{x}_2$. As in the main paper, we write $\boldsymbol{x} \triangleq (\boldsymbol{x}_1, \boldsymbol{x}_2) \in \mathbb{R}^n$. The vector field $F : \mathcal{X} \to \mathbb{R}^n$ and its Jacobian $J$ are defined as:

$$F(\boldsymbol{x}) = \begin{bmatrix} \nabla_{\boldsymbol{x}_1} f(\boldsymbol{x}) \\ -\nabla_{\boldsymbol{x}_2} f(\boldsymbol{x}) \end{bmatrix}, \qquad J(\boldsymbol{x}) = \begin{bmatrix} \nabla^2_{\boldsymbol{x}_1} f(\boldsymbol{x}) & \nabla_{\boldsymbol{x}_2} \nabla_{\boldsymbol{x}_1} f(\boldsymbol{x}) \\ -\nabla_{\boldsymbol{x}_1} \nabla_{\boldsymbol{x}_2} f(\boldsymbol{x}) & -\nabla^2_{\boldsymbol{x}_2} f(\boldsymbol{x}) \end{bmatrix}.$$

In the remaining of this section, we will only refer to the joint variable $\boldsymbol{x}$, and (with abuse of notation) the subscript will denote the step. Let $\gamma \in [0, 1]$ denote the step size.

**(Projected) Gradient Descent Ascent (GDA).** The extension of gradient descent for the cVI problem is *gradient descent ascent* (GDA). The GDA update at step $k$ is then:

$$\boldsymbol{x}_{k+1} = \Pi_\mathcal{X}\big(\boldsymbol{x}_k - \gamma F(\boldsymbol{x}_k)\big). \tag{GDA}$$

**(Projected) Extragradient (EG).** EG (Korpelevich, 1976) uses a "prediction" step to obtain an extrapolated point $\boldsymbol{x}_{k+\frac{1}{2}}$ using GDA: $\boldsymbol{x}_{k+\frac{1}{2}} = \Pi_\mathcal{X}\big(\boldsymbol{x}_k - \gamma F(\boldsymbol{x}_k)\big)$, and the gradients at the *extrapolated* point are then applied to the *current* iterate $\boldsymbol{x}_t$:

$$\boldsymbol{x}_{k+1} = \Pi_\mathcal{X}\left(\boldsymbol{x}_k - \gamma F\Big(\Pi_\mathcal{X}\big(\boldsymbol{x}_k - \gamma F(\boldsymbol{x}_k)\big)\Big)\right). \tag{EG}$$

In the original EG paper, (Korpelevich, 1976) proved that the EG method (with a fixed step size) converges for monotone VIs, as follows.

**Theorem A.5** (Korpelevich (1976)). *Given a map $F : \mathcal{X} \mapsto \mathbb{R}^n$, if the following is satisfied:*

1. *the set $\mathcal{X}$ is closed and convex,*

2. *$F$ is single-valued, definite, and monotone on $\mathcal{X}$–as per Def. 2.1,*

3. *$F$ is L-Lipschitz–as per Asm. A.1.*

*then there exists a solution $\boldsymbol{x}^\star \in \mathcal{X}$, such that the iterates $\boldsymbol{x}_k$ produced by the EG update rule with a fixed step size $\gamma \in (0, \frac{1}{L})$ converge to it, that is $\boldsymbol{x}_k \to \boldsymbol{x}^\star$, as $k \to \infty$.*

Facchinei & Pang (2003) also show that for any *convex-concave* function $f$ and any closed convex sets $\boldsymbol{x}_1 \in \mathcal{X}_1$ and $\boldsymbol{x}_2 \in \mathcal{X}_2$, the EG method converges (Facchinei & Pang, 2003, Theorem 12.1.11).

**(Projected) Optimistic Gradient Descent Ascent (OGDA).** The update rule of Optimistic Gradient Descent Ascent OGDA ((OGDA) Popov, 1980) is:

$$\boldsymbol{x}_{n+1} = \Pi_\mathcal{X}\big(\boldsymbol{x}_n - 2\gamma F(\boldsymbol{x}_n) + \gamma F(\boldsymbol{x}_{n-1})\big). \tag{OGDA}$$

**(Projected) Lookahead–Minmax (LA).** The LA algorithm for min-max optimization (Chavdarova et al., 2021), originally proposed for minimization by Zhang et al. (2019), is a general wrapper of a "base" optimizer where, at every step $t$: (i) a copy of the current iterate $\tilde{\boldsymbol{x}}_n$ is made: $\tilde{\boldsymbol{x}}_n \leftarrow \boldsymbol{x}_n$, (ii) $\tilde{\boldsymbol{x}}_n$ is updated $k \geq 1$ times, yielding $\tilde{\boldsymbol{\omega}}_{n+k}$, and finally (iii) the actual update $\boldsymbol{x}_{n+1}$ is obtained as a *point that lies on a line between* the current $\boldsymbol{x}_n$ iterate and the predicted one $\tilde{\boldsymbol{x}}_{n+k}$:

$$\boldsymbol{x}_{n+1} \leftarrow \boldsymbol{x}_n + \alpha(\tilde{\boldsymbol{x}}_{n+k} - \boldsymbol{x}_n), \quad \alpha \in [0, 1]. \tag{LA}$$

In this work, we use solely GDA as a base optimizer for LA, and denote it with *LAk-GDA*.

**Mirror-Descent.** The mirror-descent algorithm (Nemirovski & Yudin, 1983; Beck & Teboulle, 2003) can be seen as a generalization of gradient descent in which the geometry of the space is controlled by a mirror map $\Psi : \mathcal{X} \mapsto \mathbb{R}$. We define the Prox($\cdot$) mapping:

$$\text{Prox}(\boldsymbol{x}_n, \boldsymbol{g}) \triangleq \text{argmin}_{\boldsymbol{x} \in \mathcal{X}} \boldsymbol{g}^\top \boldsymbol{x} + \frac{1}{\gamma} D_\Psi(\boldsymbol{x}, \boldsymbol{x}_n),$$

where $D_\Psi$ is the Bregman divergence associated with the mirror map $\Psi : \mathcal{X} \mapsto \mathbb{R}$, characterizing the geometry of our space. The mirror descent algorithm uses the Prox mapping to obtain the next iterate:

$$\boldsymbol{x}_{n+1} \leftarrow \text{Prox}(\boldsymbol{x}_n, F(\boldsymbol{x}_n)). \tag{MD}$$

**Mirror-Prox.** Similarly to Mirror Descent, Mirror Prox (Nemirovski, 2004) generalizes extragradient to spaces where the geometry can be controlled by a mirror map $\Psi$:

$$\boldsymbol{x}_{n+1/2} \leftarrow \text{Prox}(\boldsymbol{x}_n, F(\boldsymbol{x}_n)),$$
$$\boldsymbol{x}_{n+1} \leftarrow \text{Prox}(\boldsymbol{x}_n, F(\boldsymbol{x}_{n+1/2})). \tag{MP}$$

## B  MISSING PROOFS

In this section, we provide the proofs of Theorems 3.1, 3.2 and 4.1, stated in the main part. In the subsections B.3 and B.4, we also discuss the practical implications of Theorems 3.1 and 3.2, and the algorithms that can be used for the subproblems in Algorithm 1, respectively.

### B.1  PROOF OF THEOREM 3.1: LAST-ITERATE CONVERGENCE OF ACVI FOR MONOTONE VARIATIONAL INEQUALITIES

Recall from Theorem 3.1 that we have the following assumptions:

- F is monotone on $\mathcal{C}_=$, as per Def. 2.1; and
- either $F$ is strictly monotone on $\mathcal{C}$ or one of $\varphi_i$ is strictly convex.

#### B.1.1  SETTING AND NOTATIONS

Before we proceed with the lemmas needed for the proof of Theorem 3.1, herein we introduce some definitions and notations.

**Subproblems and definitions.**  We remark that the ACVI derivation—given in App. A.3—is helpful for following the proof herein. Recall from it, that in order to derive the update rule for $\boldsymbol{x}$, we introduced a new variable $\boldsymbol{w}$, and the relevant subproblem that yields the update rule for $\boldsymbol{x}$ includes a term $\langle F(\boldsymbol{w}), \boldsymbol{x} \rangle$, where $F$ is evaluated at some fixed point. As the proof relates the $\boldsymbol{x}_k^{(t)}$ iterate of ACVI with the solution $\boldsymbol{x}_t^\mu$ of (KKT-2), in the following we will define two different maps each with fixed $\boldsymbol{w} \equiv \boldsymbol{x}^{\mu_t}$ and $\boldsymbol{w} \equiv \boldsymbol{x}_{k+1}^{(t)}$. That is, for convenience, we define the following maps from $\mathbb{R}^n$ to $\mathbb{R}$:

$$f^{(t)}(\boldsymbol{x}) \triangleq F(\boldsymbol{x}^{\mu_t})^\mathsf{T} \boldsymbol{x} + \mathbb{1}(\boldsymbol{C}\boldsymbol{x} = \boldsymbol{d}), \tag{$f^{(t)}$}$$

$$f_k^{(t)}(\boldsymbol{x}) \triangleq F(\boldsymbol{x}_{k+1}^{(t)})^\mathsf{T} \boldsymbol{x} + \mathbb{1}(\boldsymbol{C}\boldsymbol{x} = \boldsymbol{d}), \quad \text{and} \tag{$f_k^{(t)}$}$$

$$g^{(t)}(\boldsymbol{y}) \triangleq -\mu_t \sum_{i=1}^m \log\left(-\varphi_i(\boldsymbol{y})\right) = \sum_{i=1}^m \wp_1(\varphi_i(\boldsymbol{y}), \mu_t), \tag{$g^{(t)}$}$$

where $\boldsymbol{x}^{\mu_t}$ is a solution of (KKT-2) when $\mu = \mu_t$, and $\boldsymbol{x}_{k+1}^{(t)}$ is the solution of the $\boldsymbol{x}$-subproblem in ACVI at step $(t, k)$–see line 8 in Algorithm 3. Note that the existence of $\boldsymbol{x}^{\mu_t}$ is guaranteed by the existence of the central path-see App. A.4. Also, notice that $f^{(t)}, f_k^{(t)}$ and $g^{(t)}$ are all convex functions. In the following, unless otherwise specified, we drop the superscript $(t)$ of $\boldsymbol{x}_{k+1}^{(t)}$, $f^{(t)}$, $f_k^{(t)}$ and subscript $t$ of $\mu_t$ to simplify the notation.

In the remainder of this section, we introduce the notation of the solution points of the above KKT systems and that of the ACVI iterates.

Let $\boldsymbol{y}^\mu = \boldsymbol{x}^\mu$. In this case, from (KKT-2) we can see that $(\boldsymbol{x}^\mu, \boldsymbol{y}^\mu)$ is an optimal solution of:

$$\begin{cases} \min_{\boldsymbol{x},\boldsymbol{y}} f(\boldsymbol{x}) + g(\boldsymbol{y}) \\ \quad s.t. \qquad \boldsymbol{x} = \boldsymbol{y} \end{cases}. \tag{$f$-Pr}$$

There exists $\boldsymbol{\lambda}^\mu \in \mathbb{R}^n$ such that $(\boldsymbol{x}^\mu, \boldsymbol{y}^\mu, \boldsymbol{\lambda}^\mu)$ is a KKT point of ($f$-Pr). By Prop. A.4, $\boldsymbol{x}^\mu = \boldsymbol{y}^\mu$ converges to a solution of (KKT). We denote this solution by $\boldsymbol{x}^\star$. Then $(\boldsymbol{x}^\mu, \boldsymbol{y}^\mu, \boldsymbol{\lambda}^\mu)$ converges to the KKT point of (ACVI:subproblem) with $\boldsymbol{w}_k = \boldsymbol{x}^\star$. Let $(\boldsymbol{x}^\star, \boldsymbol{y}^\star, \boldsymbol{\lambda}^\star)$ denote this KKT point, where $\boldsymbol{x}^\star = \boldsymbol{y}^\star$.

On the other hand, let us denote with $(\boldsymbol{x}_k^\mu, \boldsymbol{y}_k^\mu, \boldsymbol{\lambda}_k^\mu)$ the KKT point of the analogous problem of $f_k(\cdot)$ of:

$$\begin{cases} \min_{\boldsymbol{x},\boldsymbol{y}} f_k(\boldsymbol{x}) + g(\boldsymbol{y}) \\ \quad s.t. \qquad \boldsymbol{x} = \boldsymbol{y} \end{cases}. \tag{$f_k$-Pr}$$

Note that the KKT point $(\boldsymbol{x}_k^\mu, \boldsymbol{y}_k^\mu, \boldsymbol{\lambda}_k^\mu)$ is guaranteed to exist by Slater's condition. Also, recall from the derivation of ACVI that ($f_k$-Pr) is "non-symmetric" for $\boldsymbol{x}, \boldsymbol{y}$ when using ADMM-like approach, in the sense that: when we derive the update rule for $\boldsymbol{x}$ we fix $\boldsymbol{y}$ to $\boldsymbol{y}_k$ (see Def-X), but when we derive the update rule for $\boldsymbol{y}$ we fix $\boldsymbol{x}$ to $\boldsymbol{x}_{k+1}$ (see Def-Y). This fact is used later in (LB.3-1) and LB.3-2 in Lemma B.3 for example.

Then, for the solution point, which we denoted with $(\boldsymbol{x}_k^\mu, \boldsymbol{y}_k^\mu, \boldsymbol{\lambda}_k^\mu)$, we also have that $\boldsymbol{x}_k^\mu = \boldsymbol{y}_k^\mu$. By noticing that the objective above is equivalent to $F(\boldsymbol{x}_{k+1})^\intercal \boldsymbol{x} + \mathbb{1}(\boldsymbol{C}\boldsymbol{x} = \boldsymbol{d}) - \mu_t \sum_{i=1}^m \log\big(-\varphi_i(\boldsymbol{y})\big)$, it follows that the above problem ($f_k$-Pr) is an approximation of:

$$\begin{cases} \min_{\boldsymbol{x}} \langle F(\boldsymbol{x}_{k+1}), \boldsymbol{x}\rangle + \mathbb{1}(\boldsymbol{C}\boldsymbol{x} = \boldsymbol{d}) + \mathbb{1}(\varphi(\boldsymbol{y}) \leq \boldsymbol{0}) \\ \qquad s.t. \qquad \boldsymbol{x} = \boldsymbol{y} \end{cases} , \qquad (f_k\text{-Pr-2})$$

where, as a reminder, the constraint set $\mathcal{C} \subseteq \mathcal{X}$ is defined as an intersection of finitely many inequalities and linear equalities:

$$\mathcal{C} = \{\boldsymbol{x} \in \mathbb{R}^n | \varphi_i(\boldsymbol{x}) \leq 0, i \in [m], \ \boldsymbol{C}\boldsymbol{x} = \boldsymbol{d}\}, \qquad (\text{CS})$$

where each $\varphi_i : \mathbb{R}^n \mapsto \mathbb{R}$, $\boldsymbol{C} \in \mathbb{R}^{p \times n}$, $\boldsymbol{d} \in \mathbb{R}^p$, and we assumed $rank(\boldsymbol{C}) = p$.

In fact, when $\mu \to 0+$, corollary 2.11 in (Chu, 1998) guarantees that $(\boldsymbol{x}_k^\mu, \boldsymbol{y}_k^\mu, \boldsymbol{\lambda}_k^\mu)$ converges to a KKT point of problem ($f_k$-Pr-2)—which immediately follows here since ($f_k$-Pr-2) is a convex problem. Let $(\boldsymbol{x}_k^\star, \boldsymbol{y}_k^\star, \boldsymbol{\lambda}_k^\star)$ denote this KKT point, where $\boldsymbol{x}_k^\star = \boldsymbol{y}_k^\star$.

**Summary.** To conclude, $(\boldsymbol{x}^\mu, \boldsymbol{y}^\mu, \boldsymbol{\lambda}^\mu)$—the solution of ($f$-Pr), converges to $(\boldsymbol{x}^\star, \boldsymbol{y}^\star, \boldsymbol{\lambda}^\star)$, a KKT point of (ACVI:subproblem) with $\boldsymbol{w}_k = \boldsymbol{x}^\star$, where $\boldsymbol{x}^\star = \boldsymbol{y}^\star \in \mathcal{S}_{\mathcal{C},F}^\star$; $(\boldsymbol{x}_k^\mu, \boldsymbol{y}_k^\mu, \boldsymbol{\lambda}_k^\mu)$ converges to $(\boldsymbol{x}_k^\star, \boldsymbol{y}_k^\star, \boldsymbol{\lambda}_k^\star)$—a KKT point of problem ($f_k$-Pr-2), where $(\boldsymbol{x}_k^\mu, \boldsymbol{y}_k^\mu, \boldsymbol{\lambda}_k^\mu)$ (in which $\boldsymbol{x}_k^\mu = \boldsymbol{y}_k^\mu$) is a KKT point of problem ($f_k$-Pr). Table 1 summarizes the notation for convenience.

| Solution point | Description | Problem |
|---|:---:|:---:|
| $(\boldsymbol{x}^\star, \boldsymbol{y}^\star, \boldsymbol{\lambda}^\star)$ | cVI solution, more precisely $\boldsymbol{x}^\star = \boldsymbol{y}^\star \in \mathcal{S}_{\mathcal{C},F}^\star$ | (cVI) |
| $(\boldsymbol{x}^{\mu_t}, \boldsymbol{y}^{\mu_t}, \boldsymbol{\lambda}^{\mu_t})$ | central path point, also solution point of the subproblem with fixed $F(\boldsymbol{x}^{\mu_t})$ | ($f$-Pr) |
| $(\boldsymbol{x}_k^{\mu_t}, \boldsymbol{y}_k^{\mu_t}, \boldsymbol{\lambda}_k^{\mu_t})$ | solution point of the subproblem with fixed $F(\boldsymbol{x}_{k+1}^{(t)})$ where the indicator function is replaced with log-barrier | ($f_k$-Pr) |
| $(\boldsymbol{x}_k^\star, \boldsymbol{y}_k^\star, \boldsymbol{\lambda}_k^\star)$ | solution point of the subproblem with fixed $F(\boldsymbol{x}_{k+1}^{(t)})$ | ($f_k$-Pr-2) |

Table 1: Summary of the notation used for the solution points of the different problems. ($f_k$-Pr) is an approximation of ($f_k$-Pr-2) which replaces the indicator function with log-barrier. The $t$ emphasizes that these solution points change for different $\mu(t)$. Where clear from the context that we focus on a particular step $t$, we drop the super/sub-script $t$ to simplify the notation. See App. B.1.1.

### B.1.2 INTERMEDIATE RESULTS

We will repeatedly use the following proposition that relates the output differences of $f_k(\cdot)$ and $f(\cdot)$, defined above.

**Proposition B.1** (Relation between $f_k$ and $f$). *If $F$ is monotone, then $\forall k \in \mathbb{N}$, we have that:*

$$f_k(\boldsymbol{x}_{k+1}) - f_k(\boldsymbol{x}^\mu) \geq f(\boldsymbol{x}_{k+1}) - f(\boldsymbol{x}^\mu).$$

*Proof of Proposition B.1.* It suffices to notice that:

$$f_k(\boldsymbol{x}_{k+1}) - f_k(\boldsymbol{x}^\mu) - \big(f(\boldsymbol{x}_{k+1}) - f(\boldsymbol{x}^\mu)\big) = \langle F(\boldsymbol{x}_{k+1}) - F(\boldsymbol{x}^\mu), \boldsymbol{x}_{k+1} - \boldsymbol{x}^\mu\rangle.$$

The proof follows by applying the definition of monotonicity to the right-hand side. $\qquad \square$

We will use the following lemmas.

**Lemma B.2.** *For all $\boldsymbol{x}$ and $\boldsymbol{y}$, we have:*

$$f(\boldsymbol{x}) + g(\boldsymbol{y}) - f(\boldsymbol{x}^\mu) - g(\boldsymbol{y}^\mu) + \langle \boldsymbol{\lambda}^\mu, \boldsymbol{x} - \boldsymbol{y} \rangle \geq 0, \tag{LB.2-$f$}$$

*and:*

$$f_k(\boldsymbol{x}) + g(\boldsymbol{y}) - f_k(\boldsymbol{x}_k^\mu) - g(\boldsymbol{y}_k^\mu) + \langle \boldsymbol{\lambda}_k^\mu, \boldsymbol{x} - \boldsymbol{y} \rangle \geq 0. \tag{LB.2-$f_k$}$$

*Proof.* The Lagrange function of ($f$-Pr) is:

$$L(\boldsymbol{x}, \boldsymbol{y}, \boldsymbol{\lambda}) = f(\boldsymbol{x}) + g(\boldsymbol{y}) + \langle \boldsymbol{\lambda}, \boldsymbol{x} - \boldsymbol{y} \rangle.$$

And by the property of KKT point, we have:

$$L(\boldsymbol{x}^\mu, \boldsymbol{y}^\mu, \boldsymbol{\lambda}) \leq L(\boldsymbol{x}^\mu, \boldsymbol{y}^\mu, \boldsymbol{\lambda}^\mu) \leq L(\boldsymbol{x}, \boldsymbol{y}, \boldsymbol{\lambda}^\mu), \qquad \forall (\boldsymbol{x}, \boldsymbol{y}, \boldsymbol{\lambda}),$$

from which (LB.2-$f$) follows.

(LB.2-$f_k$) can be shown in an analogous way. $\square$

The following lemma lists some simple but useful facts that we will use in the following proofs.

**Lemma B.3.** *For the problems ($f$-Pr), ($f_k$-Pr) and the $\boldsymbol{x}_k, \boldsymbol{y}_k, \boldsymbol{\lambda}_k$ of Algorithm 3, we have:*

$$\boldsymbol{0} \in \partial f_k(\boldsymbol{x}_{k+1}) + \boldsymbol{\lambda}_k + \beta(\boldsymbol{x}_{k+1} - \boldsymbol{y}_k), \tag{LB.3-1}$$

$$\boldsymbol{0} \in \partial g(\boldsymbol{y}_{k+1}) - \boldsymbol{\lambda}_k - \beta(\boldsymbol{x}_{k+1} - \boldsymbol{y}_{k+1}), \tag{LB.3-2}$$

$$\boldsymbol{\lambda}_{k+1} - \boldsymbol{\lambda}_k = \beta(\boldsymbol{x}_{k+1} - \boldsymbol{y}_{k+1}), \tag{LB.3-3}$$

$$-\boldsymbol{\lambda}^\mu \in \partial f(\boldsymbol{x}^\mu), \tag{LB.3-4}$$

$$-\boldsymbol{\lambda}_k^\mu \in \partial f_k(\boldsymbol{x}_k^\mu), \tag{LB.3-5}$$

$$\boldsymbol{\lambda}^\mu \in \partial g(\boldsymbol{y}^\mu), \tag{LB.3-6}$$

$$\boldsymbol{\lambda}_k^\mu \in \partial g(\boldsymbol{y}_k^\mu), \tag{LB.3-7}$$

$$\boldsymbol{x}^\mu = \boldsymbol{y}^\mu, \tag{LB.3-8}$$

$$\boldsymbol{x}_k^\mu = \boldsymbol{y}_k^\mu. \tag{LB.3-9}$$

*Remark* B.4. Since $g$ is differentiable, $\partial g$ could be replaced by $\nabla g$ in Lemma B.3. Here we use $\partial g$ so that the results could be easily extended to Lemma B.31 for the proofs of Theorem 4.1, where we replace the current $g(\boldsymbol{y})$ by the indicator function $\mathbb{1}(\varphi(\boldsymbol{y}) \leq \boldsymbol{0})$, which is non-differentiable.

*Proof of Lemma B.3.* We can rewrite (AL) as:

$$\mathcal{L}_\beta(\boldsymbol{x}, \boldsymbol{y}, \boldsymbol{\lambda}) = f^k(\boldsymbol{x}) + g(\boldsymbol{y}) + \langle \boldsymbol{\lambda}, \boldsymbol{x} - \boldsymbol{y} \rangle + \frac{\beta}{2} \|\boldsymbol{x} - \boldsymbol{y}\|^2. \tag{re-AL}$$

(LB.3-1) and (LB.3-2) follow directly from (Def-X) and (Def-Y), resp. (LB.3-3) follows from line 10 in Algorithm 3, and (LB.3-4)-(LB.3-9) follows by the property of the KKT point. $\square$

We also define the following two maps (whose naming will be evident from the inclusions shown after):

$$\hat{\nabla} f_k(\boldsymbol{x}_{k+1}) \triangleq -\boldsymbol{\lambda}_k - \beta(\boldsymbol{x}_{k+1} - \boldsymbol{y}_k), \qquad \text{and} \tag{$\hat{\nabla} f_k$}$$

$$\hat{\nabla} g(\boldsymbol{y}_{k+1}) \triangleq \boldsymbol{\lambda}_k + \beta(\boldsymbol{x}_{k+1} - \boldsymbol{y}_{k+1}). \tag{$\hat{\nabla} g$}$$

Then, from (LB.3-1) and (LB.3-2) it follows that:

$$\hat{\nabla} f_k(\boldsymbol{x}_{k+1}) \in \partial f_k(\boldsymbol{x}_{k+1}) \text{ and } \hat{\nabla} g(\boldsymbol{y}_{k+1}) \in \partial g(\boldsymbol{y}_{k+1}). \tag{1}$$

We continue with two equalities for the dot products involving $\hat{\nabla} f_k$ and $\hat{\nabla} g$.

**Lemma B.5.** *For the iterates $\boldsymbol{x}_{k+1}$, $\boldsymbol{y}_{k+1}$, and $\boldsymbol{\lambda}_{k+1}$ of the ACVI—Algorithm 3—we have:*

$$\langle \hat{\nabla} g(\boldsymbol{y}_{k+1}), \boldsymbol{y}_{k+1} - \boldsymbol{y} \rangle = -\langle \boldsymbol{\lambda}_{k+1}, \boldsymbol{y} - \boldsymbol{y}_{k+1} \rangle, \tag{LB.5-1}$$

*and*

$$\langle \hat{\nabla} f_k(\boldsymbol{x}_{k+1}), \boldsymbol{x}_{k+1} - \boldsymbol{x} \rangle + \langle \hat{\nabla} g(\boldsymbol{y}_{k+1}), \boldsymbol{y}_{k+1} - \boldsymbol{y} \rangle = - \langle \boldsymbol{\lambda}_{k+1}, \boldsymbol{x}_{k+1} - \boldsymbol{y}_{k+1} - \boldsymbol{x} + \boldsymbol{y} \rangle \\ + \beta \langle -\boldsymbol{y}_{k+1} + \boldsymbol{y}_k, \boldsymbol{x}_{k+1} - \boldsymbol{x} \rangle. \tag{LB.5-2}$$

*Proof of Lemma B.5.* The first part of the lemma (LB.5-1), follows trivially by noticing that $\hat{\nabla} g(\boldsymbol{y}_{k+1}) = \boldsymbol{\lambda}_{k+1}$.

For the second part, from (LB.3-3), ($\hat{\nabla} f_k$) and ($\hat{\nabla} g$) we have:

$$\langle \hat{\nabla} f_k(\boldsymbol{x}_{k+1}), \boldsymbol{x}_{k+1} - \boldsymbol{x} \rangle = - \langle \boldsymbol{\lambda}_k + \beta(\boldsymbol{x}_{k+1} - \boldsymbol{y}_k), \boldsymbol{x}_{k+1} - \boldsymbol{x} \rangle \\ = - \langle \boldsymbol{\lambda}_{k+1}, \boldsymbol{x}_{k+1} - \boldsymbol{x} \rangle + \beta \langle -\boldsymbol{y}_{k+1} + \boldsymbol{y}_k, \boldsymbol{x}_{k+1} - \boldsymbol{x} \rangle, \tag{2}$$

and

$$\langle \hat{\nabla} g(\boldsymbol{y}_{k+1}), \boldsymbol{y}_{k+1} - \boldsymbol{y} \rangle = -\langle \boldsymbol{\lambda}_{k+1}, \boldsymbol{y} - \boldsymbol{y}_{k+1} \rangle. \tag{3}$$

Adding these together yields (LB.5-2). $\square$

The following Lemma further builds on the previous Lemma B.5, and upper-bounds some dot products involving $\hat{\nabla} f_k$ and $\hat{\nabla} g$ with a sum of only squared norms.

**Lemma B.6.** *For the $\boldsymbol{x}_{k+1}$, $\boldsymbol{y}_{k+1}$, and $\boldsymbol{\lambda}_{k+1}$ iterates of the ACVI—Algorithm 3—we have:*

$$\langle \hat{\nabla} f_k(\boldsymbol{x}_{k+1}), \boldsymbol{x}_{k+1} - \boldsymbol{x}^{\mu} \rangle + \langle \hat{\nabla} g(\boldsymbol{y}_{k+1}), \boldsymbol{y}_{k+1} - \boldsymbol{y}^{\mu} \rangle + \langle \boldsymbol{\lambda}^{\mu}, \boldsymbol{x}_{k+1} - \boldsymbol{y}_{k+1} \rangle$$

$$\leq \frac{1}{2\beta} \|\boldsymbol{\lambda}_k - \boldsymbol{\lambda}^{\mu}\|^2 - \frac{1}{2\beta} \|\boldsymbol{\lambda}_{k+1} - \boldsymbol{\lambda}^{\mu}\|^2 + \frac{\beta}{2} \|\boldsymbol{y}^{\mu} - \boldsymbol{y}_k\|^2 - \frac{\beta}{2} \|\boldsymbol{y}^{\mu} - \boldsymbol{y}_{k+1}\|^2$$

$$- \frac{1}{2\beta} \|\boldsymbol{\lambda}_{k+1} - \boldsymbol{\lambda}_k\|^2 - \frac{\beta}{2} \|\boldsymbol{y}_k - \boldsymbol{y}_{k+1}\|^2,$$

*and*

$$\langle \hat{\nabla} f_k(\boldsymbol{x}_{k+1}), \boldsymbol{x}_{k+1} - \boldsymbol{x}_k^{\mu} \rangle + \langle \hat{\nabla} g(\boldsymbol{y}_{k+1}), \boldsymbol{y}_{k+1} - \boldsymbol{y}_k^{\mu} \rangle + \langle \boldsymbol{\lambda}_k^{\mu}, \boldsymbol{x}_{k+1} - \boldsymbol{y}_{k+1} \rangle$$

$$\leq \frac{1}{2\beta} \|\boldsymbol{\lambda}_k - \boldsymbol{\lambda}_k^{\mu}\|^2 - \frac{1}{2\beta} \|\boldsymbol{\lambda}_{k+1} - \boldsymbol{\lambda}_k^{\mu}\|^2 + \frac{\beta}{2} \|\boldsymbol{y}_k^{\mu} - \boldsymbol{y}_k\|^2 - \frac{\beta}{2} \|\boldsymbol{y}_k^{\mu} - \boldsymbol{y}_{k+1}\|^2$$

$$- \frac{1}{2\beta} \|\boldsymbol{\lambda}_{k+1} - \boldsymbol{\lambda}_k\|^2 - \frac{\beta}{2} \|\boldsymbol{y}_k - \boldsymbol{y}_{k+1}\|^2.$$

*Proof of Lemma B.6.* For the left-hand side of the first part of Lemma B.6:

$$LHS = \langle \hat{\nabla} f_k(\boldsymbol{x}_{k+1}), \boldsymbol{x}_{k+1} - \boldsymbol{x}^{\mu} \rangle + \langle \hat{\nabla} g(\boldsymbol{y}_{k+1}), \boldsymbol{y}_{k+1} - \boldsymbol{y}^{\mu} \rangle + \langle \boldsymbol{\lambda}^{\mu}, \boldsymbol{x}_{k+1} - \boldsymbol{y}_{k+1} \rangle,$$

we let $(\boldsymbol{x}, \boldsymbol{y}, \boldsymbol{\lambda}) = (\boldsymbol{x}^{\mu}, \boldsymbol{y}^{\mu}, \boldsymbol{\lambda}^{\mu})$ in (LB.5-2), and using the result of that lemma, we get that:

$$LHS = -\langle \boldsymbol{\lambda}_{k+1}, \boldsymbol{x}_{k+1} - \boldsymbol{y}_{k+1} - \boldsymbol{x}^{\mu} + \boldsymbol{y}^{\mu} \rangle + \beta \langle -\boldsymbol{y}_{k+1} + \boldsymbol{y}_k, \boldsymbol{x}_{k+1} - \boldsymbol{x}^{\mu} \rangle + \langle \boldsymbol{\lambda}^{\mu}, \boldsymbol{x}_{k+1} - \boldsymbol{y}_{k+1} \rangle,$$

and since $\boldsymbol{x}^{\mu} = \boldsymbol{y}^{\mu}$ (LB.3-8):

$$LHS = -\langle \boldsymbol{\lambda}_{k+1}, \boldsymbol{x}_{k+1} - \boldsymbol{y}_{k+1} \rangle + \beta \langle -\boldsymbol{y}_{k+1} + \boldsymbol{y}_k, \boldsymbol{x}_{k+1} - \boldsymbol{x}^{\mu} \rangle + \langle \boldsymbol{\lambda}^{\mu}, \boldsymbol{x}_{k+1} - \boldsymbol{y}_{k+1} \rangle$$

$$= -\langle \boldsymbol{\lambda}_{k+1} - \boldsymbol{\lambda}^{\mu}, \boldsymbol{x}_{k+1} - \boldsymbol{y}_{k+1} \rangle + \beta \langle -\boldsymbol{y}_{k+1} + \boldsymbol{y}_k, \boldsymbol{x}_{k+1} - \boldsymbol{x}^{\mu} \rangle,$$

where in the last equality, we combined the first and last terms together. Using (LB.3-3) that $\frac{1}{\beta}(\boldsymbol{\lambda}_{k+1} - \boldsymbol{\lambda}_k) = (\boldsymbol{x}_{k+1} - \boldsymbol{y}_{k+1})$ yields (for the second term above, we add and subtract $\boldsymbol{y}_{k+1}$ in its second argument, and use $\boldsymbol{x}^{\mu} = \boldsymbol{y}^{\mu}$):

$$LHS = -\frac{1}{\beta} \langle \boldsymbol{\lambda}_{k+1} - \boldsymbol{\lambda}^{\mu}, \boldsymbol{\lambda}_{k+1} - \boldsymbol{\lambda}_k \rangle + \langle -\boldsymbol{y}_{k+1} + \boldsymbol{y}_k, \boldsymbol{\lambda}_{k+1} - \boldsymbol{\lambda}_k \rangle$$

$$- \beta \langle -\boldsymbol{y}_{k+1} + \boldsymbol{y}_k, -\boldsymbol{y}_{k+1} + \boldsymbol{y}^{\mu} \rangle \tag{4}$$

Using the 3-point identity—that for any vectors $\boldsymbol{a}, \boldsymbol{b}, \boldsymbol{c}$ it holds $\langle \boldsymbol{b} - \boldsymbol{a}, \boldsymbol{b} - \boldsymbol{c} \rangle = \frac{1}{2}(\|\boldsymbol{a} - \boldsymbol{b}\|^2 + \|\boldsymbol{b} - \boldsymbol{c}\|^2 - \|\boldsymbol{a} - \boldsymbol{c}\|^2)$—for the first term above we get that:

$$\langle \boldsymbol{\lambda}_{k+1} - \boldsymbol{\lambda}^\mu, \boldsymbol{\lambda}_{k+1} - \boldsymbol{\lambda}_k \rangle = \frac{1}{2}\left( \|\boldsymbol{\lambda}_k - \boldsymbol{\lambda}^\mu\|^2 + \|\boldsymbol{\lambda}_{k+1} - \boldsymbol{\lambda}_k\|^2 - \|\boldsymbol{\lambda}_{k+1} - \boldsymbol{\lambda}^\mu\|^2 \right),$$

and similarly,

$$\langle -\boldsymbol{y}_{k+1} + \boldsymbol{y}_k, -\boldsymbol{y}_{k+1} + \boldsymbol{y}^\mu \rangle = \frac{1}{2}\left( \|-\boldsymbol{y}_k + \boldsymbol{y}^\mu\|^2 - \|-\boldsymbol{y}_{k+1} + \boldsymbol{y}^\mu\|^2 - \|-\boldsymbol{y}_{k+1} + \boldsymbol{y}_k\|^2 \right),$$

and by adding these together, we get:

$$\begin{aligned} LHS = {} & \frac{1}{2\beta}\|\boldsymbol{\lambda}_k - \boldsymbol{\lambda}^\mu\|^2 - \frac{1}{2\beta}\|\boldsymbol{\lambda}_{k+1} - \boldsymbol{\lambda}^\mu\|^2 - \frac{1}{2\beta}\|\boldsymbol{\lambda}_{k+1} - \boldsymbol{\lambda}_k\|^2 \\ & + \frac{\beta}{2}\|-\boldsymbol{y}_k + \boldsymbol{y}^\mu\|^2 - \frac{\beta}{2}\|-\boldsymbol{y}_{k+1} + \boldsymbol{y}^\mu\|^2 - \frac{\beta}{2}\|-\boldsymbol{y}_{k+1} + \boldsymbol{y}_k\|^2 \\ & + \langle -\boldsymbol{y}_{k+1} + \boldsymbol{y}_k, \boldsymbol{\lambda}_{k+1} - \boldsymbol{\lambda}_k \rangle. \end{aligned} \tag{5}$$

On the other hand, (LB.5-1) which states that $\langle \hat{\nabla} g(\boldsymbol{y}_{k+1}), \boldsymbol{y}_{k+1} - \boldsymbol{y} \rangle + \langle \boldsymbol{\lambda}_{k+1}, -\boldsymbol{y}_{k+1} + \boldsymbol{y} \rangle = 0$, also asserts:

$$\langle \hat{\nabla} g(\boldsymbol{y}_k), \boldsymbol{y}_k - \boldsymbol{y} \rangle + \langle \boldsymbol{\lambda}_k, -\boldsymbol{y}_k + \boldsymbol{y} \rangle = 0. \tag{6}$$

Letting $\boldsymbol{y} = \boldsymbol{y}_k$ in (LB.5-1), and $\boldsymbol{y} = \boldsymbol{y}_{k+1}$ in (6), and adding them together yields:

$$\langle \hat{\nabla} g(\boldsymbol{y}_{k+1}) - \hat{\nabla} g(\boldsymbol{y}_k), \boldsymbol{y}_{k+1} - \boldsymbol{y}_k \rangle + \langle \boldsymbol{\lambda}_{k+1} - \boldsymbol{\lambda}_k, -\boldsymbol{y}_{k+1} + \boldsymbol{y}_k \rangle = 0.$$

By the monotonicity of $\partial g$, we know that the first term of the above equality is non-negative. Thus, we have:

$$\langle \boldsymbol{\lambda}_{k+1} - \boldsymbol{\lambda}_k, -\boldsymbol{y}_{k+1} + \boldsymbol{y}_k \rangle \leq 0. \tag{7}$$

Lastly, plugging it into (5) gives the first inequality of Lemma B.6.

The second inequality of Lemma B.6 follows similarly. $\qquad\square$

The following Lemma upper-bounds the sum of (i) the difference of $f(\cdot)$ evaluated at $\boldsymbol{x}_{k+1}$ and at $\boldsymbol{x}^\mu$ and (ii) the difference of $g(\cdot)$ evaluated at $\boldsymbol{y}_{k+1}$ and at $\boldsymbol{y}^\mu$; up to a term that depends on $\boldsymbol{x}_{k+1} - \boldsymbol{y}_{k+1}$ as well. Recall that $(\boldsymbol{x}^\mu, \boldsymbol{y}^\mu, \boldsymbol{\lambda}^\mu)$ is a point on the central path.

**Lemma B.7.** *For the $\boldsymbol{x}_{k+1}$, $\boldsymbol{y}_{k+1}$, and $\boldsymbol{\lambda}_{k+1}$ iterates of the ACVI—Algorithm 3—we have:*

$$\begin{aligned} f(\boldsymbol{x}_{k+1}) + g(\boldsymbol{y}_{k+1}) - f(\boldsymbol{x}^\mu) - g(\boldsymbol{y}^\mu) + \langle \boldsymbol{\lambda}^\mu, \boldsymbol{x}_{k+1} - \boldsymbol{y}_{k+1} \rangle & \\ \leq {} & \frac{1}{2\beta}\|\boldsymbol{\lambda}_k - \boldsymbol{\lambda}^\mu\|^2 - \frac{1}{2\beta}\|\boldsymbol{\lambda}_{k+1} - \boldsymbol{\lambda}^\mu\|^2 \\ & + \frac{\beta}{2}\|-\boldsymbol{y}_k + \boldsymbol{y}^\mu\|^2 - \frac{\beta}{2}\|-\boldsymbol{y}_{k+1} + \boldsymbol{y}^\mu\|^2 \\ & - \frac{1}{2\beta}\|\boldsymbol{\lambda}_{k+1} - \boldsymbol{\lambda}_k\|^2 - \frac{\beta}{2}\|-\boldsymbol{y}_{k+1} + \boldsymbol{y}_k\|^2 \end{aligned} \tag{LB.7}$$

*Proof of Lemma B.7.* From the convexity of $f_k(\boldsymbol{x})$ and $g(\boldsymbol{y})$; from proposition B.1 on the relation between $f_k(\cdot)$ and $f(\cdot)$ which asserts that $f(\boldsymbol{x}_{k+1}) - f(\boldsymbol{x}^\mu) \leq f_k(\boldsymbol{x}_{k+1}) - f_k(\boldsymbol{x}^\mu)$; as well as from Eq. (1) which asserts that $\hat{\nabla} f_k(\boldsymbol{x}_{k+1}) \in \partial f_k(\boldsymbol{x}_{k+1})$ and $\hat{\nabla} g(\boldsymbol{y}_{k+1}) \in \partial g(\boldsymbol{y}_{k+1})$; it follows for the LHS of Lemma B.7 that:

$$\begin{aligned} f(\boldsymbol{x}_{k+1}) + g(\boldsymbol{y}_{k+1}) - f(\boldsymbol{x}^\mu) - g(\boldsymbol{y}^\mu) + \langle \boldsymbol{\lambda}^\mu, \boldsymbol{x}_{k+1} - \boldsymbol{y}_{k+1} \rangle & \\ \leq f_k(\boldsymbol{x}_{k+1}) + g(\boldsymbol{y}_{k+1}) - f_k(\boldsymbol{x}^\mu) - g(\boldsymbol{y}^\mu) + \langle \boldsymbol{\lambda}^\mu, \boldsymbol{x}_{k+1} - \boldsymbol{y}_{k+1} \rangle & \tag{8} \\ \leq \langle \hat{\nabla} f_k(\boldsymbol{x}_{k+1}), \boldsymbol{x}_{k+1} - \boldsymbol{x}^\mu \rangle + \langle \hat{\nabla} g(\boldsymbol{y}_{k+1}), \boldsymbol{y}_{k+1} - \boldsymbol{y}^\mu \rangle + \langle \boldsymbol{\lambda}^\mu, \boldsymbol{x}_{k+1} - \boldsymbol{y}_{k+1} \rangle & \end{aligned}$$

Finally, by plugging in the first part of Lemma B.6, Lemma B.7 follows, that is:

$$
\begin{aligned}
f(\boldsymbol{x}_{k+1}) + g(\boldsymbol{y}_{k+1}) &- f(\boldsymbol{x}^{\mu}) - g(\boldsymbol{y}^{\mu}) + \langle \boldsymbol{\lambda}^{\mu}, \boldsymbol{x}_{k+1} - \boldsymbol{y}_{k+1} \rangle \\
&\leq \frac{1}{2\beta} \|\boldsymbol{\lambda}_k - \boldsymbol{\lambda}^{\mu}\|^2 - \frac{1}{2\beta} \|\boldsymbol{\lambda}_{k+1} - \boldsymbol{\lambda}^{\mu}\|^2 \\
&\quad + \frac{\beta}{2} \|-\boldsymbol{y}_k + \boldsymbol{y}^{\mu}\|^2 - \frac{\beta}{2} \|-\boldsymbol{y}_{k+1} + \boldsymbol{y}^{\mu}\|^2 \\
&\quad - \frac{1}{2\beta} \|\boldsymbol{\lambda}_{k+1} - \boldsymbol{\lambda}_k\|^2 - \frac{\beta}{2} \|-\boldsymbol{y}_{k+1} + \boldsymbol{y}_k\|^2.
\end{aligned}
\tag{9}
$$

$\square$

The following theorem upper bounds the analogous quantity but for $f_k(\cdot)$ (instead of $f$ as does Lemma B.7), and further asserts that the difference between the $\boldsymbol{x}_{k+1}$ and $\boldsymbol{y}_{k+1}$ iterates of exact ACVI (Algorithm 3) tends to 0 asymptotically. The inequality in Theorem B.8 plays an important role later when deriving the nonasymptotic convergence rate of ACVI.

**Theorem B.8** (Asymptotic convergence of $(\boldsymbol{x}_{k+1} - \boldsymbol{y}_{k+1})$ of ACVI). *For the $\boldsymbol{x}_{k+1}$, $\boldsymbol{y}_{k+1}$, and $\boldsymbol{\lambda}_{k+1}$ iterates of the ACVI—Algorithm 3—we have:*

$$
\begin{aligned}
f_k(\boldsymbol{x}_{k+1}) - f_k(\boldsymbol{x}_k^{\mu}) &+ g(\boldsymbol{y}_{k+1}) - g(\boldsymbol{y}_k^{\mu}) \\
&\leq \|\boldsymbol{\lambda}_{k+1}\| \, \|\boldsymbol{x}_{k+1} - \boldsymbol{y}_{k+1}\| + \beta \|\boldsymbol{y}_{k+1} - \boldsymbol{y}_k\| \, \|\boldsymbol{x}_{k+1} - \boldsymbol{x}_k^{\mu}\| \to 0 \, ,
\end{aligned}
\tag{TB.8-$f_k$-UB}
$$

*and*

$$
\boldsymbol{x}_{k+1} - \boldsymbol{y}_{k+1} \to \boldsymbol{0} \, , \qquad \text{as } k \to \infty \, .
$$

*Proof of Theorem B.8: Asymptotic convergence of $(\boldsymbol{x}_{k+1} - \boldsymbol{y}_{k+1})$ of ACVI.* Recall from (LB.2-$f$) of Lemma B.2 that by setting $\boldsymbol{x} \equiv \boldsymbol{x}_{k+1}, \boldsymbol{y} \equiv \boldsymbol{y}_{k+1}$ we asserted that:

$$
f(\boldsymbol{x}_{k+1}) - f(\boldsymbol{x}^{\mu}) + g(\boldsymbol{y}_{k+1}) - g(\boldsymbol{y}^{\mu}) + \langle \boldsymbol{\lambda}^{\mu}, \boldsymbol{x}_{k+1} - \boldsymbol{y}_{k+1} \rangle \geq 0 \, .
$$

Further, notice that the LHS of the above inequality overlaps with that of (LB.7). This implies that the RHS of (LB.7) has to be non-negative. Hence, we have that:

$$
\begin{aligned}
\frac{1}{2\beta} \|\boldsymbol{\lambda}_{k+1} - \boldsymbol{\lambda}_k\|^2 + \frac{\beta}{2} \|-\boldsymbol{y}_{k+1} + \boldsymbol{y}_k\|^2 &\leq \frac{1}{2\beta} \|\boldsymbol{\lambda}_k - \boldsymbol{\lambda}^{\mu}\|^2 - \frac{1}{2\beta} \|\boldsymbol{\lambda}_{k+1} - \boldsymbol{\lambda}^{\mu}\|^2 \\
&\quad + \frac{\beta}{2} \|-\boldsymbol{y}_k + \boldsymbol{y}^{\mu}\|^2 - \frac{\beta}{2} \|-\boldsymbol{y}_{k+1} + \boldsymbol{y}^{\mu}\|^2.
\end{aligned}
\tag{10}
$$

Summing over $k = 0, \ldots, \infty$ gives:

$$
\sum_{k=0}^{\infty} \left( \frac{1}{2\beta} \|\boldsymbol{\lambda}_{k+1} - \boldsymbol{\lambda}_k\|^2 + \frac{\beta}{2} \|-\boldsymbol{y}_{k+1} + \boldsymbol{y}_k\|^2 \right) \leq \frac{1}{2\beta} \|\boldsymbol{\lambda}_0 - \boldsymbol{\lambda}^{\mu}\|^2 + \frac{\beta}{2} \|-\boldsymbol{y}_0 + \boldsymbol{y}^{\mu}\|^2 \, ,
$$

from which we deduce that $\boldsymbol{\lambda}_{k+1} - \boldsymbol{\lambda}_k \to \boldsymbol{0}$ and $\boldsymbol{y}_{k+1} - \boldsymbol{y}_k \to \boldsymbol{0}$.

Also notice that by simply reorganizing (10) we have:

$$
\begin{aligned}
\frac{1}{2\beta} \|\boldsymbol{\lambda}_{k+1} - \boldsymbol{\lambda}^{\mu}\|^2 &+ \frac{\beta}{2} \|-\boldsymbol{y}_{k+1} + \boldsymbol{y}^{\mu}\|^2 \\
&\leq \frac{1}{2\beta} \|\boldsymbol{\lambda}_k - \boldsymbol{\lambda}^{\mu}\|^2 + \frac{\beta}{2} \|-\boldsymbol{y}_k + \boldsymbol{y}^{\mu}\|^2 - \frac{1}{2\beta} \|\boldsymbol{\lambda}_{k+1} - \boldsymbol{\lambda}_k\|^2 - \frac{\beta}{2} \|-\boldsymbol{y}_{k+1} + \boldsymbol{y}_k\|^2 \\
&\leq \frac{1}{2\beta} \|\boldsymbol{\lambda}_k - \boldsymbol{\lambda}^{\mu}\|^2 + \frac{\beta}{2} \|-\boldsymbol{y}_k + \boldsymbol{y}^{\mu}\|^2 \\
&\leq \frac{1}{2\beta} \|\boldsymbol{\lambda}_0 - \boldsymbol{\lambda}^{\mu}\|^2 + \frac{\beta}{2} \|-\boldsymbol{y}_0 + \boldsymbol{y}^{\mu}\|^2 \, ,
\end{aligned}
\tag{11}
$$

where the second inequality follows because the norm is non-negative.

From (11) we can see that $\|\boldsymbol{\lambda}_k - \boldsymbol{\lambda}^\mu\|^2$ and $\|\boldsymbol{y}_k - \boldsymbol{y}^\mu\|^2$ are bounded for all $k$, as well as $\|\boldsymbol{\lambda}_k\|$. Recall that:

$$\boldsymbol{\lambda}_{k+1} - \boldsymbol{\lambda}_k = \beta(\boldsymbol{x}_{k+1} - \boldsymbol{y}_{k+1}) = \beta(\boldsymbol{x}_{k+1} - \boldsymbol{x}^\mu) + \beta(-\boldsymbol{y}_{k+1} + \boldsymbol{y}^\mu),$$

where in the last equality, we add and subtract $\boldsymbol{x}^\mu = \boldsymbol{y}^\mu$. Combining this with the fact that $\boldsymbol{\lambda}_{k+1} - \boldsymbol{\lambda}_k \to \mathbf{0}$ (see above), we deduce that $\boldsymbol{x}_{k+1} - \boldsymbol{y}_{k+1} \to \mathbf{0}$ and that $\boldsymbol{x}_{k+1} - \boldsymbol{x}^\mu$ is also bounded.

Using the convexity of $f_k(\cdot)$ and $g(\cdot)$ for the LHS of Theorem B.8 we have:

$$\begin{aligned} \text{LHS} &= f_k(\boldsymbol{x}_{k+1}) - f_k(\boldsymbol{x}_k^\mu) + g(\boldsymbol{y}_{k+1}) - g(\boldsymbol{y}_k^\mu) \\ &\leq \langle \hat{\nabla} f_k(\boldsymbol{x}_{k+1}), \boldsymbol{x}_{k+1} - \boldsymbol{x}_k^\mu \rangle + \langle \hat{\nabla} g(\boldsymbol{y}_{k+1}), \boldsymbol{y}_{k+1} - \boldsymbol{y}_k^\mu \rangle. \end{aligned}$$

Using (LB.5-2) with $\boldsymbol{x} \equiv \boldsymbol{x}_k^\mu, \boldsymbol{y} \equiv \boldsymbol{y}_k^\mu$ we have:

$$\text{LHS} \leq -\langle \boldsymbol{\lambda}_{k+1}, \boldsymbol{x}_{k+1} - \boldsymbol{y}_{k+1} \underbrace{-\boldsymbol{x}_k^\mu + \boldsymbol{y}_k^\mu}_{=\mathbf{0}, \text{ due to (LB.3-9)}} \rangle + \beta\langle -\boldsymbol{y}_{k+1} + \boldsymbol{y}_k, \boldsymbol{x}_{k+1} - \boldsymbol{x}_k^\mu \rangle.$$

Hence, it follows that:

$$\begin{aligned} f_k(\boldsymbol{x}_{k+1}) &- f_k(\boldsymbol{x}_k^\mu) + g(\boldsymbol{y}_{k+1}) - g(\boldsymbol{y}_k^\mu) \\ &\leq -\langle \boldsymbol{\lambda}_{k+1}, \boldsymbol{x}_{k+1} - \boldsymbol{y}_{k+1} \rangle + \beta\langle -\boldsymbol{y}_{k+1} + \boldsymbol{y}_k, \boldsymbol{x}_{k+1} - \boldsymbol{x}_k^\mu \rangle \\ &\leq \|\boldsymbol{\lambda}_{k+1}\| \, \|\boldsymbol{x}_{k+1} - \boldsymbol{y}_{k+1}\| + \beta \, \|\boldsymbol{y}_{k+1} - \boldsymbol{y}_k\| \, \|\boldsymbol{x}_{k+1} - \boldsymbol{x}_k^\mu\|, \end{aligned}$$

where the last inequality follows from Cauchy-Schwarz.

Recall that $\mathcal{C}$ is compact and $D$ is the diameter of $\mathcal{C}$:

$$D \triangleq \sup_{\boldsymbol{x}, \boldsymbol{y} \in \mathcal{C}} \|\boldsymbol{x} - \boldsymbol{y}\|.$$

Thus, we have:

$$\|\boldsymbol{y}_{k+1} - \boldsymbol{y}_k^\mu\| = \|\boldsymbol{y}_{k+1} - \boldsymbol{y}^\mu\| + \|\boldsymbol{y}^\mu - \boldsymbol{y}_{k+1}^\mu\| \leq \|\boldsymbol{y}_{k+1} - \boldsymbol{y}^\mu\| + D,$$

which implies that $\|\boldsymbol{y}_k - \boldsymbol{y}_k^\mu\|$ are bounded for all $k$. Since:

$$\boldsymbol{\lambda}_{k+1} - \boldsymbol{\lambda}_k = \beta(\boldsymbol{x}_{k+1} - \boldsymbol{y}_{k+1}) = \beta(\boldsymbol{x}_{k+1} - \boldsymbol{x}_k^\mu) + \beta(-\boldsymbol{y}_{k+1} + \boldsymbol{y}_k^\mu),$$

we deduce that $\boldsymbol{x}_{k+1} - \boldsymbol{x}_k^\mu$ is also bounded. Thus, we have (TB.8-$f_k$-UB). $\qquad \square$

The following lemma states an important intermediate result that ensures that $\frac{1}{2\beta}\|\boldsymbol{\lambda}_{k+1} - \boldsymbol{\lambda}_k\|^2 + \frac{\beta}{2}\|-\boldsymbol{y}_{k+1} + \boldsymbol{y}_k\|^2$ does not increase.

**Lemma B.9** (non-increment of $\frac{1}{2\beta}\|\boldsymbol{\lambda}_{k+1} - \boldsymbol{\lambda}_k\|^2 + \frac{\beta}{2}\|-\boldsymbol{y}_{k+1} + \boldsymbol{y}_k\|^2$). *For the $\boldsymbol{x}_{k+1}, \boldsymbol{y}_{k+1},$ and $\boldsymbol{\lambda}_{k+1}$ iterates of the ACVI—Algorithm 3—we have:*

$$\frac{1}{2\beta}\|\boldsymbol{\lambda}_{k+1} - \boldsymbol{\lambda}_k\|^2 + \frac{\beta}{2}\|-\boldsymbol{y}_{k+1} + \boldsymbol{y}_k\|^2 \leq \frac{1}{2\beta}\|\boldsymbol{\lambda}_k - \boldsymbol{\lambda}_{k-1}\|^2 + \frac{\beta}{2}\|-\boldsymbol{y}_k + \boldsymbol{y}_{k-1}\|^2. \quad \text{(LB.9)}$$

*Proof of Lemma B.9.* (LB.5-2) gives:

$$\begin{aligned} \langle \hat{\nabla} f_{k-1}(\boldsymbol{x}_k), \boldsymbol{x}_k - \boldsymbol{x} \rangle &+ \langle \hat{\nabla} g(\boldsymbol{y}_k), \boldsymbol{y}_k - \boldsymbol{y} \rangle \\ &= -\langle \boldsymbol{\lambda}_k, \boldsymbol{x}_k - \boldsymbol{y}_k - \boldsymbol{x} + \boldsymbol{y} \rangle + \beta\langle -\boldsymbol{y}_k + \boldsymbol{y}_{k-1}, \boldsymbol{x}_k - \boldsymbol{x} \rangle. \end{aligned} \quad (12)$$

Letting $(\boldsymbol{x}, \boldsymbol{y}, \boldsymbol{\lambda}) = (\boldsymbol{x}_k, \boldsymbol{y}_k, \boldsymbol{\lambda}_k)$ in (LB.5-2) and $(\boldsymbol{x}, \boldsymbol{y}, \boldsymbol{\lambda}) = (\boldsymbol{x}_{k+1}, \boldsymbol{y}_{k+1}, \boldsymbol{\lambda}_{k+1})$ in (12), and adding them together, and using (LB.3-3) yields:

$$
\begin{aligned}
&\langle \hat{\nabla} f_k \left(\boldsymbol{x}_{k+1}\right) - \hat{\nabla} f_{k-1} \left(\boldsymbol{x}_k\right), \boldsymbol{x}_{k+1} - \boldsymbol{x}_k \rangle + \langle \hat{\nabla} g \left(\boldsymbol{y}_{k+1}\right) - \hat{\nabla} g \left(\boldsymbol{y}_k\right), \boldsymbol{y}_{k+1} - \boldsymbol{y}_k \rangle \\
&= - \langle \boldsymbol{\lambda}_{k+1} - \boldsymbol{\lambda}_k, \boldsymbol{x}_{k+1} - \boldsymbol{y}_{k+1} - \boldsymbol{x}_k + \boldsymbol{y}_k \rangle + \beta \langle -\boldsymbol{y}_{k+1} + \boldsymbol{y}_k - \left(-\boldsymbol{y}_k + \boldsymbol{y}_{k-1}\right), \boldsymbol{x}_{k+1} - \boldsymbol{x}_k \rangle \\
&= - \frac{1}{\beta} \langle \boldsymbol{\lambda}_{k+1} - \boldsymbol{\lambda}_k, \boldsymbol{\lambda}_{k+1} - \boldsymbol{\lambda}_k - \left(\boldsymbol{\lambda}_k - \boldsymbol{\lambda}_{k-1}\right) \rangle \\
&\quad + \langle -\boldsymbol{y}_{k+1} + \boldsymbol{y}_k + \left(\boldsymbol{y}_k - \boldsymbol{y}_{k-1}\right), \boldsymbol{\lambda}_{k+1} - \boldsymbol{\lambda}_k + \beta \boldsymbol{y}_{k+1} - \left(\boldsymbol{\lambda}_k - \boldsymbol{\lambda}_{k-1} + \beta \boldsymbol{y}_k\right) \rangle \\
&= \frac{1}{2\beta} \left[ \|\boldsymbol{\lambda}_k - \boldsymbol{\lambda}_{k-1}\|^2 - \|\boldsymbol{\lambda}_{k+1} - \boldsymbol{\lambda}_k\|^2 - \|\boldsymbol{\lambda}_{k+1} - \boldsymbol{\lambda}_k - \left(\boldsymbol{\lambda}_k - \boldsymbol{\lambda}_{k-1}\right)\|^2 \right] \\
&\quad + \frac{\beta}{2} \left[ \|-\boldsymbol{y}_k + \boldsymbol{y}_{k-1}\|^2 - \|-\boldsymbol{y}_{k+1} + \boldsymbol{y}_k\|^2 - \|-\boldsymbol{y}_{k+1} + \boldsymbol{y}_k - \left(-\boldsymbol{y}_k + \boldsymbol{y}_{k-1}\right)\|^2 \right] \\
&\quad + \langle -\boldsymbol{y}_{k+1} + \boldsymbol{y}_k - \left(-\boldsymbol{y}_k + \boldsymbol{y}_{k-1}\right), \boldsymbol{\lambda}_{k+1} - \boldsymbol{\lambda}_k - \left(\boldsymbol{\lambda}_k - \boldsymbol{\lambda}_{k-1}\right) \rangle \\
&= \frac{1}{2\beta} \left( \|\boldsymbol{\lambda}_k - \boldsymbol{\lambda}_{k-1}\|^2 - \|\boldsymbol{\lambda}_{k+1} - \boldsymbol{\lambda}_k\|^2 \right) + \frac{\beta}{2} \left( \|-\boldsymbol{y}_k + \boldsymbol{y}_{k-1}\|^2 - \|-\boldsymbol{y}_{k+1} + \boldsymbol{y}_k\|^2 \right) \\
&\quad - \frac{1}{2\beta} \|\boldsymbol{\lambda}_{k+1} - \boldsymbol{\lambda}_k - \left(\boldsymbol{\lambda}_k - \boldsymbol{\lambda}_{k-1}\right)\|^2 - \frac{\beta}{2} \|-\boldsymbol{y}_{k+1} + \boldsymbol{y}_k - \left(-\boldsymbol{y}_k + \boldsymbol{y}_{k-1}\right)\|^2 \\
&\quad + \langle -\boldsymbol{y}_{k+1} + \boldsymbol{y}_k - \left(-\boldsymbol{y}_k + \boldsymbol{y}_{k-1}\right), \boldsymbol{\lambda}_{k+1} - \boldsymbol{\lambda}_k - \left(\boldsymbol{\lambda}_k - \boldsymbol{\lambda}_{k-1}\right) \rangle \\
&\leq \frac{1}{2\beta} \left( \|\boldsymbol{\lambda}_k - \boldsymbol{\lambda}_{k-1}\|^2 - \|\boldsymbol{\lambda}_{k+1} - \boldsymbol{\lambda}_k\|^2 \right) + \frac{\beta}{2} \left( \|-\boldsymbol{y}_k + \boldsymbol{y}_{k-1}\|^2 - \|-\boldsymbol{y}_{k+1} + \boldsymbol{y}_k\|^2 \right) .
\end{aligned}
$$

By the convexity of $f_k$ and $f_{k-1}$, we get:

$$
\begin{aligned}
\langle \hat{\nabla} f_k \left(\boldsymbol{x}_{k+1}\right), \boldsymbol{x}_{k+1} - \boldsymbol{x}_k \rangle &\geq f_k(\boldsymbol{x}_{k+1}) - f_k(\boldsymbol{x}_k), \\
-\langle \hat{\nabla} f_{k-1} \left(\boldsymbol{x}_k\right), \boldsymbol{x}_{k+1} - \boldsymbol{x}_k \rangle &\geq f_{k-1}(\boldsymbol{x}_k) - f_{k-1}(\boldsymbol{x}_{k+1}) .
\end{aligned}
$$

Adding them together gives that:

$$
\begin{aligned}
\langle \hat{\nabla} f_k \left(\boldsymbol{x}_{k+1}\right) - \hat{\nabla} f_{k-1} \left(\boldsymbol{x}_k\right), \boldsymbol{x}_{k+1} - \boldsymbol{x}_k \rangle &\geq f_k(\boldsymbol{x}_{k+1}) - f_{k-1}(\boldsymbol{x}_{k+1}) - f_k(\boldsymbol{x}_k) + f_{k-1}(\boldsymbol{x}_k) \\
&= \langle F(\boldsymbol{x}_{k+1}) - F(\boldsymbol{x}_k), \boldsymbol{x}_{k+1} - \boldsymbol{x}_k \rangle \geq 0 .
\end{aligned}
$$

Thus by the monotonicity of $F$ and $\hat{\nabla} g$, (LB.9) follows. $\qquad\square$

**Lemma B.10.** *If $F$ is monotone on $\mathcal{C}_=$, then for Algorithm 3, we have:*

$$
\begin{aligned}
f_K \left(\boldsymbol{x}_{K+1}\right) + g \left(\boldsymbol{y}_{K+1}\right) - f_K \left(\boldsymbol{x}_K^\mu\right) - g \left(\boldsymbol{y}_K^\mu\right) & \\
&\leq \frac{\Delta^\mu}{K+1} + \left(2\sqrt{\Delta^\mu} + \frac{1}{\sqrt{\beta}} \|\boldsymbol{\lambda}^\mu\| + \sqrt{\beta} D\right) \sqrt{\frac{\Delta^\mu}{K+1}}, \quad \text{(LB.10-1)}
\end{aligned}
$$

$$
\text{and} \qquad \|\boldsymbol{x}_{K+1} - \boldsymbol{y}_{K+1}\| \leq \sqrt{\frac{\Delta^\mu}{\beta (K+1)}}, \quad \text{(LB.10-2)}
$$

*where $\Delta^\mu \triangleq \frac{1}{\beta} \|\boldsymbol{\lambda}_0 - \boldsymbol{\lambda}^\mu\|^2 + \beta \|\boldsymbol{y}_0 - \boldsymbol{y}^\mu\|^2$.*

*Proof of Lemma B.10.* Summing (10) over $k = 0, 1, \ldots, K$ and using the monotonicity of $\frac{1}{2\beta} \|\boldsymbol{\lambda}_{k+1} - \boldsymbol{\lambda}_k\|^2 + \frac{\beta}{2} \|-\boldsymbol{y}_{k+1} + \boldsymbol{y}_k\|^2$ from Lemma B.9, we have:

$$
\begin{aligned}
\frac{1}{\beta} \|\boldsymbol{\lambda}_{K+1} - \boldsymbol{\lambda}_K\|^2 &+ \beta \|-\boldsymbol{y}_{K+1} + \boldsymbol{y}_K\|^2 \\
&\leq \frac{1}{K+1} \sum_{k=0}^{K} \left( \frac{1}{2\beta} \|\boldsymbol{\lambda}_{k+1} - \boldsymbol{\lambda}_k\|^2 + \frac{\beta}{2} \|-\boldsymbol{y}_{k+1} + \boldsymbol{y}_k\|^2 \right) \\
&\leq \frac{1}{K+1} \left( \frac{1}{\beta} \|\boldsymbol{\lambda}_0 - \boldsymbol{\lambda}^\mu\|^2 + \beta \|-\boldsymbol{y}_0 + \boldsymbol{y}^\mu\|^2 \right) . \quad \text{(13)}
\end{aligned}
$$

From this, we deduce that:

$$\beta \|\boldsymbol{x}_{K+1} - \boldsymbol{y}_{K+1}\| = \|\boldsymbol{\lambda}_{K+1} - \boldsymbol{\lambda}_K\| \le \sqrt{\frac{\beta \Delta^\mu}{K+1}},$$

$$\|-\boldsymbol{y}_{K+1} + \boldsymbol{y}_K\| \le \sqrt{\frac{\Delta^\mu}{\beta(K+1)}}.$$

On the other hand, (11) gives:

$$\frac{1}{2\beta} \|\boldsymbol{\lambda}_{K+1} - \boldsymbol{\lambda}^\mu\|^2 + \frac{\beta}{2} \|-\boldsymbol{y}_{K+1} + \boldsymbol{y}^\mu\|^2 \le \frac{1}{2} \Delta^\mu.$$

Hence, we have:

$$\|\boldsymbol{\lambda}_{K+1} - \boldsymbol{\lambda}^\mu\| \le \sqrt{\beta \Delta^\mu}, \tag{14}$$

$$\|-\boldsymbol{y}_{K+1} + \boldsymbol{y}^\mu\| \le \sqrt{\frac{\Delta^\mu}{\beta}}.$$

Furthermore, we have:

$$\|\boldsymbol{y}_{K+1} - \boldsymbol{y}_K^\mu\| \le \|\boldsymbol{y}^{K+1} - \boldsymbol{y}^\mu\| + \|\boldsymbol{y}^\mu - \boldsymbol{y}_\mu^K\| \le \sqrt{\frac{\Delta^\mu}{\beta}} + D,$$

$$\begin{aligned}
\|\boldsymbol{x}_{K+1} - \boldsymbol{x}_K^\mu\| &= \left\| \frac{1}{\beta}(\boldsymbol{\lambda}_{K+1} - \boldsymbol{\lambda}_K) - (-\boldsymbol{y}_{K+1} + \boldsymbol{y}_K^\mu) \right\| \\
&\le \frac{1}{\beta} \|\boldsymbol{\lambda}_{K+1} - \boldsymbol{\lambda}_K\| + \|\boldsymbol{y}_{K+1} - \boldsymbol{y}_K^\mu\| \\
&\le \sqrt{\frac{\Delta^\mu}{\beta(K+1)}} + \sqrt{\frac{\Delta^\mu}{\beta}} + D,
\end{aligned}$$

and

$$\|\boldsymbol{\lambda}_{K+1}\| \le \|\boldsymbol{\lambda}^\mu\| + \|\boldsymbol{\lambda}_{K+1} - \boldsymbol{\lambda}^\mu\| \le \|\boldsymbol{\lambda}^\mu\| + \sqrt{\beta \Delta^\mu}.$$

Then using (TB.8-$f_k$-UB) in Lemma B.8, we have (LB.37-1). □

**Discussion.** Lemma B.10 has an analogous form to Theorem 7 in (Yang et al., 2023), but here we change the reference points from $\boldsymbol{x}^\mu$ and $\boldsymbol{y}^\mu$ in (28) of (Yang et al., 2023) to $\boldsymbol{x}_K^\mu$ and $\boldsymbol{y}_K^\mu$ we newly introduce in our paper. We stress that this change, together with our observation that $\boldsymbol{x}_k^\star$ is the reference point of the gap function at $\boldsymbol{x}_{k+1}$ (see (16)), is crucial to weakening the assumptions in (Yang et al., 2023). We can see this from the following proof sketch of Theorem 3.1:

1. Lemma B.10 gives

$$f_K(\boldsymbol{x}_{K+1}) + g(\boldsymbol{y}_{K+1}) - f_K(\boldsymbol{x}_K^\mu) - g(\boldsymbol{y}_K^\mu) = \mathcal{O}(\frac{1}{\sqrt{K}}).$$

2. Note that $g(\boldsymbol{y}_{K+1}) - g(\boldsymbol{y}_K^\mu) \to 0$, $\boldsymbol{x}_K^\mu \to \boldsymbol{x}_K^\star$ when $\mu \to 0$. Using the above inequality, we have that when $\mu$ is small,

$$|f_K(\boldsymbol{x}_{K+1}) - f_K(\boldsymbol{x}_K^\star)| = \mathcal{O}(\frac{1}{\sqrt{K}}).$$

3. With observation (16), we could write the gap function at $\boldsymbol{x}_{K+1}$ explicitly as

$$\mathcal{G}(\boldsymbol{x}_{K+1}, \mathcal{C}) = f_K(\boldsymbol{x}_{K+1}) - f_K(\boldsymbol{x}_K^\star).$$

Combining the above two expressions, we reach our conclusion:

$$\mathcal{G}(\boldsymbol{x}_{K+1}, \mathcal{C}) = \mathcal{O}(\frac{1}{\sqrt{K}}).$$

In contrast, (Yang et al., 2023) gives the upper bound w.r.t. the gap function through two indirect steps, each introducing some extra assumptions:

1. Theorem 7 in (Yang et al., 2023) gives

$$f_K\left(\boldsymbol{x}_{K+1}\right) + g\left(\boldsymbol{y}_{K+1}\right) - f_K\left(\boldsymbol{x}^\mu\right) - g\left(\boldsymbol{y}^\mu\right) = \mathcal{O}(\frac{1}{\sqrt{K}}),$$

   which leads to $|F(\boldsymbol{x}_K)^\mathsf{T}(\boldsymbol{x}_{K+1} - \boldsymbol{x}^\star)| = \mathcal{O}(\frac{1}{\sqrt{K}})$ and $|F(\boldsymbol{x}^\star)^\mathsf{T}(\boldsymbol{x}_{K+1} - \boldsymbol{x}^\star)| = \mathcal{O}(\frac{1}{\sqrt{K}})$ (the first indirect bound) when $\mu$ is small enough.

2. Under either the $\xi$-monotonicity assumption in Thm. 2 or assumption (iii) in Thm. 3 of (Yang et al., 2023), they are able to bound the iterate distance using the above results as follows:

$$\|\boldsymbol{x}_K - \boldsymbol{x}^\star\| = \mathcal{O}(\frac{1}{\sqrt{K}})$$

   (the second indirect bound).

3. Finally, by assuming Lipschitzness of $F$, they derive from the above bound that

$$\mathcal{G}(\boldsymbol{x}_{K+1}, \mathcal{C}) = \mathcal{O}(\frac{1}{\sqrt{K}}).$$

### B.1.3  PROVING THEOREM 3.1

We are now ready to prove Theorem 3.1. Here, we give a nonasymptotic convergence rate of Algorithm 3.

**Theorem B.11** (Restatement of Theorem 3.1). *Given an continuous operator $F\colon \mathcal{X} \to \mathbb{R}^n$, assume that:*

*(i) $F$ is monotone on $\mathcal{C}_=$, as per Def. 2.1;*

*(ii) $F$ is either strictly monotone on $\mathcal{C}$ or one of $\varphi_i$ is strictly convex.*

*Let $(\boldsymbol{x}_K^{(t)}, \boldsymbol{y}_K^{(t)}, \boldsymbol{\lambda}_K^{(t)})$ denote the last iterate of Algorithm 3. Given any fixed $K \in \mathbb{N}_+$, run with sufficiently small $\mu_{-1}$, then for all $t \in [T]$, we have:*

$$\mathcal{G}(\boldsymbol{x}_K, \mathcal{C}) \leq \frac{2\Delta}{K} + 2\left(2\sqrt{\Delta} + \frac{1}{\sqrt{\beta}}\|\boldsymbol{\lambda}^\star\| + \sqrt{\beta}D + 1 + M\right)\sqrt{\frac{\Delta}{K}} \qquad \text{(na-Rate)}$$

*and*

$$\|\boldsymbol{x}^K - \boldsymbol{y}^K\| \leq 2\sqrt{\frac{\Delta}{\beta K}}, \tag{15}$$

*where $\Delta \triangleq \frac{1}{\beta}\|\boldsymbol{\lambda}_0 - \boldsymbol{\lambda}^\star\|^2 + \beta\|\boldsymbol{y}_0 - \boldsymbol{y}^\star\|^2$ and $D \triangleq \sup_{\boldsymbol{x},\boldsymbol{y}\in\mathcal{C}} \|\boldsymbol{x} - \boldsymbol{y}\|$, and $M \triangleq \sup_{\boldsymbol{x}\in\mathcal{C}} \|F(\boldsymbol{x})\|$.*

*Proof of Theorem B.11.* Note that

$$\begin{aligned}
(f_k\text{-Pr-2}) &\Leftrightarrow \min_{\boldsymbol{x}\in\mathcal{C}}\langle F(\boldsymbol{x}_{k+1}), \boldsymbol{x}\rangle \\
&\Leftrightarrow \max_{\boldsymbol{x}\in\mathcal{C}}\langle F(\boldsymbol{x}_{k+1}), \boldsymbol{x}_{k+1} - \boldsymbol{x}\rangle \\
&\Leftrightarrow \mathcal{G}(\boldsymbol{x}_{k+1}, \mathcal{C}),
\end{aligned}$$

from which we deduce

$$\mathcal{G}(\boldsymbol{x}_{k+1}, \mathcal{C}) = \langle F(\boldsymbol{x}_{k+1}), \boldsymbol{x}_{k+1} - \boldsymbol{x}_k^\star\rangle, \forall k. \tag{16}$$

For any fixed $K \in \mathbb{N}$, by Corollary 2.11 in (Chu, 1998) we know that

$$\begin{aligned}
&\boldsymbol{x}_K^\mu \to \boldsymbol{x}_K^\star, \\
&g(\boldsymbol{y}_{K+1}) - g(\boldsymbol{y}_K^\mu) \to 0, \\
&\Delta^\mu \to \frac{1}{\beta}\|\boldsymbol{\lambda}_0 - \boldsymbol{\lambda}^\star\|^2 + \beta\|\boldsymbol{y}_0 - \boldsymbol{y}^\star\|^2 = \Delta. \tag{17}
\end{aligned}$$

Thus, there exists $\mu_{-1} > 0$, $s.t. \forall 0 < \mu < \mu_{-1}$,

$$\|\boldsymbol{x}_K^\mu - \boldsymbol{x}_K^\star\| \leq \sqrt{\frac{\Delta^\mu}{K+1}},$$

$$|g(\boldsymbol{y}_{K+1}) - g(\boldsymbol{y}_K^\mu)| \leq \sqrt{\frac{\Delta^\mu}{K+1}}.$$

Combining with Lemma B.10, we have

$$
\begin{aligned}
\langle F(\boldsymbol{x}_{K+1}), &\boldsymbol{x}_{K+1} - \boldsymbol{x}_K^\mu \rangle \\
&= f_K(\boldsymbol{x}_{K+1}) - f_K(\boldsymbol{x}_K^\mu) \\
&\leq \frac{\Delta^\mu}{K+1} + \left(2\sqrt{\Delta^\mu} + \frac{1}{\sqrt{\beta}}\|\boldsymbol{\lambda}^\mu\| + \sqrt{\beta}D\right)\sqrt{\frac{\Delta^\mu}{K+1}} + g(\boldsymbol{y}_K^\mu) - g(\boldsymbol{y}_{K+1}) \quad (18) \\
&\leq \frac{\Delta^\mu}{K+1} + \left(2\sqrt{\Delta^\mu} + \frac{1}{\sqrt{\beta}}\|\boldsymbol{\lambda}^\mu\| + \sqrt{\beta}D + 1\right)\sqrt{\frac{\Delta^\mu}{K+1}}.
\end{aligned}
$$

Using the above inequality, we have

$$
\begin{aligned}
\mathcal{G}(\boldsymbol{x}_{K+1}, \mathcal{C}) &= \langle F(\boldsymbol{x}_{K+1}), \boldsymbol{x}_{K+1} - \boldsymbol{x}_K^\star \rangle & (19) \\
&= \langle F(\boldsymbol{x}_{K+1}), \boldsymbol{x}_{K+1} - \boldsymbol{x}_K^\mu \rangle + \langle F(\boldsymbol{x}_{K+1}), \boldsymbol{x}_K^\mu - \boldsymbol{x}_K^\star \rangle & (20) \\
&\leq \langle F(\boldsymbol{x}_{K+1}), \boldsymbol{x}_{K+1} - \boldsymbol{x}_K^\mu \rangle + \|F(\boldsymbol{x}_{K+1})\| \|\boldsymbol{x}_K^\mu - \boldsymbol{x}_K^\star\| & (21) \\
&\leq \frac{\Delta^\mu}{K+1} + \left(2\sqrt{\Delta^\mu} + \frac{1}{\sqrt{\beta}}\|\boldsymbol{\lambda}^\mu\| + \sqrt{\beta}D + 1 + M\right)\sqrt{\frac{\Delta^\mu}{K+1}}. & (22)
\end{aligned}
$$

Moreover, by (17), we can choose small enough $\mu_{-1}$ so that

$$\mathcal{G}(\boldsymbol{x}_{K+1}, \mathcal{C}) \leq \frac{2\Delta}{K+1} + 2\left(2\sqrt{\Delta} + \frac{1}{\sqrt{\beta}}\|\boldsymbol{\lambda}^\star\| + \sqrt{\beta}D + 1 + M\right)\sqrt{\frac{\Delta}{K+1}},$$

and

$$\|\boldsymbol{x}_{K+1} - \boldsymbol{y}_{K+1}\| \leq 2\sqrt{\frac{\Delta}{\beta(K+1)}}, \quad (23)$$

where (23) uses (LB.10-2) in Lemma B.10. $\qquad\qquad\square$

## B.2 PROOF OF THEOREM 3.2: LAST-ITERATE CONVERGENCE OF INEXACT ACVI FOR MONOTONE VARIATIONAL INEQUALITIES

### B.2.1 USEFUL LEMMAS FROM PREVIOUS WORKS

The following lemma is Lemma 1 from (Schmidt et al., 2011).

**Lemma B.12** (Lemma 1 in (Schmidt et al., 2011))**.** *Assume that the nonnegative sequence $\{u_k\}$ satisfies the following recursion for all $k \geq 1$ :*

$$u_k^2 \leq S_k + \sum_{i=1}^{k} \lambda_i u_i \,,$$

*with $\{S_k\}$ an increasing sequence, $S_0 \geq u_0^2$ and $\lambda_i \geq 0$ for all $i$. Then, for all $k \geq 1$, it follows:*

$$u_k \leq \frac{1}{2} \sum_{i=1}^{k} \lambda_i + \left( S_k + \left( \frac{1}{2} \sum_{i=1}^{k} \lambda_i \right)^2 \right)^{1/2} \,.$$

*Proof.* We prove the result by induction. It is true for $k = 0$ (by assumption). We assume it is true for $k - 1$, and we denote by $v_{k-1} = \max\{u_1, \ldots, u_{k-1}\}$. From the recursion, we deduce:

$$(u_k - \lambda_k/2)^2 \leq S_k + \frac{\lambda_k^2}{4} + v_{k-1} \sum_{i=1}^{k-1} \lambda_i \,, \tag{24}$$

leading to

$$u_k \leq \frac{\lambda_k}{2} + \left( S_k + \frac{\lambda_k^2}{4} + v_{k-1}k - 1 \sum_{i=1}^{k-1} \lambda_i \right)^{1/2} \,, \tag{25}$$

and thus

$$u_k \leq \max \left\{ v_{k-1}, \frac{\lambda_k}{2} + \left( S_k + \frac{\lambda_k^2}{4} + v_{k-1}k - 1 \sum_{i=1}^{k-1} \lambda_i \right)^{1/2} \right\} \,. \tag{26}$$

Let $v_{k-1}^{\star} \triangleq \frac{1}{2} \sum_{i=1}^{k} \lambda_i + \left( S_k + \left( \frac{1}{2} \sum_{i=1}^{k} \lambda_i \right)^2 \right)^{1/2}$. Note that

$$v_{k-1} = \frac{\lambda_k}{2} + \left( S_k + \frac{\lambda_k^2}{4} + v_{k-1}k - 1 \sum_{i=1}^{k-1} \lambda_i \right)^{1/2}$$

$$\Leftrightarrow v_{k-1} = v_{k-1}^{\star} \,.$$

Since the two terms in the max are increasing functions of $v_{k-1}$, it follows that if $v_{k-1} \leq v_{k-1}^{\star}$, then $v_k \leq v_{k-1}^{\star}$. Also note that

$$v_{k-1} \geq \frac{\lambda_k}{2} + \left( S_k + \frac{\lambda_k^2}{4} + v_{k-1}k - 1 \sum_{i=1}^{k-1} \lambda_i \right)^{1/2}$$

$$\Leftrightarrow v_{k-1} \geq v_{k-1}^{\star} \,.$$

From which we deduce that if $v_{k-1} \geq v_{k-1}^{\star}$, then $v_k \leq v_{k-1}$, and the induction hypotheses ensure that the property is satisfied for $k$. □

In the convergence rate analysis of inexact ACVI-Algorithm 1, we need the following definition (Bertsekas et al., 2003):

**Definition B.13** ($\varepsilon$-subdifferential)**.** Given a convex function $\psi : \mathbb{R}^n \to \mathbb{R}$ and a positive scalar $\varepsilon$, a vector $\boldsymbol{a} \in \mathbb{R}^n$ is called an $\varepsilon$-subgradient of $\psi$ at a point $\boldsymbol{x} \in dom\psi$ if

$$\psi(\boldsymbol{z}) \geq \psi(\boldsymbol{x}) + (\boldsymbol{z} - \boldsymbol{x})^\mathsf{T} \boldsymbol{a} - \varepsilon, \ \forall \boldsymbol{z} \in \mathbb{R}^n \,. \tag{$\varepsilon$-G}$$

The set of all $\varepsilon$-subgradients of a convex function $\psi$ at $\boldsymbol{x} \in dom\psi$ is called the $\varepsilon$-subdifferential of $\psi$ at $\boldsymbol{x}$, and is denoted by $\partial_\varepsilon \psi(\boldsymbol{x})$.

### B.2.2 INTERMEDIATE RESULTS

We first give some lemmas that will be used in the proof of Theorem 3.2.

In the following proofs, we assume $\varepsilon_0 = \sigma_0 = 0$. We need the following two lemmas to state a lemma analogous to Lemma B.3 but for the inexact ACVI.

**Lemma B.14.** *In inexact ACVI-Algorithm 1, for each $k$, $\exists\ \boldsymbol{r}_{k+1} \in \mathbb{R}^n$, $\|\boldsymbol{r}_{k+1}\| \leq \sqrt{\frac{2\varepsilon_{k+1}}{\beta}}$, s.t.*

$$\beta(\boldsymbol{x}_{k+1} + \frac{1}{\beta}\boldsymbol{\lambda}_k - \boldsymbol{y}_{k+1} - \boldsymbol{r}_{k+1}) \in \partial_{\varepsilon_{k+1}} g(\boldsymbol{y}_{k+1})\,.$$

*Proof of Lemma B.14.* We first recall some properties of $\varepsilon$-subdifferentials (see, eg. (Bertsekas et al., 2003), Section 4.3 for more details). $\boldsymbol{x}$ is an $\varepsilon$-minimizer (see Def. 2.4) of a convex function $\psi$ if and only if $\boldsymbol{0} \in \partial_{\varepsilon}\psi(\boldsymbol{x})$. Let $\psi = \psi_1 + \psi_2$, where both $\psi_1$ and $\psi_2$ are convex, we have $\partial_{\epsilon}\psi(\boldsymbol{x}) \subset \partial_{\epsilon}\psi_1(\boldsymbol{x}) + \partial_{\epsilon}\psi_2(\boldsymbol{x})$. If $\psi_1(\boldsymbol{x}) = \frac{\beta}{2}\|\boldsymbol{x} - \boldsymbol{z}\|^2$, then

$$\partial_{\varepsilon}\psi_1(\boldsymbol{x}) = \left\{\boldsymbol{y} \in \mathbb{R}^n \middle| \frac{\beta}{2}\left\|\boldsymbol{x} - \boldsymbol{z} - \frac{\boldsymbol{y}}{\beta}\right\|^2 \leq \varepsilon\right\}$$

$$= \left\{\boldsymbol{y} \in \mathbb{R}^n, \boldsymbol{y} = \beta\boldsymbol{x} - \beta\boldsymbol{z} + \beta\boldsymbol{r} \middle| \frac{\beta}{2}\|\boldsymbol{r}\|^2 \leq \varepsilon\right\}\,.$$

Let $\psi_2 = g$ and $\boldsymbol{z} = \boldsymbol{x}_{k+1} + \frac{1}{\beta}\boldsymbol{\lambda}_k$, then $\boldsymbol{y}_{k+1}$ is an $\varepsilon_{k+1}$-minimizer of $\psi_1 + \psi_2$. Thus we have $\boldsymbol{0} \in \partial_{\varepsilon_{k+1}}\psi(\boldsymbol{y}_{k+1}) \subset \partial_{\varepsilon_{k+1}}\psi_1(\boldsymbol{y}_{k+1}) + \partial_{\varepsilon_{k+1}}\psi_2(\boldsymbol{y}_{k+1})$. Hence, there is an $\boldsymbol{r}_{k+1}$ such that

$$\beta(\boldsymbol{x}_{k+1} + \frac{1}{\beta}\boldsymbol{\lambda}_k - \boldsymbol{y}_{k+1} - \boldsymbol{r}_{k+1}) \in \partial_{\varepsilon_{k+1}} g(\boldsymbol{y}_{k+1}) \quad \text{with} \quad \|\boldsymbol{r}_{k+1}\| \leq \sqrt{\frac{2\varepsilon_{k+1}}{\beta}}\,.$$

$\square$

**Lemma B.15.** *In inexact ACVI-Algorithm 1, for each $k$, $\exists\ \boldsymbol{q}_{k+1} \in \mathbb{R}^n$, $\|\boldsymbol{q}_{k+1}\| \leq \sigma_{k+1}$, s.t.*

$$\boldsymbol{x}_{k+1} + \boldsymbol{q}_{k+1} = \underset{\boldsymbol{x}}{arg\,min}\left\{f_k(\boldsymbol{x}) + \frac{\beta}{2}\left\|\boldsymbol{x} - \boldsymbol{y}_k + \frac{1}{\beta}\boldsymbol{\lambda}_k\right\|^2\right\}\,. \tag{27}$$

*where $\mathcal{L}_\beta$ is the augmented Lagrangian of problem ($f_k$-Pr).*

*Proof of Lemma B.15.* By the definition of $\boldsymbol{x}_{k+1}$ (see line 8 of inexact ACVI-Algorithm 1 and Def. 2.3) we have

$$\boldsymbol{x}_{k+1} + \boldsymbol{q}_{k+1} = -\frac{1}{\beta}\boldsymbol{P}_c F(\boldsymbol{x}) + \boldsymbol{P}_c \boldsymbol{y}_k - \frac{1}{\beta}\boldsymbol{P}_c \boldsymbol{\lambda}_k + \boldsymbol{d}_c$$

$$= \underset{\boldsymbol{x}}{arg\,min}\ \mathcal{L}_\beta(\boldsymbol{x}, \boldsymbol{y}_k, \boldsymbol{\lambda}_k)\,,$$

where $\mathcal{L}_\beta$ is the augmented Lagrangian of the problem, which is given in AL (note that $\boldsymbol{w}_k = \boldsymbol{x}_{k+1}$). ($f_k$-Pr). And from the above equation (27) follows. $\square$

Similar to Lemma B.3, and using Lemma B.14 and Lemma B.15, we give the following lemma for inexact ACVI–Algorithm 1.

**Lemma B.16.** *For the problems ($f$-Pr), ($f_k$-Pr) and inexact ACVI-Algorithm 1, we have*

$$\boldsymbol{0} \in \partial f_k(\boldsymbol{x}_{k+1} + \boldsymbol{q}_{k+1}) + \boldsymbol{\lambda}_k + \beta(\boldsymbol{x}_{k+1} - \boldsymbol{y}_k) + \beta\boldsymbol{q}_{k+1} \tag{LB.16-1}$$

$$\boldsymbol{0} \in \partial_{\varepsilon_{k+1}} g(\boldsymbol{y}_{k+1}) - \boldsymbol{\lambda}_k - \beta(\boldsymbol{x}_{k+1} - \boldsymbol{y}_{k+1}) + \beta\boldsymbol{r}_{k+1}, \tag{LB.16-2}$$

$$\boldsymbol{\lambda}_{k+1} - \boldsymbol{\lambda}_k = \beta(\boldsymbol{x}_{k+1} - \boldsymbol{y}_{k+1}), \tag{LB.16-3}$$

$$-\boldsymbol{\lambda}^\mu \in \partial f(\boldsymbol{x}^\mu), \tag{LB.16-4}$$

$$-\boldsymbol{\lambda}_k^\mu \in \partial f_k(\boldsymbol{x}_k^\mu), \tag{LB.16-5}$$

$$\boldsymbol{\lambda}^\mu = \nabla g(\boldsymbol{y}^\mu), \tag{LB.16-6}$$

$$\boldsymbol{\lambda}_k^\mu = \nabla g(\boldsymbol{y}_k^\mu), \tag{LB.16-7}$$

$$\boldsymbol{x}^\mu = \boldsymbol{y}^\mu, \tag{LB.16-8}$$

$$\boldsymbol{x}_k^\mu = \boldsymbol{y}_k^\mu, \tag{LB.16-9}$$

We define the following two maps (whose naming will be evident from the inclusions shown after):

$$\hat{\nabla} f_k(\boldsymbol{x}_{k+1} + \boldsymbol{q}_{k+1}) \triangleq -\boldsymbol{\lambda}_k - \beta(\boldsymbol{x}_{k+1} - \boldsymbol{y}_k) - \beta \boldsymbol{q}_{k+1}, \qquad \text{and} \qquad \text{(noisy-}\hat{\nabla} f_k\text{)}$$

$$\hat{\nabla}_{\varepsilon_{k+1}} g(\boldsymbol{y}_{k+1}) \triangleq \boldsymbol{\lambda}_k + \beta(\boldsymbol{x}_{k+1} - \boldsymbol{y}_{k+1}) - \beta \boldsymbol{r}_{k+1}. \qquad \text{(noisy-}\hat{\nabla} g\text{)}$$

Then, from (LB.3-1) and (LB.3-2) it follows that:

$$\hat{\nabla} f_k(\boldsymbol{x}_{k+1} + \boldsymbol{q}_{k+1}) \in \partial f_k(\boldsymbol{x}_{k+1} + \boldsymbol{q}_{k+1}) \text{ and } \hat{\nabla}_{\varepsilon_{k+1}} g(\boldsymbol{y}_{k+1}) \in \partial_{\varepsilon_{k+1}} g(\boldsymbol{y}_{k+1}). \qquad (28)$$

The following lemma is analogous to Lemma B.5 but refers to the noisy case.

**Lemma B.17.** *For the iterates $\boldsymbol{x}_{k+1}$, $\boldsymbol{y}_{k+1}$, and $\boldsymbol{\lambda}_{k+1}$ of the inexact ACVI—Algorithm 1—we have:*

$$\langle \hat{\nabla}_{\varepsilon_{k+1}} g(\boldsymbol{y}_{k+1}), \boldsymbol{y}_{k+1} - \boldsymbol{y} \rangle = -\langle \boldsymbol{\lambda}_{k+1}, \boldsymbol{y} - \boldsymbol{y}_{k+1} \rangle - \beta \langle \boldsymbol{r}_{k+1}, \boldsymbol{y}_{k+1} - \boldsymbol{y} \rangle, \qquad \text{(LB.17-1)}$$

*and*

$$\langle \hat{\nabla} f_k(\boldsymbol{x}_{k+1} + \boldsymbol{q}_{k+1}), \boldsymbol{x}_{k+1} - \boldsymbol{x} \rangle + \langle \hat{\nabla}_{\varepsilon_{k+1}} g(\boldsymbol{y}_{k+1}), \boldsymbol{y}_{k+1} - \boldsymbol{y} \rangle$$
$$= -\langle \boldsymbol{\lambda}_{k+1}, \boldsymbol{x}_{k+1} - \boldsymbol{y}_{k+1} - \boldsymbol{x} + \boldsymbol{y} \rangle + \beta \langle -\boldsymbol{y}_{k+1} + \boldsymbol{y}_k, \boldsymbol{x}_{k+1} - \boldsymbol{x} \rangle \qquad \text{(LB.17-2)}$$
$$- \beta \langle \boldsymbol{q}_{k+1}, \boldsymbol{x}_{k+1} - \boldsymbol{x} \rangle - \beta \langle \boldsymbol{r}_{k+1}, \boldsymbol{y}_{k+1} - \boldsymbol{y} \rangle.$$

*Proof of Lemma B.17.* From (LB.16-3), (noisy-$\hat{\nabla} f_k$) and (noisy-$\hat{\nabla} g$) we have:

$$\langle \hat{\nabla} f_k(\boldsymbol{x}_{k+1} + \boldsymbol{q}_{k+1}) + \beta \boldsymbol{q}_{k+1}, \boldsymbol{x}_{k+1} - \boldsymbol{x} \rangle = -\langle \boldsymbol{\lambda}_k + \beta(\boldsymbol{x}_{k+1} - \boldsymbol{y}_k), \boldsymbol{x}_{k+1} - \boldsymbol{x} \rangle$$
$$= -\langle \boldsymbol{\lambda}_{k+1}, \boldsymbol{x}_{k+1} - \boldsymbol{x} \rangle + \beta \langle -\boldsymbol{y}_{k+1} + \boldsymbol{y}_k, \boldsymbol{x}_{k+1} - \boldsymbol{x} \rangle,$$

and

$$\langle \hat{\nabla}_{\varepsilon_{k+1}} g(\boldsymbol{y}_{k+1}) + \beta \boldsymbol{r}_{k+1}, \boldsymbol{y}_{k+1} - \boldsymbol{y} \rangle = -\langle \boldsymbol{\lambda}_{k+1}, \boldsymbol{y} - \boldsymbol{y}_{k+1} \rangle.$$

Adding these together yields:

$$\langle \hat{\nabla} f_k(\boldsymbol{x}_{k+1} + \boldsymbol{q}_{k+1}) + \beta \boldsymbol{q}_{k+1}, \boldsymbol{x}_{k+1} - \boldsymbol{x} \rangle + \langle \hat{\nabla}_{\varepsilon_{k+1}} g(\boldsymbol{y}_{k+1}) + \beta \boldsymbol{r}_{k+1}, \boldsymbol{y}_{k+1} - \boldsymbol{y} \rangle$$
$$= -\langle \boldsymbol{\lambda}_{k+1}, \boldsymbol{x}_{k+1} - \boldsymbol{y}_{k+1} - \boldsymbol{x} + \boldsymbol{y} \rangle$$
$$+ \beta \langle -\boldsymbol{y}_{k+1} + \boldsymbol{y}_k, \boldsymbol{x}_{k+1} - \boldsymbol{x} \rangle.$$

Rearranging the above two equations, we obtain (LB.17-1) and (LB.17-2). $\square$

The following lemma is analogous to Lemma B.6 but refers to the noisy case.

**Lemma B.18.** *For the $\boldsymbol{x}_{k+1}$, $\boldsymbol{y}_{k+1}$, and $\boldsymbol{\lambda}_{k+1}$ iterates of the inexact ACVI—Algorithm 1—we have:*

$$\langle \hat{\nabla} f_k(\boldsymbol{x}_{k+1} + \boldsymbol{q}_{k+1}), \boldsymbol{x}_{k+1} - \boldsymbol{x}^\mu \rangle + \langle \hat{\nabla}_{\varepsilon_{k+1}} g(\boldsymbol{y}_{k+1}), \boldsymbol{y}_{k+1} - \boldsymbol{y}^\mu \rangle + \langle \boldsymbol{\lambda}^\mu, \boldsymbol{x}_{k+1} - \boldsymbol{y}_{k+1} \rangle$$

$$\leq \frac{1}{2\beta} \|\boldsymbol{\lambda}_k - \boldsymbol{\lambda}^\mu\|^2 - \frac{1}{2\beta} \|\boldsymbol{\lambda}_{k+1} - \boldsymbol{\lambda}^\mu\|^2 + \frac{\beta}{2} \|\boldsymbol{y}^\mu - \boldsymbol{y}_k\|^2 - \frac{\beta}{2} \|\boldsymbol{y}^\mu - \boldsymbol{y}_{k+1}\|^2$$

$$- \frac{1}{2\beta} \|\boldsymbol{\lambda}_{k+1} - \boldsymbol{\lambda}_k\|^2 - \frac{\beta}{2} \|\boldsymbol{y}_k - \boldsymbol{y}_{k+1}\|^2$$

$$- \beta \langle \boldsymbol{r}_{k+1} - \boldsymbol{r}_k, \boldsymbol{y}_{k+1} - \boldsymbol{y}_k \rangle - \beta \langle \boldsymbol{r}_{k+1}, \boldsymbol{y}_{k+1} - \boldsymbol{y}^\mu \rangle + \varepsilon_k + \varepsilon_{k+1} - \beta \langle \boldsymbol{q}_{k+1}, \boldsymbol{x}_{k+1} - \boldsymbol{x}^\mu \rangle,$$

*and*

$$\langle \hat{\nabla} f_k(\boldsymbol{x}_{k+1} + \boldsymbol{q}_{k+1}), \boldsymbol{x}_{k+1} - \boldsymbol{x}_k^\mu \rangle + \langle \hat{\nabla}_{\varepsilon_{k+1}} g(\boldsymbol{y}_{k+1}), \boldsymbol{y}_{k+1} - \boldsymbol{y}_k^\mu \rangle + \langle \boldsymbol{\lambda}_k^\mu, \boldsymbol{x}_{k+1} - \boldsymbol{y}_{k+1} \rangle$$

$$\leq \frac{1}{2\beta} \|\boldsymbol{\lambda}_k - \boldsymbol{\lambda}_k^\mu\|^2 - \frac{1}{2\beta} \|\boldsymbol{\lambda}_{k+1} - \boldsymbol{\lambda}_k^\mu\|^2 + \frac{\beta}{2} \|\boldsymbol{y}_k^\mu - \boldsymbol{y}_k\|^2 - \frac{\beta}{2} \|\boldsymbol{y}_k^\mu - \boldsymbol{y}_{k+1}\|^2$$

$$- \frac{1}{2\beta} \|\boldsymbol{\lambda}_{k+1} - \boldsymbol{\lambda}_k\|^2 - \frac{\beta}{2} \|\boldsymbol{y}_k - \boldsymbol{y}_{k+1}\|^2$$

$$- \beta \langle \boldsymbol{r}_{k+1} - \boldsymbol{r}_k, \boldsymbol{y}_{k+1} - \boldsymbol{y}_k \rangle - \beta \langle \boldsymbol{r}_{k+1}, \boldsymbol{y}_{k+1} - \boldsymbol{y}_k^\mu \rangle + \varepsilon_k + \varepsilon_{k+1} - \beta \langle \boldsymbol{q}_{k+1}, \boldsymbol{x}_{k+1} - \boldsymbol{x}_k^\mu \rangle.$$

*Proof of Lemma B.18.* For the left-hand side of the first part of Lemma B.18:

$$LHS = \langle \hat{\nabla} f_k(\boldsymbol{x}_{k+1} + \boldsymbol{q}_{k+1}), \boldsymbol{x}_{k+1} - \boldsymbol{x}_k^{\mu} \rangle + \langle \hat{\nabla}_{\varepsilon_{k+1}} g(\boldsymbol{y}_{k+1}), \boldsymbol{y}_{k+1} - \boldsymbol{y}_k^{\mu} \rangle + \langle \boldsymbol{\lambda}_k^{\mu}, \boldsymbol{x}_{k+1} - \boldsymbol{y}_{k+1} \rangle,$$

we let $(\boldsymbol{x}, \boldsymbol{y}, \boldsymbol{\lambda}) = (\boldsymbol{x}^{\mu}, \boldsymbol{y}^{\mu}, \boldsymbol{\lambda}^{\mu})$ in (LB.17-2), and using the result of that lemma we get that:

$$LHS = - \langle \boldsymbol{\lambda}_{k+1}, \boldsymbol{x}_{k+1} - \boldsymbol{y}_{k+1} - \boldsymbol{x}^{\mu} + \boldsymbol{y}^{\mu} \rangle + \beta \langle -\boldsymbol{y}_{k+1} + \boldsymbol{y}_k, \boldsymbol{x}_{k+1} - \boldsymbol{x}^{\mu} \rangle$$
$$- \beta \langle \boldsymbol{q}_{k+1}, \boldsymbol{x}_{k+1} - \boldsymbol{x}^{\mu} \rangle - \beta \langle \boldsymbol{r}_{k+1}, \boldsymbol{y}_{k+1} - \boldsymbol{y}^{\mu} \rangle + \langle \boldsymbol{\lambda}^{\mu}, \boldsymbol{x}_{k+1} - \boldsymbol{y}_{k+1} \rangle,$$

and since $\boldsymbol{x}^{\mu} = \boldsymbol{y}^{\mu}$ (LB.3-8):

$$LHS = - \langle \boldsymbol{\lambda}_{k+1}, \boldsymbol{x}_{k+1} - \boldsymbol{y}_{k+1} \rangle + \beta \langle -\boldsymbol{y}_{k+1} + \boldsymbol{y}_k, \boldsymbol{x}_{k+1} - \boldsymbol{x}^{\mu} \rangle + \langle \boldsymbol{\lambda}^{\mu}, \boldsymbol{x}_{k+1} - \boldsymbol{y}_{k+1} \rangle$$
$$- \beta \langle \boldsymbol{q}_{k+1}, \boldsymbol{x}_{k+1} - \boldsymbol{x}^{\mu} \rangle - \beta \langle \boldsymbol{r}_{k+1}, \boldsymbol{y}_{k+1} - \boldsymbol{y}^{\mu} \rangle$$
$$= - \langle \boldsymbol{\lambda}_{k+1} - \boldsymbol{\lambda}^{\mu}, \boldsymbol{x}_{k+1} - \boldsymbol{y}_{k+1} \rangle + \beta \langle -\boldsymbol{y}_{k+1} + \boldsymbol{y}_k, \boldsymbol{x}_{k+1} - \boldsymbol{x}^{\mu} \rangle$$
$$- \beta \langle \boldsymbol{q}_{k+1}, \boldsymbol{x}_{k+1} - \boldsymbol{x}^{\mu} \rangle - \beta \langle \boldsymbol{r}_{k+1}, \boldsymbol{y}_{k+1} - \boldsymbol{y}^{\mu} \rangle,$$

where in the last equality, we combined the first and third terms together. Using (LB.16-3) that $\frac{1}{\beta}(\boldsymbol{\lambda}_{k+1} - \boldsymbol{\lambda}_k) = \boldsymbol{x}_{k+1} - \boldsymbol{y}_{k+1}$ yields (for the second term above, we add and subtract $\boldsymbol{y}_{k+1}$ in its second argument, and use $\boldsymbol{x}^{\mu} = \boldsymbol{y}^{\mu}$):

$$LHS = - \frac{1}{\beta} \langle \boldsymbol{\lambda}_{k+1} - \boldsymbol{\lambda}^{\mu}, \boldsymbol{\lambda}_{k+1} - \boldsymbol{\lambda}_k \rangle + \langle -\boldsymbol{y}_{k+1} + \boldsymbol{y}_k, \boldsymbol{\lambda}_{k+1} - \boldsymbol{\lambda}_k \rangle$$
$$- \beta \langle -\boldsymbol{y}_{k+1} + \boldsymbol{y}_k, -\boldsymbol{y}_{k+1} + \boldsymbol{y}^{\mu} \rangle - \beta \langle \boldsymbol{q}_{k+1}, \boldsymbol{x}_{k+1} - \boldsymbol{x}^{\mu} \rangle - \beta \langle \boldsymbol{r}_{k+1}, \boldsymbol{y}_{k+1} - \boldsymbol{y}^{\mu} \rangle. \tag{29}$$

Using the 3-point identity, that for any vectors $\boldsymbol{a}, \boldsymbol{b}, \boldsymbol{c}$ it holds $\langle \boldsymbol{b} - \boldsymbol{a}, \boldsymbol{b} - \boldsymbol{c} \rangle = \frac{1}{2}(\|\boldsymbol{a} - \boldsymbol{b}\|^2 + \|\boldsymbol{b} - \boldsymbol{c}\|^2 - \|\boldsymbol{a} - \boldsymbol{c}\|^2)$, for the first term above, we get that:

$$\langle \boldsymbol{\lambda}_{k+1} - \boldsymbol{\lambda}^{\mu}, \boldsymbol{\lambda}_{k+1} - \boldsymbol{\lambda}_k \rangle = \frac{1}{2}\left( \|\boldsymbol{\lambda}_k - \boldsymbol{\lambda}^{\mu}\| + \|\boldsymbol{\lambda}_{k+1} - \boldsymbol{\lambda}_k\| - \|\boldsymbol{\lambda}_{k+1} - \boldsymbol{\lambda}^{\mu}\| \right),$$

and similarly,

$$\langle -\boldsymbol{y}_{k+1} + \boldsymbol{y}_k, -\boldsymbol{y}_{k+1} + \boldsymbol{y}^{\mu} \rangle = \frac{1}{2}\left( \|-\boldsymbol{y}_k + \boldsymbol{y}^{\mu}\| - \|-\boldsymbol{y}_{k+1} + \boldsymbol{y}^{\mu}\| - \|-\boldsymbol{y}_{k+1} + \boldsymbol{y}_k\| \right),$$

and by plugging these into (29) we get:

$$LHS = \frac{1}{2\beta}\|\boldsymbol{\lambda}_k - \boldsymbol{\lambda}^{\mu}\|^2 - \frac{1}{2\beta}\|\boldsymbol{\lambda}_{k+1} - \boldsymbol{\lambda}^{\mu}\|^2 - \frac{1}{2\beta}\|\boldsymbol{\lambda}_{k+1} - \boldsymbol{\lambda}_k\|^2$$
$$+ \frac{\beta}{2}\|-\boldsymbol{y}_k + \boldsymbol{y}^{\mu}\|^2 - \frac{\beta}{2}\|-\boldsymbol{y}_{k+1} + \boldsymbol{y}^{\mu}\|^2 - \frac{\beta}{2}\|-\boldsymbol{y}_{k+1} + \boldsymbol{y}_k\|^2$$
$$+ \langle -\boldsymbol{y}_{k+1} + \boldsymbol{y}_k, \boldsymbol{\lambda}_{k+1} - \boldsymbol{\lambda}_k \rangle - \beta \langle \boldsymbol{q}_{k+1}, \boldsymbol{x}_{k+1} - \boldsymbol{x}^{\mu} \rangle - \beta \langle \boldsymbol{r}_{k+1}, \boldsymbol{y}_{k+1} - \boldsymbol{y}^{\mu} \rangle. \tag{30}$$

On the other hand, (LB.17-1) which states that $\langle \hat{\nabla}_{\varepsilon_{k+1}} g(\boldsymbol{y}_{k+1}), \boldsymbol{y}_{k+1} - \boldsymbol{y} \rangle + \langle \boldsymbol{\lambda}_{k+1}, -\boldsymbol{y}_{k+1} + \boldsymbol{y} \rangle = -\beta \langle \boldsymbol{r}_{k+1}, \boldsymbol{y}_{k+1} - \boldsymbol{y} \rangle$, also asserts:

$$\langle \hat{\nabla}_{\varepsilon_k} g(\boldsymbol{y}_k), \boldsymbol{y}_k - \boldsymbol{y} \rangle + \langle \boldsymbol{\lambda}_k, -\boldsymbol{y}_k + \boldsymbol{y} \rangle = -\beta \langle \boldsymbol{r}_k, \boldsymbol{y}_k - \boldsymbol{y} \rangle. \tag{31}$$

Letting $\boldsymbol{y} = \boldsymbol{y}_k$ in (LB.17-1), and $\boldsymbol{y} = \boldsymbol{y}_{k+1}$ in (31), and adding them together yields:

$$\langle \hat{\nabla}_{\varepsilon_{k+1}} g(\boldsymbol{y}_{k+1}) - \hat{\nabla}_{\varepsilon_k} g(\boldsymbol{y}_k), \boldsymbol{y}_{k+1} - \boldsymbol{y}_k \rangle + \langle \boldsymbol{\lambda}_{k+1} - \boldsymbol{\lambda}_k, -\boldsymbol{y}_{k+1} + \boldsymbol{y}_k \rangle \tag{32}$$
$$= -\beta \langle \boldsymbol{r}_{k+1} - \boldsymbol{r}_k, \boldsymbol{y}_{k+1} - \boldsymbol{y}_k \rangle. \tag{33}$$

By the definition of $\epsilon$-subdifferential as per Def.B.13 we have:

$$\varepsilon_k + g(\boldsymbol{y}_{k+1}) \geq g(\boldsymbol{y}_k) + \langle \hat{\nabla}_{\varepsilon_k} g(\boldsymbol{y}_k), \boldsymbol{y}_{k+1} - \boldsymbol{y}_k \rangle, \quad and$$
$$\varepsilon_{k+1} + g(\boldsymbol{y}_k) \geq g(\boldsymbol{y}_{k+1}) + \langle \hat{\nabla}_{\varepsilon_{k+1}} g(\boldsymbol{y}_{k+1}), \boldsymbol{y}_k - \boldsymbol{y}_{k+1} \rangle.$$

Adding together the above two inequalities, we obtain:

$$\langle \hat{\nabla}_{\varepsilon_{k+1}} g(\boldsymbol{y}_{k+1}) - \hat{\nabla}_{\varepsilon_k} g(\boldsymbol{y}_k), \boldsymbol{y}_{k+1} - \boldsymbol{y}_k \rangle \geq -\varepsilon_{k+1} - \varepsilon_k \,. \tag{34}$$

Combining (32) and (34), we deduce:

$$\langle \boldsymbol{\lambda}_{k+1} - \boldsymbol{\lambda}_k, -\boldsymbol{y}_{k+1} + \boldsymbol{y}_k \rangle \leq -\beta \langle \boldsymbol{r}_{k+1} - \boldsymbol{r}_k, \boldsymbol{y}_{k+1} - \boldsymbol{y}_k \rangle + \varepsilon_{k+1} + \varepsilon_k \,. \tag{35}$$

Lastly, plugging it into (30) gives the first inequality of Lemma B.18.

The second inequality of Lemma B.18 follows similarly. $\qquad\square$

**Lemma B.19.** *For the $\boldsymbol{x}_{k+1}$, $\boldsymbol{y}_{k+1}$, and $\boldsymbol{\lambda}_{k+1}$ iterates of the inexact ACVI—Algorithm 1—we have:*

$$
\begin{aligned}
f_k(\boldsymbol{x}_{k+1} &+ \boldsymbol{q}_{k+1}) + g(\boldsymbol{y}_{k+1}) - f_k(\boldsymbol{x}^\mu) - g(\boldsymbol{y}^\mu) + \langle \boldsymbol{\lambda}^\mu, \boldsymbol{x}_{k+1} + \boldsymbol{q}_{k+1} - \boldsymbol{y}_{k+1} \rangle \\
&\leq \frac{1}{2\beta} \|\boldsymbol{\lambda}_k - \boldsymbol{\lambda}^\mu\|^2 - \frac{1}{2\beta} \|\boldsymbol{\lambda}_{k+1} - \boldsymbol{\lambda}^\mu\|^2 \\
&\quad + \frac{\beta}{2} \|-\boldsymbol{y}_k + \boldsymbol{y}^\mu\|^2 - \frac{\beta}{2} \|-\boldsymbol{y}_{k+1} + \boldsymbol{y}^\mu\|^2 \\
&\quad - \frac{1}{2\beta} \|\boldsymbol{\lambda}_{k+1} - \boldsymbol{\lambda}_k\|^2 - \frac{\beta}{2} \|-\boldsymbol{y}_{k+1} + \boldsymbol{y}_k\|^2 \\
&\quad - \frac{1}{2\beta} \|\boldsymbol{\lambda}_{k+1} - \boldsymbol{\lambda}_k\|^2 - \frac{\beta}{2} \|\boldsymbol{y}_k - \boldsymbol{y}_{k+1}\|^2 \\
&\quad - \beta \langle \boldsymbol{r}_{k+1} - \boldsymbol{r}_k, \boldsymbol{y}_{k+1} - \boldsymbol{y}_k \rangle - \beta \langle \boldsymbol{r}_{k+1}, \boldsymbol{y}_{k+1} - \boldsymbol{y}^\mu \rangle \\
&\quad + \varepsilon_k + 2\varepsilon_{k+1} - \langle \boldsymbol{q}_{k+1}, \boldsymbol{\lambda}_k - \boldsymbol{\lambda}^\mu + \beta(\boldsymbol{x}_{k+1} - \boldsymbol{y}_k) + \beta(\boldsymbol{x}_{k+1} - \boldsymbol{x}^\mu) \rangle \,.
\end{aligned}
\tag{LB.19}
$$

*Proof of Lemma B.19.* From the convexity of $f_k(\boldsymbol{x})$ and $g(\boldsymbol{y})$ and Eq. (28) which asserts that $\hat{\nabla} f_k(\boldsymbol{x}_{k+1} + \boldsymbol{q}_{k+1}) \in \partial f_k(\boldsymbol{x}_{k+1} + \boldsymbol{q}_{k+1})$ and $\hat{\nabla}_{\varepsilon_{k+1}} g(\boldsymbol{y}_{k+1}) \in \partial_{\varepsilon_{k+1}} g(\boldsymbol{y}_{k+1})$, it follows for the LHS of Lemma B.19 that:

$$
\begin{aligned}
f_k(\boldsymbol{x}_{k+1} &+ \boldsymbol{q}_{k+1}) + g(\boldsymbol{y}_{k+1}) - f_k(\boldsymbol{x}^\mu) - g(\boldsymbol{y}^\mu) + \langle \boldsymbol{\lambda}^\mu, \boldsymbol{x}_{k+1} + \boldsymbol{q}_{k+1} - \boldsymbol{y}_{k+1} \rangle \\
&\leq \langle \hat{\nabla} f_k(\boldsymbol{x}_{k+1} + \boldsymbol{q}_{k+1}), \boldsymbol{x}_{k+1} + \boldsymbol{q}_{k+1} - \boldsymbol{x}^\mu \rangle + \langle \hat{\nabla}_{\varepsilon_{k+1}} g(\boldsymbol{y}_{k+1}), \boldsymbol{y}_{k+1} - \boldsymbol{y}^\mu \rangle \\
&\quad + \varepsilon_{k+1} + \langle \boldsymbol{\lambda}^\mu, \boldsymbol{x}_{k+1} + \boldsymbol{q}_{k+1} - \boldsymbol{y}_{k+1} \rangle \,.
\end{aligned}
$$

Finally, by plugging in the first part of Lemma B.18 and using (noisy-$\hat{\nabla} f_k$), Lemma B.19 follows, that is:

$$
\begin{aligned}
f(\boldsymbol{x}_{k+1} &+ \boldsymbol{q}_{k+1}) + g(\boldsymbol{y}_{k+1}) - f(\boldsymbol{x}^\mu) - g(\boldsymbol{y}^\mu) + \langle \boldsymbol{\lambda}^\mu, \boldsymbol{x}_{k+1} + \boldsymbol{q}_{k+1} - \boldsymbol{y}_{k+1} \rangle \\
&\leq \frac{1}{2\beta} \|\boldsymbol{\lambda}_k - \boldsymbol{\lambda}^\mu\|^2 - \frac{1}{2\beta} \|\boldsymbol{\lambda}_{k+1} - \boldsymbol{\lambda}^\mu\|^2 + \frac{\beta}{2} \|-\boldsymbol{y}_k + \boldsymbol{y}^\mu\|^2 - \frac{\beta}{2} \|-\boldsymbol{y}_{k+1} + \boldsymbol{y}^\mu\|^2 \\
&\quad - \frac{1}{2\beta} \|\boldsymbol{\lambda}_{k+1} - \boldsymbol{\lambda}_k\|^2 - \frac{\beta}{2} \|-\boldsymbol{y}_{k+1} + \boldsymbol{y}_k\|^2 - \beta \langle \boldsymbol{r}_{k+1} - \boldsymbol{r}_k, \boldsymbol{y}_{k+1} - \boldsymbol{y}_k \rangle \\
&\quad - \beta \langle \boldsymbol{r}_{k+1}, \boldsymbol{y}_{k+1} - \boldsymbol{y}^\mu \rangle + \varepsilon_k + 2\varepsilon_{k+1} \\
&\quad - \beta \langle \boldsymbol{q}_{k+1}, \boldsymbol{x}_{k+1} - \boldsymbol{x}^\mu \rangle - \langle \boldsymbol{q}_{k+1}, \boldsymbol{\lambda}_k - \boldsymbol{\lambda}^\mu + \beta(\boldsymbol{x}_{k+1} - \boldsymbol{y}_k) + \beta \boldsymbol{q}_{k+1} \rangle \\
&\leq \frac{1}{2\beta} \|\boldsymbol{\lambda}_k - \boldsymbol{\lambda}^\mu\|^2 - \frac{1}{2\beta} \|\boldsymbol{\lambda}_{k+1} - \boldsymbol{\lambda}^\mu\|^2 + \frac{\beta}{2} \|-\boldsymbol{y}_k + \boldsymbol{y}^\mu\|^2 - \frac{\beta}{2} \|-\boldsymbol{y}_{k+1} + \boldsymbol{y}^\mu\|^2 \\
&\quad - \frac{1}{2\beta} \|\boldsymbol{\lambda}_{k+1} - \boldsymbol{\lambda}_k\|^2 - \frac{\beta}{2} \|-\boldsymbol{y}_{k+1} + \boldsymbol{y}_k\|^2 - \beta \langle \boldsymbol{r}_{k+1} - \boldsymbol{r}_k, \boldsymbol{y}_{k+1} - \boldsymbol{y}_k \rangle \\
&\quad - \beta \langle \boldsymbol{r}_{k+1}, \boldsymbol{y}_{k+1} - \boldsymbol{y}^\mu \rangle + \varepsilon_k + 2\varepsilon_{k+1} \\
&\quad - \langle \boldsymbol{q}_{k+1}, \boldsymbol{\lambda}_k - \boldsymbol{\lambda}^\mu + \beta(\boldsymbol{x}_{k+1} - \boldsymbol{y}_k) + \beta(\boldsymbol{x}_{k+1} - \boldsymbol{x}^\mu) \rangle \,.
\end{aligned}
$$

$\qquad\square$

The following theorem upper bounds the analogous quantity but for $f_k(\cdot)$ (instead of $f$), and further asserts that the difference between the $\boldsymbol{x}_{k+1}$ and $\boldsymbol{y}_{k+1}$ iterates of inexact ACVI (Algorithm 1) tends to 0 asymptotically.

**Theorem B.20** (Asymptotic convergence of $(\boldsymbol{x}_{k+1} - \boldsymbol{y}_{k+1})$ of I-ACVI). *Assume that $\sum_{i=1}^{\infty}(\sigma_i + \sqrt{\varepsilon_i}) < +\infty$, then for the $\boldsymbol{x}_{k+1}$, $\boldsymbol{y}_{k+1}$, and $\boldsymbol{\lambda}_{k+1}$ iterates of the inexact ACVI—Algorithm 1—we have:*

$$
\begin{aligned}
f_k(\boldsymbol{x}_{k+1} + \boldsymbol{q}_{k+1}) &- f_k(\boldsymbol{x}_k^{\mu}) + g(\boldsymbol{y}_{k+1}) - g(\boldsymbol{y}_k^{\mu}) \\
&\leq \|\boldsymbol{\lambda}_{k+1}\| \, \|\boldsymbol{x}_{k+1} - \boldsymbol{y}_{k+1}\| + \beta \, \|\boldsymbol{y}_{k+1} - \boldsymbol{y}_k\| \, \|\boldsymbol{x}_{k+1} - \boldsymbol{x}_k^{\mu}\| \\
&\quad + \beta \sigma_{k+1} \, \|\boldsymbol{x}_{k+1} - \boldsymbol{x}_k^{\mu}\| + \sqrt{2\varepsilon_{k+1}\beta} \, \|\boldsymbol{y}_{k+1} - \boldsymbol{y}_k^{\mu}\| + \varepsilon_{k+1} \to 0 \,,
\end{aligned}
$$
$$\text{(TB.20-}f_k\text{-UB)}$$

*and*

$$
\boldsymbol{x}_{k+1} - \boldsymbol{y}_{k+1} \to \boldsymbol{0} \,, \qquad \text{as } k \to \infty \,.
$$

*Proof of Lemma B.20.* Recall from (LB.2-$f$) of Lemma B.2 that by setting $\boldsymbol{x} \equiv \boldsymbol{x}_{k+1} + \boldsymbol{q}_{k+1}, \boldsymbol{y} \equiv \boldsymbol{y}_{k+1}$ it asserts that:

$$
f(\boldsymbol{x}_{k+1} + \boldsymbol{q}_{k+1}) - f(\boldsymbol{x}^{\mu}) + g(\boldsymbol{y}_{k+1}) - g(\boldsymbol{y}^{\mu}) + \langle \boldsymbol{\lambda}^{\mu}, \boldsymbol{x}_{k+1} + \boldsymbol{q}_{k+1} - \boldsymbol{y}_{k+1} \rangle \geq 0 \,.
$$

Further, notice that the LHS of the above inequality overlaps with that of (LB.19). This implies that the RHS of (LB.19) has to be non-negative. Hence, we have that:

$$
\begin{aligned}
\frac{1}{2\beta} &\|\boldsymbol{\lambda}_{k+1} - \boldsymbol{\lambda}_k\|^2 + \frac{\beta}{2} \| -\boldsymbol{y}_{k+1} + \boldsymbol{y}_k\|^2 \\
\leq\, & \frac{1}{2\beta} \|\boldsymbol{\lambda}_k - \boldsymbol{\lambda}^{\mu}\|^2 - \frac{1}{2\beta} \|\boldsymbol{\lambda}_{k+1} - \boldsymbol{\lambda}^{\mu}\|^2 \\
& + \frac{\beta}{2} \| -\boldsymbol{y}_k + \boldsymbol{y}^{\mu}\|^2 - \frac{\beta}{2} \| -\boldsymbol{y}_{k+1} + \boldsymbol{y}^{\mu}\|^2 \\
& - \beta \langle \boldsymbol{r}_{k+1} - \boldsymbol{r}_k, \boldsymbol{y}_{k+1} - \boldsymbol{y}_k \rangle - \beta \langle \boldsymbol{r}_{k+1}, \boldsymbol{y}_{k+1} - \boldsymbol{y}^{\mu} \rangle \\
& + \varepsilon_k + 2\varepsilon_{k+1} - \langle \boldsymbol{q}_{k+1}, \boldsymbol{\lambda}_k - \boldsymbol{\lambda}^{\mu} + \beta(\boldsymbol{x}_{k+1} - \boldsymbol{y}_k) + \beta(\boldsymbol{x}_{k+1} - \boldsymbol{x}^{\mu}) \rangle \,.
\end{aligned}
$$

Recall that $\|\boldsymbol{r}_{k+1}\| \leq \sqrt{\frac{2\varepsilon_{k+1}}{\beta}}$ and $\|\boldsymbol{q}_{k+1}\| \leq \sigma_{k+1}$ (see Lemma B.14 and Lemma B.15), by Cauchy-Schwarz inequality we have:

$$
\begin{aligned}
\frac{1}{2\beta} &\|\boldsymbol{\lambda}_{k+1} - \boldsymbol{\lambda}_k\|^2 + \frac{\beta}{2} \| -\boldsymbol{y}_{k+1} + \boldsymbol{y}_k\|^2 \\
\leq\, & \frac{1}{2\beta} \|\boldsymbol{\lambda}_k - \boldsymbol{\lambda}^{\mu}\|^2 - \frac{1}{2\beta} \|\boldsymbol{\lambda}_{k+1} - \boldsymbol{\lambda}^{\mu}\|^2 \\
& + \frac{\beta}{2} \| -\boldsymbol{y}_k + \boldsymbol{y}^{\mu}\|^2 - \frac{\beta}{2} \| -\boldsymbol{y}_{k+1} + \boldsymbol{y}^{\mu}\|^2 \\
& + \sqrt{2\beta}(\sqrt{\varepsilon_{k+1}} + \sqrt{\varepsilon_k}) \, \|\boldsymbol{y}_{k+1} - \boldsymbol{y}_k\| + \sqrt{2\beta\varepsilon_{k+1}} \, \|\boldsymbol{y}_{k+1} - \boldsymbol{y}^{\mu}\| \\
& + \varepsilon_k + 2\varepsilon_{k+1} + \sigma_{k+1}(\|\boldsymbol{\lambda}_k - \boldsymbol{\lambda}^{\mu}\| + \beta \, \|\boldsymbol{x}_{k+1} - \boldsymbol{y}_k\| + \beta \, \|\boldsymbol{x}_{k+1} - \boldsymbol{x}^{\mu}\|) \,.
\end{aligned}
\tag{36}
$$

Summing over $k = 0, \ldots, \infty$, we have:

$$
\begin{aligned}
\sum_{k=0}^{\infty} &\left( \frac{1}{2\beta} \|\boldsymbol{\lambda}_{k+1} - \boldsymbol{\lambda}_k\|^2 + \frac{\beta}{2} \| -\boldsymbol{y}_{k+1} + \boldsymbol{y}_k\|^2 \right) \\
\leq\, & \frac{1}{2\beta} \|\boldsymbol{\lambda}_0 - \boldsymbol{\lambda}^{\mu}\|^2 + \frac{\beta}{2} \| -\boldsymbol{y}_0 + \boldsymbol{y}^{\mu}\|^2 \\
& + \sqrt{2\beta} \sum_{k=0}^{\infty} \left( (\sqrt{\varepsilon_{k+1}} + \sqrt{\varepsilon_k}) \, \|\boldsymbol{y}_{k+1} - \boldsymbol{y}_k\| + \sqrt{\varepsilon_{k+1}} \, \|\boldsymbol{y}_{k+1} - \boldsymbol{y}^{\mu}\| \right) + 3 \sum_{k=0}^{\infty} \varepsilon_k \\
& + \sum_{k=0}^{\infty} \sigma_{k+1} \left( \|\boldsymbol{\lambda}_k - \boldsymbol{\lambda}^{\mu}\| + \beta \, \|\boldsymbol{x}_{k+1} - \boldsymbol{y}_k\| + \beta \, \|\boldsymbol{x}_{k+1} - \boldsymbol{x}^{\mu}\| \right) \,.
\end{aligned}
\tag{37}
$$

Also notice that by simply reorganizing (36) we have:

$$
\frac{1}{2\beta}\|\boldsymbol{\lambda}_{k+1} - \boldsymbol{\lambda}^{\mu}\|^2 + \frac{\beta}{2}\|-\boldsymbol{y}_{k+1} + \boldsymbol{y}^{\mu}\|^2
$$

$$
\begin{aligned}
\leq &\frac{1}{2\beta}\|\boldsymbol{\lambda}_k - \boldsymbol{\lambda}^{\mu}\|^2 + \frac{\beta}{2}\|-\boldsymbol{y}_k + \boldsymbol{y}^{\mu}\|^2 - \frac{1}{2\beta}\|\boldsymbol{\lambda}_{k+1} - \boldsymbol{\lambda}_k\|^2 - \frac{\beta}{2}\|-\boldsymbol{y}_{k+1} + \boldsymbol{y}_k\|^2 \\
&+ \sqrt{2\beta}(\sqrt{\varepsilon_{k+1}} + \sqrt{\varepsilon_k})\|\boldsymbol{y}_{k+1} - \boldsymbol{y}_k\| + \sqrt{2\beta\varepsilon_{k+1}}\|\boldsymbol{y}_{k+1} - \boldsymbol{y}^{\mu}\| \\
&+ \varepsilon_k + 2\varepsilon_{k+1} + \sigma_{k+1}(\|\boldsymbol{\lambda}_k - \boldsymbol{\lambda}^{\mu}\| + \beta\|\boldsymbol{x}_{k+1} - \boldsymbol{y}_k\| + \beta\|\boldsymbol{x}_{k+1} - \boldsymbol{x}^{\mu}\|) \\
\leq &\frac{1}{2\beta}\|\boldsymbol{\lambda}_k - \boldsymbol{\lambda}^{\mu}\|^2 + \frac{\beta}{2}\|-\boldsymbol{y}_k + \boldsymbol{y}^{\mu}\|^2 \\
&+ \sqrt{2\beta}(\sqrt{\varepsilon_{k+1}} + \sqrt{\varepsilon_k})\|\boldsymbol{y}_{k+1} - \boldsymbol{y}_k\| + \sqrt{2\beta\varepsilon_{k+1}}\|\boldsymbol{y}_{k+1} - \boldsymbol{y}^{\mu}\| \\
&+ \varepsilon_k + 2\varepsilon_{k+1} + \sigma_{k+1}(\|\boldsymbol{\lambda}_k - \boldsymbol{\lambda}^{\mu}\| + \beta\|\boldsymbol{x}_{k+1} - \boldsymbol{y}_k\| + \beta\|\boldsymbol{x}_{k+1} - \boldsymbol{x}^{\mu}\|) \\
\leq &\frac{1}{2\beta}\|\boldsymbol{\lambda}_0 - \boldsymbol{\lambda}^{\mu}\|^2 + \frac{\beta}{2}\|-\boldsymbol{y}_0 + \boldsymbol{y}^{\mu}\|^2 \\
&+ \sqrt{2\beta}\sum_{i=0}^k(\sqrt{\varepsilon_{i+1}} + \sqrt{\varepsilon_i})\|\boldsymbol{y}_{i+1} - \boldsymbol{y}_i\| + \sqrt{2\beta}\sum_{i=0}^k\sqrt{\varepsilon_{i+1}}\|\boldsymbol{y}_{i+1} - \boldsymbol{y}^{\mu}\| \\
&+ \sum_{i=0}^k \varepsilon_i + 2\sum_{i=0}^k \varepsilon_{i+1} \\
&+ \sum_{i=0}^k \sigma_{i+1}(\|\boldsymbol{\lambda}_i - \boldsymbol{\lambda}^{\mu}\| + \beta\|\boldsymbol{x}_{i+1} - \boldsymbol{y}_i\| + \beta\|\boldsymbol{x}_{i+1} - \boldsymbol{x}^{\mu}\|),
\end{aligned}
\tag{38}
$$

where the second inequality follows because the norm is non-negative.

From the above inequality we deduce:

$$
\begin{aligned}
&\frac{1}{4\beta}\big(\|\boldsymbol{\lambda}_{k+1} - \boldsymbol{\lambda}^{\mu}\| + \beta\|\boldsymbol{y}_{k+1} - \boldsymbol{y}^{\mu}\|\big)^2 \\
\leq &\frac{1}{2\beta}\|\boldsymbol{\lambda}_{k+1} - \boldsymbol{\lambda}^{\mu}\|^2 + \frac{\beta}{2}\|\boldsymbol{y}_{k+1} - \boldsymbol{y}^{\mu}\|^2 \\
\leq &\frac{1}{2\beta}\|\boldsymbol{\lambda}_0 - \boldsymbol{\lambda}^{\mu}\|^2 + \frac{\beta}{2}\|-\boldsymbol{y}_0 + \boldsymbol{y}^{\mu}\|^2 \\
&+ \sqrt{2\beta}\sum_{i=0}^k(\sqrt{\varepsilon_{i+1}} + \sqrt{\varepsilon_i})\|\boldsymbol{y}_{i+1} - \boldsymbol{y}_i\| + \sqrt{2\beta}\sum_{i=0}^k\sqrt{\varepsilon_{i+1}}\|\boldsymbol{y}_{i+1} - \boldsymbol{y}^{\mu}\| \\
&+ \sum_{i=0}^k \varepsilon_i + 2\sum_{i=0}^k \varepsilon_{i+1} + \sum_{i=0}^k \sigma_{i+1}(\|\boldsymbol{\lambda}_i - \boldsymbol{\lambda}^{\mu}\| + \beta\|\boldsymbol{x}_{i+1} - \boldsymbol{y}_i\| + \beta\|\boldsymbol{x}_{i+1} - \boldsymbol{x}^{\mu}\|),
\end{aligned}
\tag{39}
$$

where the first inequality is the Cauchy-Schwarz inequality.

Using (LB.16-3) and (LB.16-8), we have:

$$
\begin{aligned}
\|\boldsymbol{x}_{i+1} - \boldsymbol{x}^{\mu}\| &= \|\boldsymbol{y}_{i+1} - \boldsymbol{y}^{\mu} + \boldsymbol{x}_{i+1} - \boldsymbol{y}_{i+1}\| \\
&\leq \|\boldsymbol{y}_{i+1} - \boldsymbol{y}^{\mu}\| + \frac{1}{\beta}\|\boldsymbol{\lambda}_{i+1} - \boldsymbol{\lambda}_i\| \\
&\leq \|\boldsymbol{y}_{i+1} - \boldsymbol{y}^{\mu}\| + \frac{1}{\beta}\|\boldsymbol{\lambda}_{i+1} - \boldsymbol{\lambda}^{\mu}\| + \frac{1}{\beta}\|\boldsymbol{\lambda}_i - \boldsymbol{\lambda}^{\mu}\|,
\end{aligned}
\tag{40}
$$

$$
\begin{aligned}
\|\boldsymbol{x}_{i+1} - \boldsymbol{y}_i\| &\leq \|\boldsymbol{x}_{i+1} - \boldsymbol{x}^{\mu}\| + \|\boldsymbol{y}_i - \boldsymbol{y}^{\mu}\| \\
&\leq \|\boldsymbol{y}_i - \boldsymbol{y}^{\mu}\| + \|\boldsymbol{y}_{i+1} - \boldsymbol{y}^{\mu}\| + \frac{1}{\beta}\|\boldsymbol{\lambda}_{i+1} - \boldsymbol{\lambda}^{\mu}\| + \frac{1}{\beta}\|\boldsymbol{\lambda}_i - \boldsymbol{\lambda}^{\mu}\|,
\end{aligned}
\tag{41}
$$

$$\|\boldsymbol{y}_{i+1} - \boldsymbol{y}_i\| \le \|\boldsymbol{y}_{i+1} - \boldsymbol{y}^\mu\| + \|\boldsymbol{y}_i - \boldsymbol{y}^\mu\| \,. \tag{42}$$

Plugging these into (39), we obtain:

$$\frac{1}{4\beta}\big(\,\|\boldsymbol{\lambda}_{k+1} - \boldsymbol{\lambda}^\mu\| + \beta\,\|\boldsymbol{y}_{k+1} - \boldsymbol{y}^\mu\|\,\big)^2$$

$$\le \frac{1}{2\beta}\,\|\boldsymbol{\lambda}_0 - \boldsymbol{\lambda}^\mu\|^2 + \frac{\beta}{2}\,\|\boldsymbol{y}_0 - \boldsymbol{y}^\mu\|^2 + \sqrt{2\beta}\sum_{i=0}^{k}(\sqrt{\varepsilon_{i+1}} + \sqrt{\varepsilon_i})\big(\,\|\boldsymbol{y}_{i+1} - \boldsymbol{y}^\mu\| + \|\boldsymbol{y}_i - \boldsymbol{y}^\mu\|\,\big)$$

$$+ \sqrt{2\beta}\sum_{i=0}^{k}\sqrt{\varepsilon_{i+1}}\,\|\boldsymbol{y}_{i+1} - \boldsymbol{y}^\mu\| + \sum_{i=0}^{k}\varepsilon_i + 2\sum_{i=0}^{k}\varepsilon_{i+1}$$

$$+ \sum_{i=0}^{k}\sigma_{i+1}\bigg(\,\|\boldsymbol{\lambda}_i - \boldsymbol{\lambda}^\mu\| + \beta\big(\,\|\boldsymbol{y}_i - \boldsymbol{y}^\mu\| + \|\boldsymbol{y}_{i+1} - \boldsymbol{y}^\mu\| + \frac{1}{\beta}\,\|\boldsymbol{\lambda}_{i+1} - \boldsymbol{\lambda}^\mu\|$$

$$+ \frac{1}{\beta}\,\|\boldsymbol{\lambda}_i - \boldsymbol{\lambda}^\mu\|\,\big) + \beta\big(\,\|\boldsymbol{y}_{i+1} - \boldsymbol{y}^\mu\| + \frac{1}{\beta}\,\|\boldsymbol{\lambda}_{i+1} - \boldsymbol{\lambda}^\mu\| + \frac{1}{\beta}\,\|\boldsymbol{\lambda}_i - \boldsymbol{\lambda}^\mu\|\,\big)\bigg)$$

$$\le \frac{1}{2\beta}\,\|\boldsymbol{\lambda}_0 - \boldsymbol{\lambda}^\mu\|^2 + \frac{\beta}{2}\,\|\boldsymbol{y}_0 - \boldsymbol{y}^\mu\|^2 + \sum_{i=0}^{k}\varepsilon_i + 2\sum_{i=0}^{k}\varepsilon_{i+1}$$

$$+ \sum_{i=0}^{k}\Big(\sqrt{2\beta}\big(\sqrt{\varepsilon_{i+1}} + \sqrt{\varepsilon_i}\big) + \beta\sigma_{i+1}\Big)\,\|\boldsymbol{y}_i - \boldsymbol{y}^\mu\| + \sum_{i=0}^{k}3\sigma_{i+1}\,\|\boldsymbol{\lambda}_i - \boldsymbol{\lambda}^\mu\|$$

$$+ \sum_{i=0}^{k}\Big(\sqrt{2\beta}\big(2\sqrt{\varepsilon_{i+1}} + \sqrt{\varepsilon_i}\big) + 2\beta\sigma_{i+1}\Big)\,\|\boldsymbol{y}_{i+1} - \boldsymbol{y}^\mu\| + \sum_{i=0}^{k}2\sigma_{i+1}\,\|\boldsymbol{\lambda}_{i+1} - \boldsymbol{\lambda}^\mu\|$$

$$= \frac{1}{2\beta}\,\|\boldsymbol{\lambda}_0 - \boldsymbol{\lambda}^\mu\|^2 + \frac{\beta}{2}\,\|\boldsymbol{y}_0 - \boldsymbol{y}^\mu\|^2 + \sum_{i=0}^{k}\varepsilon_i + 2\sum_{i=0}^{k}\varepsilon_{i+1}$$

$$+ \big(\sqrt{2\beta}\sqrt{\varepsilon_1} + \beta\sigma_1\big)\,\|\boldsymbol{y}_0 - \boldsymbol{y}^\mu\| + 3\sigma_1\,\|\boldsymbol{\lambda}_0 - \boldsymbol{\lambda}^\mu\|$$

$$+ \sum_{i=1}^{k}\big(\sqrt{2\beta}\big(\sqrt{\varepsilon_{i+1}} + \sqrt{\varepsilon_i}\big) + \beta\sigma_{i+1}\big)\,\|\boldsymbol{y}_i - \boldsymbol{y}^\mu\|$$

$$+ \sum_{i=1}^{k+1}\big(\sqrt{2\beta}\big(2\sqrt{\varepsilon_i} + \sqrt{\varepsilon_{i-1}}\big) + 2\beta\sigma_i\big)\,\|\boldsymbol{y}_i - \boldsymbol{y}^\mu\|$$

$$+ \sum_{i=1}^{k}3\sigma_{i+1}\,\|\boldsymbol{\lambda}_i - \boldsymbol{\lambda}^\mu\| + \sum_{i=1}^{k+1}2\sigma_i\,\|\boldsymbol{\lambda}_i - \boldsymbol{\lambda}^\mu\|$$

$$\le \frac{1}{2\beta}\,\|\boldsymbol{\lambda}_0 - \boldsymbol{\lambda}^\mu\|^2 + \frac{\beta}{2}\,\|\boldsymbol{y}_0 - \boldsymbol{y}^\mu\|^2 + \big(\sqrt{2\beta}\sqrt{\varepsilon_1} + \beta\sigma_1\big)\,\|\boldsymbol{y}_0 - \boldsymbol{y}^\mu\| + 3\sigma_1\,\|\boldsymbol{\lambda}_0 - \boldsymbol{\lambda}^\mu\| + 3\sum_{i=1}^{k+1}\varepsilon_i$$

$$+ \sum_{i=1}^{k+1}\big(\sqrt{2\beta}\big(\sqrt{\varepsilon_{i+1}} + 3\sqrt{\varepsilon_i} + \sqrt{\varepsilon_{i-1}}\big) + \beta(2\sigma_i + \sigma_{i+1})\big)\,\|\boldsymbol{y}_i - \boldsymbol{y}^\mu\|$$

$$+ \sum_{i=1}^{k+1}(2\sigma_i + 3\sigma_{i+1})\,\|\boldsymbol{\lambda}_i - \boldsymbol{\lambda}^\mu\|$$

$$\le \frac{1}{2\beta}\,\|\boldsymbol{\lambda}_0 - \boldsymbol{\lambda}^\mu\|^2 + \frac{\beta}{2}\,\|\boldsymbol{y}_0 - \boldsymbol{y}^\mu\|^2 + \big(\sqrt{2\beta}\sqrt{\varepsilon_1} + \beta\sigma_1\big)\,\|\boldsymbol{y}_0 - \boldsymbol{y}^\mu\| + 3\sigma_1\,\|\boldsymbol{\lambda}_0 - \boldsymbol{\lambda}^\mu\| + 3\sum_{i=1}^{k+1}\varepsilon_i$$

$$+ \sum_{i=1}^{k+1}\big(\sqrt{\tfrac{2}{\beta}}\big(\sqrt{\varepsilon_{i+1}} + 3\sqrt{\varepsilon_i} + \sqrt{\varepsilon_{i-1}}\big) + (2\sigma_i + 3\sigma_{i+1})\big)\big(\beta\,\|\boldsymbol{y}_i - \boldsymbol{y}^\mu\| + \|\boldsymbol{\lambda}_i - \boldsymbol{\lambda}^\mu\|\big)\,,$$

From which we deduce:

$$
\begin{aligned}
&\left( \|\boldsymbol{\lambda}_{k+1} - \boldsymbol{\lambda}^\mu\| + \beta \|\boldsymbol{y}_{k+1} - \boldsymbol{y}^\mu\| \right)^2 \\
&\leq 2 \|\boldsymbol{\lambda}_0 - \boldsymbol{\lambda}^\mu\|^2 + 2\beta^2 \|\boldsymbol{y}_0 - \boldsymbol{y}^\mu\|^2 + 4\beta\left(\sqrt{2\beta}\sqrt{\varepsilon_1} + \beta\sigma_1\right) \|\boldsymbol{y}_0 - \boldsymbol{y}^\mu\| \\
&\quad + 12\beta\sigma_1 \|\boldsymbol{\lambda}_0 - \boldsymbol{\lambda}^\mu\| + 12\beta \sum_{i=1}^{k+1} \varepsilon_i \\
&\quad + 4\beta \sum_{i=1}^{k+1} \left( \sqrt{\frac{2}{\beta}}\left(\sqrt{\varepsilon_{i+1}} + 3\sqrt{\varepsilon_i} + \sqrt{\varepsilon_{i-1}}\right) + (2\sigma_i + 3\sigma_{i+1}) \right)\left(\beta \|\boldsymbol{y}_i - \boldsymbol{y}^\mu\| + \|\boldsymbol{\lambda}_i - \boldsymbol{\lambda}^\mu\|\right).
\end{aligned}
$$

Now we set $u_i \triangleq \beta \|\boldsymbol{y}_i - \boldsymbol{y}^\mu\| + \|\boldsymbol{\lambda}_i - \boldsymbol{\lambda}^\mu\|$, $\lambda_i \triangleq 4\beta\left(\sqrt{\frac{2}{\beta}}\left(\sqrt{\varepsilon_{i+1}} + 3\sqrt{\varepsilon_i} + \sqrt{\varepsilon_{i-1}}\right) + (2\sigma_i + 3\sigma_{i+1})\right)$ and $S_{k+1} \triangleq 2 \|\boldsymbol{\lambda}_0 - \boldsymbol{\lambda}^\mu\|^2 + 2\beta^2 \|\boldsymbol{y}_0 - \boldsymbol{y}^\mu\|^2 + 4\beta\left(\sqrt{2\beta}\sqrt{\varepsilon_1} + \beta\sigma_1\right) \|\boldsymbol{y}_0 - \boldsymbol{y}^\mu\| + 12\beta\sigma_1 \|\boldsymbol{\lambda}_0 - \boldsymbol{\lambda}^\mu\| + 12\beta \sum_{i=1}^{k+1} \varepsilon_i$ and Lemma B.12 to get:

$$
u_{k+1} \leq \underbrace{\frac{1}{2}\sum_{i=1}^{k+1} \lambda_i + \left( S_{k+1} + \left(\frac{1}{2}\sum_{i=1}^{k+1} \lambda_i\right)^2 \right)^{1/2}}_{A_{k+1}}, \tag{43}
$$

where we set the RHS of (43) to be $A_{k+1}$.

Note that when $\sum_{i=1}^{\infty}(\sigma_i + \sqrt{\varepsilon_i}) < +\infty$, we have $A^\mu \triangleq \lim_{k\to+\infty} A_k < +\infty$, and

$$
\|\boldsymbol{y}_k - \boldsymbol{y}^\mu\| \leq \frac{1}{\beta} A^\mu, \tag{44}
$$

$$
\|\boldsymbol{\lambda}_k - \boldsymbol{\lambda}^\mu\| \leq A^\mu. \tag{45}
$$

Using Eq. (37) we could further get:

$$
\|\boldsymbol{x}_k - \boldsymbol{x}^\mu\| \leq \frac{3}{\beta} A^\mu. \tag{46}
$$

Combining (40),(41) and (42) with (37) and using the above inequalities, we have:

$$
\begin{aligned}
&\sum_{k=0}^{\infty} \left( \frac{1}{2\beta}\|\boldsymbol{\lambda}_{k+1} - \boldsymbol{\lambda}_k\|^2 + \frac{\beta}{2}\|-\boldsymbol{y}_{k+1} + \boldsymbol{y}_k\|^2 \right) \\
&\leq \frac{1}{2\beta}\|\boldsymbol{\lambda}_0 - \boldsymbol{\lambda}^\mu\|^2 + \frac{\beta}{2}\|-\boldsymbol{y}_0 + \boldsymbol{y}^\mu\|^2 \\
&\quad + \sqrt{2\beta}\sum_{k=0}^{\infty} \left( \left(\sqrt{\varepsilon_{k+1}} + \sqrt{\varepsilon_k}\right) \cdot \frac{2}{\beta} A^\mu + \sqrt{\varepsilon_{k+1}} \cdot \frac{1}{\beta} A^\mu \right) + 3\sum_{k=0}^{\infty} \varepsilon_k \\
&\quad + \sum_{k=0}^{\infty} \sigma_{k+1}\left( A^\mu + \beta \cdot \frac{4}{\beta} A^\mu + \beta \cdot \frac{3}{\beta} A^\mu \right) \\
&\leq \frac{1}{2\beta}\|\boldsymbol{\lambda}_0 - \boldsymbol{\lambda}^\mu\|^2 + \frac{\beta}{2}\|-\boldsymbol{y}_0 + \boldsymbol{y}^\mu\|^2 + 5\sqrt{\frac{2}{\beta}} A^\mu \sum_{k=1}^{\infty} \sqrt{\varepsilon_k} + 3\sum_{k=1}^{\infty} \varepsilon_k + 8A^\mu \sum_{k=1}^{\infty} \sigma_k, \tag{47}
\end{aligned}
$$

from which we can see that $\boldsymbol{\lambda}_{k+1} - \boldsymbol{\lambda}_k \to \boldsymbol{0}$ and $\boldsymbol{y}_{k+1} - \boldsymbol{y}_k \to \boldsymbol{0}$.

Recall that:

$$
\boldsymbol{\lambda}_{k+1} - \boldsymbol{\lambda}_k = \beta(\boldsymbol{x}_{k+1} - \boldsymbol{y}_{k+1}),
$$

from which we deduce $\boldsymbol{x}_{k+1} - \boldsymbol{y}_{k+1} \to \boldsymbol{0}$.

Using the convexity of $f_k(\cdot)$ and $g(\cdot)$ for the LHS of Theorem B.20 we have:

$$\text{LHS} = f_k(\boldsymbol{x}_{k+1} + \boldsymbol{q}_{k+1}) - f_k(\boldsymbol{x}_k^\mu) + g(\boldsymbol{y}_{k+1}) - g(\boldsymbol{y}_k^\mu)$$
$$\le \langle \hat{\nabla} f_k(\boldsymbol{x}_{k+1} + \boldsymbol{q}_{k+1}), \boldsymbol{x}_{k+1} - \boldsymbol{x}_k^\mu \rangle + \langle \hat{\nabla}_{\varepsilon_{k+1}} g(\boldsymbol{y}_{k+1}), \boldsymbol{y}_{k+1} - \boldsymbol{y}_k^\mu \rangle + \varepsilon_{k+1}$$

Using (LB.17-2) with $\boldsymbol{x} \equiv \boldsymbol{x}_k^\mu, \boldsymbol{y} \equiv \boldsymbol{y}_k^\mu$ we have:

$$\text{LHS} \le - \langle \boldsymbol{\lambda}_{k+1}, \boldsymbol{x}_{k+1} - \boldsymbol{y}_{k+1} \underbrace{-\boldsymbol{x}_k^\mu + \boldsymbol{y}_k^\mu}_{=\boldsymbol{0}, \text{ due to (LB.3-9)}} \rangle + \beta \langle -\boldsymbol{y}_{k+1} + \boldsymbol{y}_k, \boldsymbol{x}_{k+1} - \boldsymbol{x}_k^\mu \rangle$$
$$- \beta \langle \boldsymbol{q}_{k+1}, \boldsymbol{x}_{k+1} - \boldsymbol{x}_k^\mu \rangle - \beta \langle \boldsymbol{r}_{k+1}, \boldsymbol{y}_{k+1} - \boldsymbol{y}_k^\mu \rangle + \varepsilon_{k+1} \,.$$

.

Hence, it follows that:

$$f_k(\boldsymbol{x}_{k+1} + \boldsymbol{q}_{k+1}) - f_k(\boldsymbol{x}_k^\mu) + g(\boldsymbol{y}_{k+1}) - g(\boldsymbol{y}_k^\mu)$$
$$\le - \langle \boldsymbol{\lambda}_{k+1}, \boldsymbol{x}_{k+1} - \boldsymbol{y}_{k+1} \rangle + \beta \langle -\boldsymbol{y}_{k+1} + \boldsymbol{y}_k, \boldsymbol{x}_{k+1} - \boldsymbol{x}_k^\mu \rangle$$
$$- \beta \langle \boldsymbol{q}_{k+1}, \boldsymbol{x}_{k+1} - \boldsymbol{x}_k^\mu \rangle - \beta \langle \boldsymbol{r}_{k+1}, \boldsymbol{y}_{k+1} - \boldsymbol{y}_k^\mu \rangle + \varepsilon_{k+1}$$
$$\le \|\boldsymbol{\lambda}_{k+1}\| \, \|\boldsymbol{x}_{k+1} - \boldsymbol{y}_{k+1}\| + \beta \, \|\boldsymbol{y}_{k+1} - \boldsymbol{y}_k\| \, \|\boldsymbol{x}_{k+1} - \boldsymbol{x}_k^\mu\|$$
$$+ \beta \sigma_{k+1} \, \|\boldsymbol{x}_{k+1} - \boldsymbol{x}_k^\mu\| + \sqrt{2\varepsilon_{k+1}\beta} \, \|\boldsymbol{y}_{k+1} - \boldsymbol{y}_k^\mu\| + \varepsilon_{k+1} \,,$$

where the last inequality follows from Cauchy-Schwarz.

Recall that $\mathcal{C}$ is compact and $D$ is the diameter of $\mathcal{C}$:

$$D \triangleq \sup_{\boldsymbol{x}, \boldsymbol{y} \in \mathcal{C}} \|\boldsymbol{x} - \boldsymbol{y}\| \,.$$

Combining with (42), we have:

$$\|\boldsymbol{y}_{k+1} - \boldsymbol{y}_k^\mu\| = \|\boldsymbol{y}_{k+1} - \boldsymbol{y}^\mu\| + \|\boldsymbol{y}^\mu - \boldsymbol{y}_{k+1}^\mu\| \le \frac{1}{\beta} A^\mu + D \,, \tag{48}$$

which implies that $\|\boldsymbol{y}_k - \boldsymbol{y}_k^\mu\|$ are bounded for all $k$. Similarly, using (40), we deduce:

$$\|\boldsymbol{x}_{k+1} - \boldsymbol{x}_k^\mu\| = \|\boldsymbol{x}_{k+1} - \boldsymbol{x}^\mu\| + \|\boldsymbol{x}^\mu - \boldsymbol{x}_{k+1}^\mu\| \le \frac{3}{\beta} A^\mu + D \,, \tag{49}$$

which implies that $\boldsymbol{x}_{k+1} - \boldsymbol{x}_k^\mu$ is also bounded. Note that when $\sum_{i=1}^\infty (\sigma_i + \sqrt{\varepsilon_i}) < +\infty$, we have $\lim_{k \to \infty} \sigma_k = \lim_{k \to \infty} \varepsilon_k = 0$. Thus, we have (TB.20-$f_k$-UB). $\qquad\square$

**Lemma B.21.** *Assume that $F$ is $L$-Lipschitz. For the $\boldsymbol{x}_{k+1}$, $\boldsymbol{y}_{k+1}$, and $\boldsymbol{\lambda}_{k+1}$ iterates of the ACVI— Algorithm 3—we have:*

$$\frac{1}{2\beta} \|\boldsymbol{\lambda}_{k+1} - \boldsymbol{\lambda}_k\|^2 + \frac{\beta}{2} \|-\boldsymbol{y}_{k+1} + \boldsymbol{y}_k\|^2$$

$$\le \frac{1}{2\beta} \|\boldsymbol{\lambda}_k - \boldsymbol{\lambda}_{k-1}\|^2 + \frac{\beta}{2} \|-\boldsymbol{y}_k + \boldsymbol{y}_{k-1}\|^2$$
$$+ (\sigma_{k+1} + \sigma_k) \Big( \beta \, \|\boldsymbol{y}_k - \boldsymbol{y}_{k-1}\| + (2\beta + L) \, \|\boldsymbol{y}_{k+1} - \boldsymbol{y}_k\|$$
$$+ \Big(2 + \frac{L}{\beta}\Big) \|\boldsymbol{\lambda}_{k+1} - \boldsymbol{\lambda}_k\| + \Big(\frac{L}{\beta} + 1\Big) \|\boldsymbol{\lambda}_k - \boldsymbol{\lambda}_{k-1}\| \Big)$$
$$+ \sqrt{2\beta}(\sqrt{\varepsilon_k} + \sqrt{\varepsilon_{k+1}}) \, \|\boldsymbol{y}_{k+1} - \boldsymbol{y}_k\| + \varepsilon_k + \varepsilon_{k+1} \,. \tag{LB.21}$$

*Proof of Lemma B.21.* (LB.17-2) gives:

$$\langle \hat{\nabla} f_{k-1}(\boldsymbol{x}_k + \boldsymbol{q}_k), \boldsymbol{x}_k - \boldsymbol{x} \rangle + \langle \hat{\nabla}_{\varepsilon_k} g(\boldsymbol{y}_k), \boldsymbol{y}_k - \boldsymbol{y} \rangle$$
$$= - \langle \boldsymbol{\lambda}_k, \boldsymbol{x}_k - \boldsymbol{y}_k - \boldsymbol{x} + \boldsymbol{y} \rangle + \beta \langle -\boldsymbol{y}_k + \boldsymbol{y}_{k-1}, \boldsymbol{x}_k - \boldsymbol{x} \rangle - \beta \langle \boldsymbol{q}_k, \boldsymbol{x}_k - \boldsymbol{x} \rangle - \beta \langle \boldsymbol{r}_k, \boldsymbol{y}_k - \boldsymbol{y} \rangle \,. \tag{50}$$

Letting $(\boldsymbol{x}, \boldsymbol{y}, \boldsymbol{\lambda}) = (\boldsymbol{x}_k, \boldsymbol{y}_k, \boldsymbol{\lambda}_k)$ in (LB.17-2) and $(\boldsymbol{x}, \boldsymbol{y}, \boldsymbol{\lambda}) = (\boldsymbol{x}_{k+1}, \boldsymbol{y}_{k+1}, \boldsymbol{\lambda}_{k+1})$ in (50), and adding them together, and using (LB.16-3), we have

$$
\begin{aligned}
&\langle \hat{\nabla} f_k(\boldsymbol{x}_{k+1} + \boldsymbol{q}_{k+1}) - \hat{\nabla} f_{k-1}(\boldsymbol{x}_k + \boldsymbol{q}_k), \boldsymbol{x}_{k+1} - \boldsymbol{x}_k \rangle + \langle \hat{\nabla}_{\varepsilon_{k+1}} g(\boldsymbol{y}_{k+1}) - \hat{\nabla}_{\varepsilon_k} g(\boldsymbol{y}_k), \boldsymbol{y}_{k+1} - \boldsymbol{y}_k \rangle \\
&= -\langle \boldsymbol{\lambda}_{k+1} - \boldsymbol{\lambda}_k, \boldsymbol{x}_{k+1} - \boldsymbol{y}_{k+1} - \boldsymbol{x}_k + \boldsymbol{y}_k \rangle + \beta \langle -\boldsymbol{y}_{k+1} + \boldsymbol{y}_k - (-\boldsymbol{y}_k + \boldsymbol{y}_{k-1}), \boldsymbol{x}_{k+1} - \boldsymbol{x}_k \rangle \\
&\quad - \beta \langle \boldsymbol{q}_{k+1} - \boldsymbol{q}_k, \boldsymbol{x}_{k+1} - \boldsymbol{x}_k \rangle - \beta \langle \boldsymbol{r}_{k+1} - \boldsymbol{r}_k, \boldsymbol{y}_{k+1} - \boldsymbol{y}_k \rangle \\
&= -\frac{1}{\beta} \langle \boldsymbol{\lambda}_{k+1} - \boldsymbol{\lambda}_k, \boldsymbol{\lambda}_{k+1} - \boldsymbol{\lambda}_k - (\boldsymbol{\lambda}_k - \boldsymbol{\lambda}_{k-1}) \rangle \\
&\quad + \langle -\boldsymbol{y}_{k+1} + \boldsymbol{y}_k + (\boldsymbol{y}_k - \boldsymbol{y}_{k-1}), \boldsymbol{\lambda}_{k+1} - \boldsymbol{\lambda}_k + \beta \boldsymbol{y}_{k+1} - (\boldsymbol{\lambda}_k - \boldsymbol{\lambda}_{k-1} + \beta \boldsymbol{y}_k) \rangle \\
&\quad - \beta \langle \boldsymbol{q}_{k+1} - \boldsymbol{q}_k, \boldsymbol{x}_{k+1} - \boldsymbol{x}_k \rangle - \beta \langle \boldsymbol{r}_{k+1} - \boldsymbol{r}_k, \boldsymbol{y}_{k+1} - \boldsymbol{y}_k \rangle \\
&= \frac{1}{2\beta} \left[ \|\boldsymbol{\lambda}_k - \boldsymbol{\lambda}_{k-1}\|^2 - \|\boldsymbol{\lambda}_{k+1} - \boldsymbol{\lambda}_k\|^2 - \|\boldsymbol{\lambda}_{k+1} - \boldsymbol{\lambda}_k - (\boldsymbol{\lambda}_k - \boldsymbol{\lambda}_{k-1})\|^2 \right] \\
&\quad + \frac{\beta}{2} \left[ \|-\boldsymbol{y}_k + \boldsymbol{y}_{k-1}\|^2 - \|-\boldsymbol{y}_{k+1} + \boldsymbol{y}_k\|^2 - \|-\boldsymbol{y}_{k+1} + \boldsymbol{y}_k - (-\boldsymbol{y}_k + \boldsymbol{y}_{k-1})\|^2 \right] \\
&\quad + \langle -\boldsymbol{y}_{k+1} + \boldsymbol{y}_k - (-\boldsymbol{y}_k + \boldsymbol{y}_{k-1}), \boldsymbol{\lambda}_{k+1} - \boldsymbol{\lambda}_k - (\boldsymbol{\lambda}_k - \boldsymbol{\lambda}_{k-1}) \rangle \\
&\quad - \beta \langle \boldsymbol{q}_{k+1} - \boldsymbol{q}_k, \boldsymbol{x}_{k+1} - \boldsymbol{x}_k \rangle - \beta \langle \boldsymbol{r}_{k+1} - \boldsymbol{r}_k, \boldsymbol{y}_{k+1} - \boldsymbol{y}_k \rangle \\
&= \frac{1}{2\beta} \left( \|\boldsymbol{\lambda}_k - \boldsymbol{\lambda}_{k-1}\|^2 - \|\boldsymbol{\lambda}_{k+1} - \boldsymbol{\lambda}_k\|^2 \right) + \frac{\beta}{2} \left( \|-\boldsymbol{y}_k + \boldsymbol{y}_{k-1}\|^2 - \|-\boldsymbol{y}_{k+1} + \boldsymbol{y}_k\|^2 \right) \\
&\quad - \frac{1}{2\beta} \|\boldsymbol{\lambda}_{k+1} - \boldsymbol{\lambda}_k - (\boldsymbol{\lambda}_k - \boldsymbol{\lambda}_{k-1})\|^2 - \frac{\beta}{2} \|-\boldsymbol{y}_{k+1} + \boldsymbol{y}_k - (-\boldsymbol{y}_k + \boldsymbol{y}_{k-1})\|^2 \\
&\quad + \langle -\boldsymbol{y}_{k+1} + \boldsymbol{y}_k - (-\boldsymbol{y}_k + \boldsymbol{y}_{k-1}), \boldsymbol{\lambda}_{k+1} - \boldsymbol{\lambda}_k - (\boldsymbol{\lambda}_k - \boldsymbol{\lambda}_{k-1}) \rangle \\
&\quad - \beta \langle \boldsymbol{q}_{k+1} - \boldsymbol{q}_k, \boldsymbol{x}_{k+1} - \boldsymbol{x}_k \rangle - \beta \langle \boldsymbol{r}_{k+1} - \boldsymbol{r}_k, \boldsymbol{y}_{k+1} - \boldsymbol{y}_k \rangle \\
&\leq \frac{1}{2\beta} \left( \|\boldsymbol{\lambda}_k - \boldsymbol{\lambda}_{k-1}\|^2 - \|\boldsymbol{\lambda}_{k+1} - \boldsymbol{\lambda}_k\|^2 \right) + \frac{\beta}{2} \left( \|-\boldsymbol{y}_k + \boldsymbol{y}_{k-1}\|^2 - \|-\boldsymbol{y}_{k+1} + \boldsymbol{y}_k\|^2 \right) \\
&\quad + \beta(\sigma_{k+1} + \sigma_k) \|\boldsymbol{x}_{k+1} - \boldsymbol{x}_k\| + \sqrt{2\beta}(\sqrt{\varepsilon_k} + \sqrt{\varepsilon_{k+1}}) \|\boldsymbol{y}_{k+1} - \boldsymbol{y}_k\| .
\end{aligned}
$$

Equation (51) labels the middle block and (52) labels the final inequality.

Using the monotonicity of $f_k$ and $f_{k-1}$, we deduce:

$$
\langle \hat{\nabla} f_k(\boldsymbol{x}_{k+1} + \boldsymbol{q}_{k+1}), \boldsymbol{x}_k + \boldsymbol{q}_k - (\boldsymbol{x}_{k+1} + \boldsymbol{q}_{k+1}) \rangle + f_k(\boldsymbol{x}_{k+1} + \boldsymbol{q}_{k+1}) \leq f_k(\boldsymbol{x}_k + \boldsymbol{q}_k) ,
$$
$$
\langle \hat{\nabla} f_{k-1}(\boldsymbol{x}_k + \boldsymbol{q}_k), \boldsymbol{x}_{k+1} + \boldsymbol{q}_{k+1} - (\boldsymbol{x}_k + \boldsymbol{q}_k) \rangle + f_{k-1}(\boldsymbol{x}_k + \boldsymbol{q}_k) \leq f_{k-1}(\boldsymbol{x}_{k+1} + \boldsymbol{q}_{k+1}) .
$$

Adding together the above two inequalites and rearranging the terms, we have:

$$
\begin{aligned}
&\langle \hat{\nabla} f_k(\boldsymbol{x}_{k+1} + \boldsymbol{q}_{k+1}) - \hat{\nabla} f_{k-1}(\boldsymbol{x}_k + \boldsymbol{q}_k), \boldsymbol{x}_k + \boldsymbol{q}_k - (\boldsymbol{x}_{k+1} + \boldsymbol{q}_{k+1}) \rangle \\
&\quad + f_k(\boldsymbol{x}_{k+1} + \boldsymbol{q}_{k+1}) - f_{k-1}(\boldsymbol{x}_{k+1} + \boldsymbol{q}_{k+1}) + f_{k-1}(\boldsymbol{x}_k + \boldsymbol{q}_k) - f_k(\boldsymbol{x}_k + \boldsymbol{q}_k) \leq 0 ,
\end{aligned}
$$

which gives:

$$
\begin{aligned}
&\langle \hat{\nabla} f_k(\boldsymbol{x}_{k+1} + \boldsymbol{q}_{k+1}) - \hat{\nabla} f_{k-1}(\boldsymbol{x}_k + \boldsymbol{q}_k), \boldsymbol{x}_{k+1} - \boldsymbol{x}_k \rangle \\
&\geq \langle \hat{\nabla} f_k(\boldsymbol{x}_{k+1} + \boldsymbol{q}_{k+1}) - \hat{\nabla} f_{k-1}(\boldsymbol{x}_k + \boldsymbol{q}_k), \boldsymbol{q}_k - \boldsymbol{q}_{k+1} \rangle \\
&\quad + f_k(\boldsymbol{x}_{k+1} + \boldsymbol{q}_{k+1}) - f_{k-1}(\boldsymbol{x}_{k+1} + \boldsymbol{q}_{k+1}) + f_{k-1}(\boldsymbol{x}_k + \boldsymbol{q}_k) - f_k(\boldsymbol{x}_k + \boldsymbol{q}_k) \\
&= \langle \hat{\nabla} f_k(\boldsymbol{x}_{k+1} + \boldsymbol{q}_{k+1}) - \hat{\nabla} f_{k-1}(\boldsymbol{x}_k + \boldsymbol{q}_k), \boldsymbol{q}_k - \boldsymbol{q}_{k+1} \rangle \\
&\quad + \langle F(\boldsymbol{x}_{k+1}) - F(\boldsymbol{x}_k), \boldsymbol{x}_{k+1} + \boldsymbol{q}_{k+1} - \boldsymbol{x}_k - \boldsymbol{q}_k \rangle \\
&\geq \langle -\boldsymbol{\lambda}_k - \beta(\boldsymbol{x}_{k+1} - \boldsymbol{y}_k) - \beta \boldsymbol{q}_{k+1} - (-\boldsymbol{\lambda}_{k-1} - \beta(\boldsymbol{x}_k - \boldsymbol{y}_{k-1}) - \beta \boldsymbol{q}_k), \boldsymbol{q}_k - \boldsymbol{q}_{k+1} \rangle \\
&\quad + \langle F(\boldsymbol{x}_{k+1}) - F(\boldsymbol{x}_k), \boldsymbol{x}_{k+1} + \boldsymbol{q}_{k+1} - \boldsymbol{x}_k - \boldsymbol{q}_k \rangle \\
&\geq \langle \boldsymbol{\lambda}_{k-1} - \boldsymbol{\lambda}_k - \beta(\boldsymbol{x}_{k+1} - \boldsymbol{y}_k) + \beta(\boldsymbol{x}_k - \boldsymbol{y}_{k-1}), \boldsymbol{q}_k - \boldsymbol{q}_{k+1} \rangle \\
&\quad + \langle F(\boldsymbol{x}_{k+1}) - F(\boldsymbol{x}_k), \boldsymbol{x}_{k+1} + \boldsymbol{q}_{k+1} - \boldsymbol{x}_k - \boldsymbol{q}_k \rangle \\
&\geq -(\sigma_{k+1} + \sigma_k) \left( \|\boldsymbol{\lambda}_{k-1} - \boldsymbol{\lambda}_k - \beta(\boldsymbol{x}_{k+1} - \boldsymbol{y}_k) + \beta(\boldsymbol{x}_k - \boldsymbol{y}_{k-1})\| + \|F(\boldsymbol{x}_{k+1}) - F(\boldsymbol{x}_k)\| \right) ,
\end{aligned}
$$
(53)

where the second inequality uses (noisy-$\hat{\nabla} f_k$), the penultimate inequality uses the nonnegativity of $\langle \boldsymbol{q}_k - \boldsymbol{q}_{k+1}, \boldsymbol{q}_k - \boldsymbol{q}_{k+1} \rangle$, and the last inequality follows from the monotonicity of $F$, the Cauchy-Schwarz inequality and the fact that $\|\boldsymbol{q}_k\| \leq \sigma_k$.

Note that by (LB.16-3) we have:

$$
\begin{aligned}
\boldsymbol{\lambda}_{k-1} - \boldsymbol{\lambda}_k &- \beta(\boldsymbol{x}_{k+1} - \boldsymbol{y}_k) + \beta(\boldsymbol{x}_k - \boldsymbol{y}_{k-1}) \\
&= \beta(\boldsymbol{y}_k - \boldsymbol{x}_k - (\boldsymbol{x}_{k+1} - \boldsymbol{y}_k) + (\boldsymbol{x}_k - \boldsymbol{y}_{k-1})) \\
&= \beta(2\boldsymbol{y}_k - \boldsymbol{x}_{k+1} - \boldsymbol{y}_{k-1}) \\
&= \beta((\boldsymbol{y}_k - \boldsymbol{y}_{k-1}) - (\boldsymbol{y}_{k+1} - \boldsymbol{y}_k) - (\boldsymbol{x}_{k+1} - \boldsymbol{y}_{k+1})) \\
&= \beta((\boldsymbol{y}_k - \boldsymbol{y}_{k-1}) - (\boldsymbol{y}_{k+1} - \boldsymbol{y}_k)) - (\boldsymbol{\lambda}_{k+1} - \boldsymbol{\lambda}_k),
\end{aligned} \tag{54}
$$

$$
\boldsymbol{x}_{k+1} - \boldsymbol{x}_k = \boldsymbol{x}_{k+1} - \boldsymbol{y}_{k+1} + \boldsymbol{y}_{k+1} - \boldsymbol{y}_k + \boldsymbol{y}_k - \boldsymbol{x}_k = \frac{1}{\beta}(\boldsymbol{\lambda}_{k+1} - \boldsymbol{\lambda}_k) + \boldsymbol{y}_{k+1} - \boldsymbol{y}_k + \frac{1}{\beta}(\boldsymbol{\lambda}_k - \boldsymbol{\lambda}_{k-1}). \tag{55}
$$

Using (53),(54), (55) and the L-smoothness property of $F$, we get:

$$
\begin{aligned}
\langle \hat{\nabla} f_k &(\boldsymbol{x}_{k+1} + \boldsymbol{q}_{k+1}) - \hat{\nabla} f_{k-1}(\boldsymbol{x}_k + \boldsymbol{q}_k), \boldsymbol{x}_{k+1} - \boldsymbol{x}_k \rangle \\
&\geq -(\sigma_{k+1} + \sigma_k)(\beta \|\boldsymbol{y}_k - \boldsymbol{y}_{k-1}\| + \beta \|\boldsymbol{y}_{k+1} - \boldsymbol{y}_k\| + \|\boldsymbol{\lambda}_{k+1} - \boldsymbol{\lambda}_k\| + L\|\boldsymbol{x}_{k+1} - \boldsymbol{x}_k\|) \\
&\geq -(\sigma_{k+1} + \sigma_k)\Big(\beta \|\boldsymbol{y}_k - \boldsymbol{y}_{k-1}\| + (\beta + L)\|\boldsymbol{y}_{k+1} - \boldsymbol{y}_k\| \\
&\quad + \left(1 + \frac{L}{\beta}\right)\|\boldsymbol{\lambda}_{k+1} - \boldsymbol{\lambda}_k\| + \frac{L}{\beta}\|\boldsymbol{\lambda}_k - \boldsymbol{\lambda}_{k-1}\|\Big).
\end{aligned}
$$

Combining the above inequality with (34) and (52), and using (44) and (46), it follows that:

$$
\begin{aligned}
\frac{1}{2\beta}&\|\boldsymbol{\lambda}_{k+1} - \boldsymbol{\lambda}_k\|^2 + \frac{\beta}{2}\|-\boldsymbol{y}_{k+1} + \boldsymbol{y}_k\|^2 \\
\leq& \frac{1}{2\beta}\|\boldsymbol{\lambda}_k - \boldsymbol{\lambda}_{k-1}\|^2 + \frac{\beta}{2}\|-\boldsymbol{y}_k + \boldsymbol{y}_{k-1}\|^2 + \beta(\sigma_{k+1} + \sigma_k)\|\boldsymbol{x}_{k+1} - \boldsymbol{x}_k\| \\
&+ \sqrt{2\beta}(\sqrt{\varepsilon_k} + \sqrt{\varepsilon_{k+1}})\|\boldsymbol{y}_{k+1} - \boldsymbol{y}_k\| \\
&+ (\sigma_{k+1} + \sigma_k)\Big(\beta\|\boldsymbol{y}_k - \boldsymbol{y}_{k-1}\| + (\beta + L)\|\boldsymbol{y}_{k+1} - \boldsymbol{y}_k\| + \left(1 + \frac{L}{\beta}\right)\|\boldsymbol{\lambda}_{k+1} - \boldsymbol{\lambda}_k\| \\
&\qquad\qquad + \frac{L}{\beta}\|\boldsymbol{\lambda}_k - \boldsymbol{\lambda}_{k-1}\|\Big) \\
&+ \varepsilon_k + \varepsilon_{k+1} \\
\leq& \frac{1}{2\beta}\|\boldsymbol{\lambda}_k - \boldsymbol{\lambda}_{k-1}\|^2 + \frac{\beta}{2}\|-\boldsymbol{y}_k + \boldsymbol{y}_{k-1}\|^2 \\
&+ (\sigma_{k+1} + \sigma_k)\Big(\beta\|\boldsymbol{y}_k - \boldsymbol{y}_{k-1}\| + (2\beta + L)\|\boldsymbol{y}_{k+1} - \boldsymbol{y}_k\| + \left(2 + \frac{L}{\beta}\right)\|\boldsymbol{\lambda}_{k+1} - \boldsymbol{\lambda}_k\| \\
&\qquad\qquad + \left(\frac{L}{\beta} + 1\right)\|\boldsymbol{\lambda}_k - \boldsymbol{\lambda}_{k-1}\|\Big) \\
&+ \sqrt{2\beta}(\sqrt{\varepsilon_k} + \sqrt{\varepsilon_{k+1}})\|\boldsymbol{y}_{k+1} - \boldsymbol{y}_k\| + \varepsilon_k + \varepsilon_{k+1}.
\end{aligned}
$$

$\square$

**Lemma B.22.** *If* $\lim_{K\to+\infty} \frac{1}{\sqrt{K}} \sum_{k=1}^{K+1} k(\sigma_k + \sqrt{\varepsilon_k}) < +\infty$, *then we have:*

$$\sum_{k=1}^{\infty} \sigma_k + \sqrt{\varepsilon_k} < +\infty \,, \tag{56}$$

$$\sum_{k=1}^{\infty} k\varepsilon_k < +\infty \,. \tag{57}$$

$$\sigma_K + \sqrt{\varepsilon_k} \le \mathcal{O}\left(\frac{1}{\sqrt{K}}\right) \,. \tag{58}$$

*Proof.* Let $T_K \triangleq \lim_{K\to+\infty} \frac{1}{\sqrt{K}} \sum_{k=1}^{K+1} k(\sigma_k + \sqrt{\varepsilon_k})$. If $\lim_{K\to+\infty} T_K < +\infty$, then by Cauchy's convergence test, $\forall p \in \mathbb{N}_+, T_{K+p} - T_K \to 0, \ K \to +\infty$.

Note that

$$T_{K+p} - T_K = \frac{1}{\sqrt{K+p}} \sum_{k=K+2}^{K+p+1} k(\sigma + \sqrt{\varepsilon_k}) + \left(\frac{1}{\sqrt{K+p}} - \frac{1}{\sqrt{K}}\right) \sum_{k=1}^{K+1} k(\sigma + \sqrt{\varepsilon_k})$$

$$= \frac{1}{\sqrt{K+p}} \sum_{k=K+2}^{K+p+1} k(\sigma + \sqrt{\varepsilon_k}) - \frac{p}{\sqrt{K+p}\sqrt{K}\left(\sqrt{K+p} + \sqrt{K}\right)} \sum_{k=1}^{K+1} k(\sigma + \sqrt{\varepsilon_k}) \,,$$

where the second term

$$\frac{p}{\sqrt{K+p}\sqrt{K}\left(\sqrt{K+p} + \sqrt{K}\right)} \sum_{k=1}^{K+1} k(\sigma + \sqrt{\varepsilon_k})$$

$$\le \frac{1}{\sqrt{K}} \sum_{k=1}^{K+1} k(\sigma + \sqrt{\varepsilon_k}) \to 0, \ K \to +\infty, \ \forall p \in \mathbb{N}_+ \,. \tag{59}$$

Thus for any $p \in \mathbb{N}_+$, we have

$$\frac{1}{\sqrt{K+p}} \sum_{k=K+2}^{K+p+1} k(\sigma + \sqrt{\varepsilon_k}) \to 0, \ K \to +\infty \,. \tag{60}$$

From which we deduce that for any $p \in \mathbb{N}_+$,

$$\sum_{K+2}^{K+p+1} (\sigma + \sqrt{\varepsilon_k}) \le \frac{\sqrt{K+p}}{K+2} \cdot \frac{K+2}{\sqrt{K+p}} \sum_{K+2}^{K+p+1} (\sigma + \sqrt{\varepsilon_k})$$

$$\le \frac{\sqrt{K+p}}{K+2} \cdot \frac{1}{\sqrt{K+p}} \sum_{K+2}^{K+p+1} k(\sigma + \sqrt{\varepsilon_k}) \to 0, \ \forall K \to +\infty \,. \tag{61}$$

Again by Cauchy's convergence test, we have

$$\sum_{k=1}^{\infty} \sigma_k + \sqrt{\varepsilon_k} < +\infty \,,$$

which is (56).

Note that $\lim_{K\to\infty} T_K = T_0 + \sum_{k=0}^{\infty} T_{k+1} - T_k$. And

$$T_{k+1} - T_k = \mathcal{O}\left(\sqrt{K}(\sigma_k + \sqrt{\varepsilon_k})\right) \ge \mathcal{O}(k\varepsilon_k) \,.$$

Thus by the comparison test, we have

$$\sum_{k=1}^{\infty} k\varepsilon_k < +\infty \,,$$

$$\sigma_K + \sqrt{\varepsilon_k} \leq \mathcal{O}\left(\frac{1}{\sqrt{K}}\right),$$

which gives (57), (58). $\qquad\square$

**Lemma B.23.** *Assume that $F$ is monotone on $\mathcal{C}_=$, and $\lim_{K\to+\infty} \frac{1}{\sqrt{K}} \sum_{k=1}^{K+1} k(\sigma_k + \sqrt{\varepsilon_k}) < +\infty$, then for the inexact ACVI—Alg. 1, we have:*

$$f_K(\boldsymbol{x}_{K+1} + \boldsymbol{q}_{K+1}) + g(\boldsymbol{y}_{K+1}) - f_K(\boldsymbol{x}_K^\mu) - g(\boldsymbol{y}_K^\mu)$$
$$\leq (\|\boldsymbol{\lambda}^\mu\| + 4A + \beta D)\frac{E^\mu}{\beta\sqrt{K}} + (3A^\mu + \beta D)\sigma_{k+1} + \sqrt{2\beta}\left(\frac{A^\mu}{\beta} + D\right)\sqrt{\varepsilon_{k+1}} + \varepsilon_{k+1},$$
$$\text{(LB.23-1)}$$

$$and \qquad \|\boldsymbol{x}_{K+1} - \boldsymbol{y}_{K+1}\| \leq \frac{E^\mu}{\beta\sqrt{K}}, \qquad\qquad \text{(LB.23-2)}$$

*where $A^\mu$ is defined in Theorem B.20.*

*Proof of Lemma B.23.* First, we define: $\Delta^\mu \triangleq \frac{1}{\beta}\|\boldsymbol{\lambda}_0 - \boldsymbol{\lambda}^\mu\|^2 + \beta\|\boldsymbol{y}_0 - \boldsymbol{y}^\mu\|^2$.

Summing (36) over $k = 0, 1, \ldots, K$, we have:

$$\sum_{i=0}^{K}\left(\frac{1}{2\beta}\|\boldsymbol{\lambda}_{k+1} - \boldsymbol{\lambda}_k\|^2 + \frac{\beta}{2}\|-\boldsymbol{y}_{k+1} + \boldsymbol{y}_k\|^2\right)$$

$$\leq \frac{1}{2\beta}\|\boldsymbol{\lambda}_0 - \boldsymbol{\lambda}^\mu\|^2 + \frac{\beta}{2}\|-\boldsymbol{y}_0 + \boldsymbol{y}^\mu\|^2$$

$$+ \sum_{i=0}^{K}\sqrt{2\beta}(\sqrt{\varepsilon_{k+1}} + \sqrt{\varepsilon_k})\|\boldsymbol{y}_{k+1} - \boldsymbol{y}_k\| + \sum_{i=0}^{K}\sqrt{2\beta\varepsilon_{k+1}}\|\boldsymbol{y}_{k+1} - \boldsymbol{y}^\mu\|$$

$$+ \sum_{k=0}^{K}\varepsilon_k + 2\sum_{k=0}^{K}\varepsilon_{k+1} + \sum_{k=0}^{K}\sigma_{k+1}(\|\boldsymbol{\lambda}_k - \boldsymbol{\lambda}^\mu\| + \beta\|\boldsymbol{x}_{k+1} - \boldsymbol{y}_k\| + \beta\|\boldsymbol{x}_{k+1} - \boldsymbol{x}^\mu\|)$$

$$\leq \frac{1}{2\beta}\|\boldsymbol{\lambda}_0 - \boldsymbol{\lambda}^\mu\|^2 + \frac{\beta}{2}\|-\boldsymbol{y}_0 + \boldsymbol{y}^\mu\|^2$$

$$+ 2\sum_{k=0}^{K}\sqrt{\frac{2}{\beta}}A^\mu(\sqrt{\varepsilon_{k+1}} + \sqrt{\varepsilon_k}) + \sum_{k=0}^{K}\sqrt{\frac{2}{\beta}}A^\mu\sqrt{\varepsilon_{k+1}}$$

$$+ \sum_{k=0}^{K}\varepsilon_k + 2\sum_{k=0}^{K}\varepsilon_{k+1} + \sum_{k=0}^{K}\sigma_{k+1}\left(A + \beta \cdot \frac{4}{\beta}A^\mu + \beta \cdot \frac{3}{\beta}A^\mu\right)$$

$$\leq \Delta^\mu + 5\sqrt{\frac{2}{\beta}}A^\mu\sum_{k=1}^{K+1}\sqrt{\varepsilon_k} + 8A^\mu\sum_{i=1}^{K+1}\sigma_i + 3\sum_{k=1}^{K+1}\varepsilon_k, \qquad (62)$$

where the penultimate inequality follows from (41), (44), (45) and (46), and $A^\mu$ is defined in Theorem B.20.

Recall that Lemma B.21 gives:

$$\frac{1}{2\beta}\|\boldsymbol{\lambda}_{k+1} - \boldsymbol{\lambda}_k\|^2 + \frac{\beta}{2}\|-\boldsymbol{y}_{k+1} + \boldsymbol{y}_k\|^2$$

$$\leq \frac{1}{2\beta}\|\boldsymbol{\lambda}_k - \boldsymbol{\lambda}_{k-1}\|^2 + \frac{\beta}{2}\|-\boldsymbol{y}_k + \boldsymbol{y}_{k-1}\|^2$$

$$+ (\sigma_{k+1} + \sigma_k)\left(\beta\|\boldsymbol{y}_k - \boldsymbol{y}_{k-1}\| + (2\beta + L)\|\boldsymbol{y}_{k+1} - \boldsymbol{y}_k\| + \left(2 + \frac{L}{\beta}\right)\|\boldsymbol{\lambda}_{k+1} - \boldsymbol{\lambda}_k\|\right.$$

$$\left. + \left(\frac{L}{\beta} + 1\right)\|\boldsymbol{\lambda}_k - \boldsymbol{\lambda}_{k-1}\|\right)$$

$$+ \sqrt{2\beta}(\sqrt{\varepsilon_k} + \sqrt{\varepsilon_{k+1}})\|\boldsymbol{y}_{k+1} - \boldsymbol{y}_k\| + \varepsilon_k + \varepsilon_{k+1}.$$

Let:

$$\delta_{k+1} \triangleq (\sigma_{k+1} + \sigma_k) \left( \beta \left\| \boldsymbol{y}_k - \boldsymbol{y}_{k-1} \right\| + (2\beta + L) \left\| \boldsymbol{y}_{k+1} - \boldsymbol{y}_k \right\| + \left( 2 + \frac{L}{\beta} \right) \left\| \boldsymbol{\lambda}_{k+1} - \boldsymbol{\lambda}_k \right\| \right.$$

$$\left. + \left( \frac{L}{\beta} + 1 \right) \left\| \boldsymbol{\lambda}_k - \boldsymbol{\lambda}_{k-1} \right\| \right) \qquad (\delta)$$

$$+ \sqrt{2\beta}(\sqrt{\varepsilon_k} + \sqrt{\varepsilon_{k+1}}) \left\| \boldsymbol{y}_{k+1} - \boldsymbol{y}_k \right\| + \varepsilon_k + \varepsilon_{k+1} \,.$$

Then the above inequality could be rewritten as:

$$\frac{1}{2\beta} \left\| \boldsymbol{\lambda}_{k+1} - \boldsymbol{\lambda}_k \right\|^2 + \frac{\beta}{2} \left\| -\boldsymbol{y}_{k+1} + \boldsymbol{y}_k \right\|^2$$

$$\leq \frac{1}{2\beta} \left\| \boldsymbol{\lambda}_k - \boldsymbol{\lambda}_{k-1} \right\|^2 + \frac{\beta}{2} \left\| -\boldsymbol{y}_k + \boldsymbol{y}_{k-1} \right\|^2 + \delta_{k+1} \,,$$

which gives:

$$\frac{1}{2\beta} \left\| \boldsymbol{\lambda}_{K+1} - \boldsymbol{\lambda}_K \right\|^2 + \frac{\beta}{2} \left\| -\boldsymbol{y}_{K+1} + \boldsymbol{y}_K \right\|^2$$

$$\leq \frac{1}{2\beta} \left\| \boldsymbol{\lambda}_k - \boldsymbol{\lambda}_{k-1} \right\|^2 + \frac{\beta}{2} \left\| -\boldsymbol{y}_k + \boldsymbol{y}_{k-1} \right\|^2 + \sum_{i=k}^{K} \delta_{i+1} \,. \qquad (63)$$

Combining (63) with (62), we obtain:

$$K \left( \frac{1}{2\beta} \left\| \boldsymbol{\lambda}_{K+1} - \boldsymbol{\lambda}_K \right\|^2 + \frac{\beta}{2} \left\| -\boldsymbol{y}_{K+1} + \boldsymbol{y}_K \right\|^2 \right)$$

$$\leq \sum_{i=0}^{K} \left( \frac{1}{2\beta} \left\| \boldsymbol{\lambda}_{k+1} - \boldsymbol{\lambda}_k \right\|^2 + \frac{\beta}{2} \left\| -\boldsymbol{y}_{k+1} + \boldsymbol{y}_k \right\|^2 \right) + \sum_{i=0}^{K-1} \sum_{j=k+1}^{K} \delta_{j+1}$$

$$\leq \Delta^{\mu} + 5 \sqrt{\frac{2}{\beta}} A \sum_{k=1}^{K+1} \sqrt{\varepsilon_k} + 8A \sum_{k=1}^{K+1} \sigma_i + 3 \sum_{k=1}^{K+1} \varepsilon_k + \sum_{k=1}^{K} k \delta_{k+1} \,. \qquad (64)$$

We define:

$$a_{k+1} \triangleq (\sigma_{k+1} + \sigma_k) \left( 1 + \frac{L}{\beta} \right) \,, \qquad (a)$$

$$b_{k+1} \triangleq (\sigma_{k+1} + \sigma_k) \left( 2 + \frac{L}{\beta} \right) + \sqrt{\frac{2}{\beta}}(\sqrt{\varepsilon_{k+1}} + \sqrt{\varepsilon_k}) \,, \qquad (b)$$

$$u'_{k+1} \triangleq \left\| \boldsymbol{\lambda}_{k+1} - \boldsymbol{\lambda}_k \right\| + \beta \left\| \boldsymbol{y}_{k+1} - \boldsymbol{y}_k \right\| \,. \qquad (u')$$

Note that:

$$\delta_{k+1} \leq \varepsilon_k + \varepsilon_{k+1} + \underbrace{(\sigma_{k+1} + \sigma_k) \left( 1 + \frac{L}{\beta} \right)}_{a_{k+1}} \underbrace{(\left\| \boldsymbol{\lambda}_k - \boldsymbol{\lambda}_{k+1} \right\| + \beta \left\| \boldsymbol{y}_k - \boldsymbol{y}_{k-1} \right\|)}_{u'_k}$$

$$+ \underbrace{\left( (\sigma_{k+1} + \sigma_k) \left( 2 + \frac{L}{\beta} \right) + \sqrt{\frac{2}{\beta}}(\sqrt{\varepsilon_{k+1}} + \sqrt{\varepsilon_k}) \right)}_{b_{k+1}} \underbrace{(\left\| \boldsymbol{\lambda}_{k+1} - \boldsymbol{\lambda}_k \right\| + \beta \left\| \boldsymbol{y}_{k+1} - \boldsymbol{y}_k \right\|)}_{u'_{k+1}} \,,$$

from which we deduce that:

$$\sum_{k=1}^{K} k\delta_{k+1} \leq \sum_{k=1}^{K} (ka_{k+1}u'_k + (k+1)b_{k+1}u'_{k+1}) + \sum_{k=1}^{K} k(\varepsilon_k + \varepsilon_{k+1}) \tag{65}$$

$$\leq \sum_{k=1}^{K+1} (a_{k+1} + b_k)ku'_k + 2\sum_{k=1}^{K+1} k\varepsilon_k \tag{66}$$

$$= \sum_{k=1}^{K+1} \underbrace{\left((\sigma_{k+1} + \sigma_k)\left(1 + \frac{L}{\beta}\right) + (\sigma_{k-1} + \sigma_k)\left(2 + \frac{L}{\beta}\right) + \sqrt{\frac{2}{\beta}}(\sqrt{\varepsilon_{k-1}} + \sqrt{\varepsilon_k})\right)}_{c_k} ku'_k$$

$$+ 2\sum_{k=1}^{K+1} k\varepsilon_k \, , \tag{67}$$

where we define

$$c_k \triangleq \left((\sigma_{k+1} + \sigma_k)\left(1 + \frac{L}{\beta}\right) + (\sigma_{k-1} + \sigma_k)\left(2 + \frac{L}{\beta}\right) + \sqrt{\frac{2}{\beta}}(\sqrt{\varepsilon_{k-1}} + \sqrt{\varepsilon_k})\right). \tag{c}$$

Note that by Cauchy-Schwarz inequality, we have:

$$\frac{K}{4\beta}u'^2_{k+1} = \frac{K}{4\beta}\left(\|\boldsymbol{\lambda}_{k+1} - \boldsymbol{\lambda}_k\| + \beta\|\boldsymbol{y}_{k+1} - \boldsymbol{y}_k\|\right)^2$$

$$\leq \frac{K}{2\beta}\left(\|\boldsymbol{\lambda}_{K+1} - \boldsymbol{\lambda}_K\|^2 + \beta^2\|-\boldsymbol{y}_{K+1} + \boldsymbol{y}_K\|^2\right).$$

Combining this inequality with (64), (67), and letting:

$$B_{k+1} \triangleq \Delta^{\mu} + 5\sqrt{\frac{2}{\beta}}A^{\mu}\sum_{k=1}^{K+1}\sqrt{\varepsilon_k} + 8A^{\mu}\sum_{k=1}^{K+1}\sigma_i + 3\sum_{k=1}^{K+1}\varepsilon_k \, , \tag{B}$$

gives:

$$u'^2_{k+1} \leq \frac{4\beta}{K}\left(B_{k+1} + 2\sum_{k=1}^{K+1} k\varepsilon_k\right) + \frac{4\beta}{K}\sum_{k=1}^{K+1} kc_k u'_k \, .$$

Using Lemma B.12, we obtain:

$$u'_{k+1} \leq \frac{1}{\sqrt{K}}\underbrace{\left(\frac{2\beta}{\sqrt{K}}\sum_{k=1}^{K+1} kc_k + \left(4\beta\left(B_{k+1} + 2\sum_{k=1}^{K+1} k\varepsilon_k\right) + \left(\frac{2\beta}{\sqrt{K}\sum_{k=1}^{K+1} kc_k}\right)^2\right)^{\frac{1}{2}}\right)}_{E_{k+1}}. \tag{68}$$

Using the assumption that $\lim_{K\to+\infty}\frac{1}{\sqrt{K}}\sum_{k=1}^{K+1} k(\sigma_k + \sqrt{\varepsilon_k}) < +\infty$ and (56) in Lamma B.22, we have $B_{k+1}$ is bounded; using (57), we know that $E_{k+1}$ in the RHS of (68) is bounded.

Let $E^{\mu} = \lim_{k\to\infty} E_k$, then by (68) we have

$$\beta\|\boldsymbol{x}_{K+1} - \boldsymbol{y}_{K+1}\| = \|\boldsymbol{\lambda}_{K+1} - \boldsymbol{\lambda}_K\| \leq \frac{E^{\mu}}{\sqrt{K}} \, , \tag{69}$$

$$\|-\boldsymbol{y}_{K+1} + \boldsymbol{y}_K\| \leq \frac{E^{\mu}}{\beta\sqrt{K}} \, . \tag{70}$$

On the other hand, (44), (45) and (46) gives:

$$\|\boldsymbol{x}_k - \boldsymbol{x}_k^{\mu}\| \leq \|\boldsymbol{x}_k - \boldsymbol{x}^{\mu}\| + \|\boldsymbol{x}^{\mu} - \boldsymbol{x}_k^{\mu}\| \leq \frac{3}{\beta}A^{\mu} + D \, ,$$

$$\|\boldsymbol{y}_k - \boldsymbol{y}_k^{\mu}\| \leq \|\boldsymbol{y}_k - \boldsymbol{y}^{\mu}\| + \|\boldsymbol{y}^{\mu} - \boldsymbol{y}_k^{\mu}\| \leq \frac{1}{\beta}A^{\mu} + D \, ,$$

$$\|\boldsymbol{\lambda}_{k+1}\| \leq \|\boldsymbol{\lambda}_{k+1} - \boldsymbol{\lambda}^{\mu}\| + \|\boldsymbol{\lambda}^{\mu}\| \leq A^{\mu} + \|\boldsymbol{\lambda}^{\mu}\| \, .$$

Plugging these into (TB.20-$f_k$-UB) yields (LB.23-1).

$\square$

### B.2.3 Analogous intermediate results for the extended log barrier

Recall from § 3.2 that we defined the following barrier functions,

$$\wp_1(\boldsymbol{z}, \mu) \triangleq -\mu \, \log(-\boldsymbol{z}) \tag{$\wp_1$}$$

$$\wp_2(\boldsymbol{z}, \mu) \triangleq \begin{cases} -\mu \log(-\boldsymbol{z}), & \boldsymbol{z} \le -e^{-\frac{c}{\mu}} \\ \mu e^{\frac{c}{\mu}} \boldsymbol{z} + \mu + c, & \text{otherwise} \end{cases} \tag{$\wp_2$}$$

where $c$ in ($\wp_2$) is a fixed constant. For convenience, we also define herein:

$$\tilde{g}^{(t)}(\boldsymbol{y}) \triangleq \sum_{i=1}^{m} \wp_2\big(\varphi_i(\boldsymbol{y}), \mu_t\big) . \tag{$\tilde{g}^{(t)}$}$$

In the previous subsections, we focused on the standard barrier function used for IP methods ($\wp_1$). In this subsection, we first show that when we use barrier map ($\wp_2$) in Alg. 1, where constant $c$ in ($\wp_2$) is properly chosen, $\boldsymbol{y}_{k+1}^{(t)}$—the solution to the minimization problem in line 9 of Alg. 1 is the same when we use standard barrier map ($\wp_1$). Thus, all the above results hold if we substitute $g^{(t)}$ by $\tilde{g}^{(t)}$.

**Proposition B.24** (Equivalent solutions of the $\boldsymbol{y}$-subproblems with $\wp_1$ and with $\wp_2$). *For any fixed $t \in \{0, \ldots, T-1\}$ and $k \in \{0, \ldots, K-1\}$, let $\tau_i \triangleq \min_{\boldsymbol{y}} \sum_{j=1, j\neq i}^{m} \wp_2\big(\varphi_j(\boldsymbol{y}), \mu_t\big) + \frac{\beta}{2} \left\| \boldsymbol{y} - \boldsymbol{x}_{k+1}^{(t)} - \frac{1}{\beta} \boldsymbol{\lambda}_k^{(t)} \right\|^2$, $\tau \ge -\min_{1 \le i \le m}\{\tau_i\}$, and $c_k^t \triangleq \psi_k^t(\boldsymbol{y}_k^{(t)}) + \tau$. Further, define:*

$$\psi_k^t(\boldsymbol{y}) \triangleq \sum_{i=1}^{m} \wp_1\big(\varphi_i(\boldsymbol{y}), \mu_t\big) + \frac{\beta}{2} \left\| \boldsymbol{y} - \boldsymbol{x}_{k+1}^{(t)} - \frac{1}{\beta} \boldsymbol{\lambda}_k^{(t)} \right\|^2 , \tag{$\psi$}$$

*and*

$$\tilde{\psi}_k^t(\boldsymbol{y}) \triangleq \sum_{i=1}^{m} \wp_2\big(\varphi_i(\boldsymbol{y}), \mu_t\big) + \frac{\beta}{2} \left\| \boldsymbol{y} - \boldsymbol{x}_{k+1}^{(t)} - \frac{1}{\beta} \boldsymbol{\lambda}_k^{(t)} \right\|^2 , \tag{$\tilde{\psi}$}$$

*where we let $c = c_k^t$ in ($\wp_2$). Then, it holds that:*

$$\boldsymbol{y}_{k+1}^{(t)} = \underset{\boldsymbol{y}}{\arg\min}\, \tilde{\psi}_k^t(\boldsymbol{y}) = \underset{\boldsymbol{y}}{\arg\min}\, \psi_k^t(\boldsymbol{y}; c_k^t) . \tag{71}$$

*Proof of Prop. B.24: Equivalent solutions of the $\boldsymbol{y}$-subproblems with $\wp_1$ and with $\wp_2$.* When $c = c_k^t$ in ($\wp_2$), $\forall \boldsymbol{y} \in \mathbb{R}^n$, if $\exists i \in [m]$, s.t. $\varphi_i(\boldsymbol{y}) > -e^{-c_k^t/\mu}$, then we have

$$\wp_2(\varphi_i(\boldsymbol{y}), \mu_t) > c_k^t = \psi_k^t(\boldsymbol{y}_k^{(t)}) + \tau . \tag{72}$$

Note that

$$\wp_2(x, \mu_t) \le \wp_1(x, \mu_t), \ \forall x , \tag{73}$$

thus, we have:

$$\tilde{\psi}_k^t(\boldsymbol{y}') \le \psi_k^t(\boldsymbol{y}'), \ \forall \boldsymbol{y}' . \tag{74}$$

Let $\tilde{\boldsymbol{y}}_{k+1}^{(t)} = \underset{\boldsymbol{y}}{\arg\min}\, \tilde{\psi}_k^t(\boldsymbol{y})$. If $\tau \ge -\min_{1 \le i \le m}\{\tau_i\}$, then (72) and (74) give:

$$\tilde{\psi}_k^t(\boldsymbol{y}) = \sum_{i=1}^{m} \wp_2(\varphi_i(\boldsymbol{y}); c_k^t) + \frac{\beta}{2} \left\| \boldsymbol{y} - \boldsymbol{x}_{k+1}^{(t)} - \frac{1}{\beta} \boldsymbol{\lambda}_k^{(t)} \right\|^2$$
$$> \psi_k^t(\boldsymbol{y}_k^{(t)}) + \tau + \tau_i \ge \psi_k^t(\boldsymbol{y}_k^{(t)}) \ge \tilde{\psi}_k^t(\boldsymbol{y}_k^{(t)}) \ge \tilde{\psi}_k^t(\tilde{\boldsymbol{y}}_{k+1}^{(t)}) , \tag{75}$$

which indicates $\tilde{\boldsymbol{y}}_{k+1}^{(t)}$, the minimum of $\tilde{\psi}_k^t(\boldsymbol{y}_k^{(t)})$, must be in the set $\mathcal{S} \triangleq \{\boldsymbol{x} | \varphi(\boldsymbol{x}) \le -e^{-c_k^t/\mu} \boldsymbol{e}\}$. Note that $\tilde{\psi}_k^t(\cdot) \equiv \psi_k^t(\cdot)$ on $\mathcal{S}$. Therefore, $\tilde{\boldsymbol{y}}_{k+1}^{(t)} = \boldsymbol{y}_{k+1}^{(t)}$. $\square$

The next Proposition shows the smoothness of the objective in line 9 of Alg. 1 when we use the extended log barrier term ($\wp_2$).

**Proposition B.25** (Smoothness of ($\wp_2$)). *Suppose for all* $i \in [m]$, *we have* $\|\nabla\varphi_i(\boldsymbol{y})\| \leq M_i$, $\forall \boldsymbol{y} \in \mathbb{R}^n$, *and* $\varphi_i$ *is* $L_i$-*smooth in* $\mathbb{R}^n$, *then* $\tilde{\psi}_k^t(\cdot)$ *is* $\tilde{L}_{c_k^t}^{\mu_t}$-*smooth, where* $\tilde{L}_{c_k^t}^{\mu_t} = \sum_{i=1}^m \left( \mu_t e^{c_k^t/\mu_t} L_i + \mu_t e^{2c_k^t/\mu_t} M_i \right) + \beta$.

*Proof of Prop. B.25: Smoothness of* ($\wp_2$). Note that for any $x, c \in \mathbb{R}$ and $\mu > 0$, we have $0 \leq \wp_2'(x,\mu) \leq \mu e^{c/\mu}$ and $0 \leq \wp_2''(x,\mu) \leq \mu e^{2c/\mu}$. Thus, we have:

$$
\begin{aligned}
\|\nabla\tilde{\psi}_k^t&(\boldsymbol{y}) - \nabla\tilde{\psi}_k^t(\boldsymbol{x})\| \\
&= \|\wp_2'(\varphi_i(\boldsymbol{y}),\mu)\nabla\varphi_i(\boldsymbol{y}) - \wp_2'(\varphi_i(\boldsymbol{x}),\mu)\nabla\varphi_i(\boldsymbol{x})\| \\
&= \|\wp_2'(\varphi_i(\boldsymbol{y}),\mu)(\nabla\varphi_i(\boldsymbol{y}) - \nabla\varphi_i(\boldsymbol{x})) + (\wp_2'(\varphi_i(\boldsymbol{y}),\mu) - \wp_2'(\varphi_i(\boldsymbol{x}),\mu))\nabla\varphi_i(\boldsymbol{x})\| \\
&\leq \mu e^{c/\mu}\|\nabla\varphi_i(\boldsymbol{y}) - \nabla\varphi_i(\boldsymbol{x})\| + M_i|\wp_2'(\varphi_i(\boldsymbol{y}),\mu) - \wp_2'(\varphi_i(\boldsymbol{x}),\mu)| \\
&\leq \left( \mu e^{c/\mu} L_i + \mu e^{2c/\mu} M_i \right) \|\boldsymbol{y} - \boldsymbol{x}\| , \ \forall \boldsymbol{x}, \boldsymbol{y} \in \mathbb{R}^n ,
\end{aligned}
$$

from which we can easily see that the proposition is true. $\qquad\square$

*Remark* B.26. We note the following remarks regarding the above result.

(i) When $t$ is large, $\tau_i$ defined in Prop. B.24 $> 0$ or is very close to 0. Therefore, in order to let $\tau$ satisfy $\tau \geq -\min_{1 \leq i \leq m}\{\tau_i\}$, it suffices to set it to a small positive number.

(ii) $\psi_k^t(\boldsymbol{y}_k^{(t)})$ is bounded and thus $c_k^t$ is bounded. Suppose $c_k^t$ is upper bounded by $c^\star$, i.e., $c_k^t \leq c^\star$, $\forall k, t$. Then $\forall t \in \{0, \ldots, T-1\}$, $\tilde{\psi}_k^t(\cdot)$ is $\tilde{L}_{c^\star}^{\mu_t}$-smooth, and $\beta$-strongly convex and the subproblem in line 9 could be solved by commonly used first-order methods such as gradient descent at a linear rate.

(iii) Alternatively, instead of updating $c_k^t$ for each $k, t$ as it is suggested in line 9 of Alg. 1, for any $t$, we could fix $c_k^t$ to be $\psi_k^t(\boldsymbol{y}') + \tau$, where $\boldsymbol{y}'$ is an arbitrary interior point of $\mathcal{C}_\leq$; see Appendix C for detailed implementation.

### B.2.4 PROVING THEOREM 3.2

We are now ready to prove Theorem 3.2. Here we give a nonasymptotic convergence rate of Algorithm 1.

**Theorem B.27** (Restatement of Theorem 3.2). *Given a continuous operator* $F \colon \mathcal{X} \to \mathbb{R}^n$, *assume that:*

(i) *$F$ is monotone on $\mathcal{C}_=$, as per Def. 2.1;*

(ii) *$F$ is $L$-Lipschitz on $\mathcal{X}$;*

(iii) *$F$ is either strictly monotone on $\mathcal{C}$ or one of $\varphi_i$ is strictly convex.*

*For any fixed* $K \in \mathbb{N}_+$, *let* $(\boldsymbol{x}_K^{(t)}, \boldsymbol{y}_K^{(t)}, \boldsymbol{\lambda}_K^{(t)})$ *denote the last iterate of Algorithm 1. Let* $\wp$ *be* $\wp_1$ *or* $\wp_2$ *with* $c = c_k^t$ *(see Prop. B.24 for the definition of* $c_k^t$). *Run with sufficiently small* $\mu_{-1}$. *Further, suppose:*

$$
\lim_{K\to+\infty} \frac{1}{\sqrt{K}} \sum_{k=1}^{K+1} k(\sigma_k + \sqrt{\varepsilon_k}) < +\infty .
$$

*We define:*

$$
\lambda_i \triangleq 4\beta\big(\sqrt{\tfrac{2}{\beta}}\big(\sqrt{\varepsilon_{i+1}} + 3\sqrt{\varepsilon_i} + \sqrt{\varepsilon_{i-1}}\big) + (2\sigma_i + 3\sigma_{i+1})\big) ,
$$

$$S_{k+1}^{\star} \triangleq 2 \left\| \boldsymbol{\lambda}_0 - \boldsymbol{\lambda}^{\star} \right\|^2 + 2\beta^2 \left\| \boldsymbol{y}_0 - \boldsymbol{y}^{\star} \right\|^2 + 4\beta \big( \sqrt{2\beta}\sqrt{\varepsilon_1} + \beta\sigma_1 \big) \left\| \boldsymbol{y}_0 - \boldsymbol{y}^{\star} \right\|$$

$$+ 12\beta\sigma_1 \left\| \boldsymbol{\lambda}_0 - \boldsymbol{\lambda}^{\star} \right\| + 12\beta \sum_{i=1}^{k+1} \varepsilon_i \,,$$

$$A_{k+1}^{\star} \triangleq \frac{1}{2} \sum_{i=1}^{k+1} \lambda_i + \left( S_{k+1}^{\star} + \left( \frac{1}{2} \sum_{i=1}^{k+1} \lambda_i \right)^2 \right)^{1/2} ,$$

*and*

$$A \triangleq \lim_{k \to +\infty} A_k^{\star} < +\infty \,.$$

*We define*

$$c_k \triangleq \left( (\sigma_{k+1} + \sigma_k) \left( 1 + \frac{L}{\beta} \right) + (\sigma_{k-1} + \sigma_k) \left( 2 + \frac{L}{\beta} \right) + \sqrt{\frac{2}{\beta}} (\sqrt{\varepsilon_{k-1}} + \sqrt{\varepsilon_k}) \right) ,$$

$$\Delta \triangleq \frac{1}{\beta} \| \boldsymbol{\lambda}_0 - \boldsymbol{\lambda}^{\star} \|^2 + \beta \| \boldsymbol{y}_0 - \boldsymbol{y}^{\star} \|^2 \,,$$

$$B_{k+1}^{\star} \triangleq \Delta + 5\sqrt{\frac{2}{\beta}} A \sum_{k=1}^{K+1} \sqrt{\varepsilon_k} + 8A \sum_{k=1}^{K+1} \sigma_i + 3 \sum_{k=1}^{K+1} \varepsilon_k \,,$$

$$E_{k+1}^{\star} \triangleq \frac{2\beta}{\sqrt{K}} \sum_{k=1}^{K+1} k c_k + \left( 4\beta \left( B_{k+1}^{\star} + 2 \sum_{k=1}^{K+1} k\varepsilon_k \right) + \left( \frac{2\beta}{\sqrt{K} \sum_{k=1}^{K+1} k c_k} \right)^2 \right)^{\frac{1}{2}} ,$$

*and*

$$E = \lim_{k \to \infty} E_k^{\star} \,.$$

*Then, we have:*

$$\mathcal{G}(\boldsymbol{x}_{K+1}, \mathcal{C}) \leq \left( 2 \left\| \boldsymbol{\lambda}^{\star} \right\| + 5A + \beta D + 1 + M \right) \frac{E}{\beta\sqrt{K}} + (4A + \beta D + M)\sigma_{k+1}$$

$$+ \sqrt{2\beta} \left( \frac{2A}{\beta} + D \right) \sqrt{\varepsilon_{k+1}} + \varepsilon_{k+1}$$

$$= \mathcal{O}\left( \frac{1}{\sqrt{K}} \right) .$$

*and*

$$\| \boldsymbol{x}_{K+1} - \boldsymbol{y}_{K+1} \| \leq \frac{2E}{\beta\sqrt{K}},$$

*where* $\Delta \triangleq \frac{1}{\beta} \| \boldsymbol{\lambda}_0 - \boldsymbol{\lambda}^{\star} \|^2 + \beta \| \boldsymbol{y}_0 - \boldsymbol{y}^{\star} \|^2$ *and* $D \triangleq \sup_{\boldsymbol{x}, \boldsymbol{y} \in \mathcal{C}} \| \boldsymbol{x} - \boldsymbol{y} \|$, *and* $M \triangleq \sup_{\boldsymbol{x} \in \mathcal{C}} \| F(\boldsymbol{x}) \|$.

*Proof of Theorem B.27.* Note that

$$(f_k\text{-Pr-2}) \Leftrightarrow \min_{\boldsymbol{x} \in \mathcal{C}} \langle F(\boldsymbol{x}_{k+1}), \boldsymbol{x} \rangle$$

$$\Leftrightarrow \max_{\boldsymbol{x} \in \mathcal{C}} \langle F(\boldsymbol{x}_{k+1}), \boldsymbol{x}_{k+1} - \boldsymbol{x} \rangle$$

$$\Leftrightarrow \mathcal{G}(\boldsymbol{x}_{k+1}, \mathcal{C}) \,,$$

from which we deduce

$$\mathcal{G}(\boldsymbol{x}_{k+1}, \mathcal{C}) = \langle F(\boldsymbol{x}_{k+1}), \boldsymbol{x}_{k+1} - \boldsymbol{x}_k^\star \rangle, \forall k. \tag{76}$$

For any $K \in \mathbb{N}$, by (Chu, 1998) we know that:

$$\boldsymbol{x}_K^\mu \to \boldsymbol{x}_K^\star,$$

$$g(\boldsymbol{y}_{K+1}) - g(\boldsymbol{y}_K^\mu) \to 0,$$

$$\Delta^\mu \to \frac{1}{\beta}\|\boldsymbol{\lambda}_0 - \boldsymbol{\lambda}^\star\|^2 + \beta\|\boldsymbol{y}_0 - \boldsymbol{y}^\star\|^2 = \Delta, \tag{77}$$

$$A^\mu \to A, \tag{78}$$

$$E^\mu \to E. \tag{79}$$

Thus, there exists $\mu_{-1} > 0$, $s.t. \forall 0 < \mu < \mu_{-1}$,

$$\|\boldsymbol{x}_K^\mu - \boldsymbol{x}_K^\star\| \le \frac{E^\mu}{\beta\sqrt{K}},$$

$$|g(\boldsymbol{y}_{K+1}) - g(\boldsymbol{y}_K^\mu)| \le \frac{E^\mu}{\beta\sqrt{K}}.$$

Combining with Lemma B.23, we have that:

$$\langle F(\boldsymbol{x}_{K+1}), \boldsymbol{x}_{K+1} - \boldsymbol{x}_K^\mu \rangle$$
$$= \langle F(\boldsymbol{x}_{K+1}), \boldsymbol{x}_{K+1} + \boldsymbol{q}_{K+1} - \boldsymbol{x}_K^\mu \rangle - \langle F(\boldsymbol{x}_{K+1}), \boldsymbol{q}_{K+1} \rangle$$
$$= f_K(\boldsymbol{x}_{K+1} + \boldsymbol{q}_{K+1}) - f_K(\boldsymbol{x}_K^\mu) - \langle F(\boldsymbol{x}_{K+1}), \boldsymbol{q}_{K+1} \rangle$$
$$\le (\|\boldsymbol{\lambda}^\mu\| + 4A^\mu + \beta D)\frac{E^\mu}{\beta\sqrt{K}} + (3A^\mu + \beta D)\sigma_{k+1}$$
$$\quad + \sqrt{2\beta}\left(\frac{A^\mu}{\beta} + D\right)\sqrt{\varepsilon_{k+1}} + \varepsilon_{k+1}$$
$$\quad + \|\boldsymbol{q}_{K+1}\|\|F(\boldsymbol{x}_{K+1})\| + g(\boldsymbol{y}_K^\mu) - g(\boldsymbol{y}_{K+1})$$
$$\le (\|\boldsymbol{\lambda}^\mu\| + 4A^\mu + \beta D + 1)\frac{E^\mu}{\beta\sqrt{K}} + (3A^\mu + \beta D + M)\sigma_{k+1}$$
$$\quad + \sqrt{2\beta}\left(\frac{A^\mu}{\beta} + D\right)\sqrt{\varepsilon_{k+1}} + \varepsilon_{k+1}.$$

Using the above inequality, we have

$$\mathcal{G}(\boldsymbol{x}_{K+1}, \mathcal{C}) = \langle F(\boldsymbol{x}_{K+1}), \boldsymbol{x}_{K+1} - \boldsymbol{x}_K^\star \rangle$$
$$= \langle F(\boldsymbol{x}_{K+1}), \boldsymbol{x}_{K+1} - \boldsymbol{x}_K^\mu \rangle + \langle F(\boldsymbol{x}_{K+1}), \boldsymbol{x}_K^\mu - \boldsymbol{x}_K^\star \rangle$$
$$\le \langle F(\boldsymbol{x}_{K+1}), \boldsymbol{x}_{K+1} - \boldsymbol{x}_K^\mu \rangle + \|F(\boldsymbol{x}_{K+1})\|\|\boldsymbol{x}_K^\mu - \boldsymbol{x}_K^\star\|$$
$$\le (\|\boldsymbol{\lambda}^\mu\| + 4A^\mu + \beta D + 1 + M)\frac{E^\mu}{\beta\sqrt{K}} + (3A^\mu + \beta D + M)\sigma_{k+1}$$
$$\quad + \sqrt{2\beta}\left(\frac{A^\mu}{\beta} + D\right)\sqrt{\varepsilon_{k+1}} + \varepsilon_{k+1}.$$

Moreover, by (77), we can choose small enough $\mu_{-1}$ so that

$$\mathcal{G}(\boldsymbol{x}_{K+1}, \mathcal{C}) \le (2\|\boldsymbol{\lambda}^\star\| + 5A + \beta D + 1 + M)\frac{E}{\beta\sqrt{K}} + (4A + \beta D + M)\sigma_{k+1} \tag{80}$$

$$\quad + \sqrt{2\beta}\left(\frac{2A}{\beta} + D\right)\sqrt{\varepsilon_{k+1}} + \varepsilon_{k+1}. \tag{81}$$

and

$$\|\boldsymbol{x}_{K+1} - \boldsymbol{y}_{K+1}\| \le \frac{2E}{\beta\sqrt{K}}, \tag{82}$$

where (82) uses (LB.23-2) in Lemma B.23. By (64), (80), (82) and Prop. B.24, we draw the conclusion. $\qquad\square$

## B.3 Discussion on Theorems 3.1 and 3.2 and Practical Implications

We adopt the same way of stating our theorems 3.1 and 3.2 as in the main part of (Yang et al., 2023) (see remark 2 therein) for clarity, easier comparison with one-loop algorithms, and because these are without loss of generality provided that $K, T$ are selected appropriately, as Yang et al. (2023) showed. In particular, we require knowing a *sufficiently small* $\mu_{-1}$ which depends on the selected $K$. Note that we cannot prove a faster rate than $\mathcal{O}(1/\sqrt{K})$ for the inner loop for our algorithm; so even if we further adjust $\mu_{-1}$, the rate would still be $\mathcal{O}(1/\sqrt{K})$. Given the statements in our paper, the same convergence rate of $\mathcal{O}(\frac{1}{\sqrt{K}})$ is implied for cases when we do not know a sufficiently small $\mu_{-1}$ by the argument in App. B.4 of (Yang et al., 2023). For completeness, herein, we focus on clarifying why this is the case.

*Remark* B.28. Notice that only when we require a *sufficiently small* $\mu_{-1}$ (as we do in our statements) can we use any $T, K \in N_+$. For versions of the theorems that do not require a sufficiently small $\mu_{-1}$, $K, T$ must be appropriately selected.

We obtain explicitly how $\mu_{-1}$ depends on the given $K$ by re-writing an equivalent re-formulation of App. B.4 in (Yang et al., 2023, Remark 5). In particular, for any fixed $\mu_{-1} > 0$, $K \in N_+$ and any $T \geq \mathcal{O}(\log K)$, for Algorithms 1 and 3 we have $\mathcal{G}(x_K^{(T)}, C) = \mathcal{O}(1/\sqrt{K})$.

As an example, since $\mu_{-1}$ could be an arbitrary positive number, without loss of generality, we could let $\mu_{-1} = 1$; then the above implies that when $\mu_T = \mathcal{O}(\delta^{\log K})$, we have $\mathcal{G}(x_K^{(T)}, C) = \mathcal{O}(1/\sqrt{K})$. This implies that for our Theorems 3.1 and 3.2, setting $\mu_{-1} = \mathcal{O}(\delta^{\log K})$ is enough.

**Interpretation.** Here, we provide an intuitive explanation for the above statement. For any $T \in N$, the inner loop of ACVI is solving (KKT-2), where $\mu = \mu_T$. Note that (KKT-2) is a modified problem of the original VI problem, approaching the original problem when $\mu \to 0$. Thus, when $\mu_{-1}$ is large, larger $T$ is needed in order to let (KKT-2) be a good enough approximation of the original problem.

**Practical implications.** Suppose you need $\epsilon$-accurate solution. Then $K$ is selected to satisfy $K \geq \frac{1}{\epsilon^2}$, and then $T \geq \log(K) = 2\log(1/\epsilon)$. Notice that the overall complexity to reach $\epsilon$ precision is still $\mathcal{O}(1/\epsilon^2)$ up to a log factor.

## B.4 Algorithms for solving the subproblems in Alg. 1

As in (Schmidt et al., 2011), Theorem 3.2 provides sufficient conditions on the errors so that the order of the rate is maintained. In other words, one can think of running a single step of a gradient-based method for the sub-problems. Thus, the inner loop has a complexity of the order of one (or a constant number of) gradient computations. Below, we discuss the algorithms that satisfy the assumptions of Theorem 3.2 so as the shown convergence rate holds.

**Choosing the $\mathcal{A}_x$ method.** Let $G(\boldsymbol{x}) \triangleq \boldsymbol{x} + \frac{1}{\beta}\boldsymbol{P}_c F(\boldsymbol{x}) - \boldsymbol{P}_c \boldsymbol{y}_k^{(t)} + \frac{1}{\beta}\boldsymbol{P}_c \boldsymbol{\lambda}_k^{(t)} - \boldsymbol{d}_c$, then from the proof of Thm. 1 in Yang et al. (2023) we know that $G$ is strongly monotone on $\mathcal{C}_=$. Moreover, when $F$ is $L$-Lipschitz continuous, $G$ is also Lipschitz continuous. Many common VI methods have a linear rate on the $\boldsymbol{x}$ subproblem, thus satisfying the condition we give (Tseng, 1995; Gidel et al., 2019a; Mokhtari et al., 2019). Hence, for $\mathcal{A}_x$, we could use the first-order methods for VIs listed in App. A.5 to find the unique solution of the VI problem defined by the tuple $(\mathcal{C}_=, G)$, at a linear convergence rate. To solve a VI defined by $(\mathcal{C}_=, G)$, we need to compute the projection $\Pi_{\mathcal{C}_=}$, which is straightforward by noticing that $\Pi_{\mathcal{C}_=}(\boldsymbol{x}) = \boldsymbol{P}_c \boldsymbol{x} + \boldsymbol{d}_c, \forall \boldsymbol{x}$.

**Choosing the $\mathcal{A}_y$ method.** If using $\wp$ to be $\wp_1$, the objective of the $\boldsymbol{y}$ subproblem is strongly convex but non-smooth. Thus, to our knowledge, there is no known method to achieve a linear rate for general constraints without additional assumptions. However, one could satisfy the condition we give by using methods for $\mathcal{A}_y$ to exploit the constraint structure further. For example, if the constraints are linear, it is straightforward to derive the update rule for the $\boldsymbol{y}$ subproblem, which satisfies the conditions of the theorem.

On the other hand, as discussed in Remark B.26, when choosing $\wp$ to be $\wp_2$, the objective of the subproblem in line 9 of Alg. 1 is smooth and strongly convex and thus could be solved by commonly used unconstrained first-order solvers such as gradient descent at a linear rate.

**Discussion.** The above facts allude to the advantages of ACVI, as the sub-problems are "easier" than the original problem. To summarize, *(i)* if $F$ is monotone, the $x$ subproblem is strongly monotone; and *(ii)* the $y$ subproblem is regular minimization which is significantly less challenging to solve in practice, and also it is strongly convex.

### B.5 PROOF OF THEOREM 4.1: CONVERGENCE OF P-ACVI

#### B.5.1 SETTING AND NOTATIONS

We define the following maps from $\mathbb{R}^n$ to $\mathbb{R}^n$:

$$f(\boldsymbol{x}) \triangleq F(\boldsymbol{x}^\star)^\mathsf{T}\boldsymbol{x} + \mathbb{1}(\boldsymbol{C}\boldsymbol{x} = \boldsymbol{d})\,, \tag{$f_k$}$$

$$f_k(\boldsymbol{x}) \triangleq F(\boldsymbol{x}_{k+1})^\mathsf{T}\boldsymbol{x} + \mathbb{1}(\boldsymbol{C}\boldsymbol{x} = \boldsymbol{d})\,, \quad \text{and} \tag{$f$}$$

$$g(\boldsymbol{y}) \triangleq \mathbb{1}(\varphi(\boldsymbol{y}) \leq \boldsymbol{0})\,, \tag{$g$}$$

where $\boldsymbol{x}^\star$ is a solution of (KKT). Let $\boldsymbol{y}^\star = \boldsymbol{x}^\star$. Then $(\boldsymbol{x}^\star, \boldsymbol{y}^\star)$ is an optimal solution of ($f$-Pr). Let us denote with $(\boldsymbol{x}_k^\star, \boldsymbol{y}_k^\star, \boldsymbol{\lambda}_k^\star)$ the KKT point of ($f_k$-Pr). Note that in this case, the problem ($f_k$-Pr) is equivalent to ($f_k$-Pr-2).

#### B.5.2 INTERMEDIATE RESULTS

In P-ACVI-Algorithm 2, by the definition of $\boldsymbol{y}_{k+1}$ (line 7 of Algorithm 2), $\boldsymbol{y}_k^\star$ and $\boldsymbol{y}^\star$ we immediately know that

$$g(\boldsymbol{y}_{k+1}) = g(\boldsymbol{y}_k^\star) = g(\boldsymbol{y}^\star) = 0. \tag{83}$$

The intermediate results for the proofs of Theorem 3.1 still hold in this case only with a little modification, and the proofs of them are very close to the previous ones. To avoid redundancy, we omit these proofs.

**Proposition B.29** (Relation between $f_k$ and $f$)**.** *If $F$ is monotone, then $\forall k \in \mathbb{N}$, we have that:*

$$f_k(\boldsymbol{x}_{k+1}) - f_k(\boldsymbol{x}^\star) \geq f(\boldsymbol{x}_{k+1}) - f(\boldsymbol{x}^\star).$$

**Lemma B.30.** *For all $\boldsymbol{x}$ and $\boldsymbol{y}$, we have*

$$f(\boldsymbol{x}) + g(\boldsymbol{y}) - f(\boldsymbol{x}^\star) - g(\boldsymbol{y}^\star) + \langle \boldsymbol{\lambda}^\star, \boldsymbol{x} - \boldsymbol{y} \rangle \geq 0, \tag{LB.30-$f$}$$

*and:*

$$f_k(\boldsymbol{x}) + g(\boldsymbol{y}) - f_k(\boldsymbol{x}_k^\star) - g(\boldsymbol{y}_k^\star) + \langle \boldsymbol{\lambda}_k^\star, \boldsymbol{x} - \boldsymbol{y} \rangle \geq 0\,. \tag{LB.30-$f_k$}$$

The following lemma lists some simple but useful facts that we will use in the following proofs.

**Lemma B.31.** *For the problems ($f$-Pr), ($f_k$-Pr) and Algorithm 2, we have*

$$\boldsymbol{0} \in \partial f_k(\boldsymbol{x}_{k+1}) + \boldsymbol{\lambda}_k + \beta(\boldsymbol{x}_{k+1} - \boldsymbol{y}_k) \tag{LB.31-1}$$

$$\boldsymbol{0} \in \partial g(\boldsymbol{y}_{k+1}) - \boldsymbol{\lambda}_k - \beta(\boldsymbol{x}_{k+1} - \boldsymbol{y}_{k+1}), \tag{LB.31-2}$$

$$\boldsymbol{\lambda}_{k+1} - \boldsymbol{\lambda}_k = \beta(\boldsymbol{x}_{k+1} - \boldsymbol{y}_{k+1}), \tag{LB.31-3}$$

$$-\boldsymbol{\lambda}^\star \in \partial f(\boldsymbol{x}^\star), \tag{LB.31-4}$$

$$-\boldsymbol{\lambda}_k^\star \in \partial f_k(\boldsymbol{x}_k^\star), \tag{LB.31-5}$$

$$\boldsymbol{\lambda}^\star \in \partial g(\boldsymbol{y}^\star), \tag{LB.31-6}$$

$$\boldsymbol{\lambda}_k^\star \in \partial g(\boldsymbol{y}_k^\star), \tag{LB.31-7}$$

$$\boldsymbol{x}^\star = \boldsymbol{y}^\star, \tag{LB.31-8}$$

$$\boldsymbol{x}_k^\star = \boldsymbol{y}_k^\star, \tag{LB.31-9}$$

Like in App.B.1.2, we also define $\hat{\nabla} f_k(\boldsymbol{x}_{k+1})$ and $\hat{\nabla} g(\boldsymbol{y}_{k+1})$ by ($\hat{\nabla} f_k$) and ($\hat{\nabla} g$), resp.

Then, from (LB.31-1) and (LB.31-2) it follows that:

$$\hat{\nabla} f_k(\boldsymbol{x}_{k+1}) \in \partial f_k(\boldsymbol{x}_{k+1}) \text{ and } \hat{\nabla} g(\boldsymbol{y}_{k+1}) \in \partial g(\boldsymbol{y}_{k+1})\,. \tag{84}$$

**Lemma B.32.** *For the iterates $\boldsymbol{x}_{k+1}$, $\boldsymbol{y}_{k+1}$, and $\boldsymbol{\lambda}_{k+1}$ of the P-ACVI—Algorithm 2—we have:*

$$\langle \hat{\nabla} g(\boldsymbol{y}_{k+1}), \boldsymbol{y}_{k+1} - \boldsymbol{y} \rangle = -\langle \boldsymbol{\lambda}_{k+1}, \boldsymbol{y} - \boldsymbol{y}_{k+1} \rangle, \tag{85}$$

*and*

$$\langle \hat{\nabla} f_k(\boldsymbol{x}_{k+1}), \boldsymbol{x}_{k+1} - \boldsymbol{x} \rangle + \langle \hat{\nabla} g(\boldsymbol{y}_{k+1}), \boldsymbol{y}_{k+1} - \boldsymbol{y} \rangle = -\langle \boldsymbol{\lambda}_{k+1}, \boldsymbol{x}_{k+1} - \boldsymbol{y}_{k+1} - \boldsymbol{x} + \boldsymbol{y} \rangle \\ + \beta \langle -\boldsymbol{y}_{k+1} + \boldsymbol{y}_k, \boldsymbol{x}_{k+1} - \boldsymbol{x} \rangle. \tag{86}$$

**Lemma B.33.** *For the $\boldsymbol{x}_{k+1}$, $\boldsymbol{y}_{k+1}$, and $\boldsymbol{\lambda}_{k+1}$ iterates of the P-ACVI—Algorithm 2—we have:*

$$\langle \hat{\nabla} f_k(\boldsymbol{x}_{k+1}), \boldsymbol{x}_{k+1} - \boldsymbol{x}^\star \rangle + \langle \hat{\nabla} g(\boldsymbol{y}_{k+1}), \boldsymbol{y}_{k+1} - \boldsymbol{y}^\star \rangle + \langle \boldsymbol{\lambda}^\star, \boldsymbol{x}_{k+1} - \boldsymbol{y}_{k+1} \rangle$$
$$\leq \frac{1}{2\beta} \|\boldsymbol{\lambda}_k - \boldsymbol{\lambda}^\star\|^2 - \frac{1}{2\beta} \|\boldsymbol{\lambda}_{k+1} - \boldsymbol{\lambda}^\star\|^2 + \frac{\beta}{2} \|\boldsymbol{y}^\star - \boldsymbol{y}_k\|^2 - \frac{\beta}{2} \|\boldsymbol{y}^\star - \boldsymbol{y}_{k+1}\|^2$$
$$- \frac{1}{2\beta} \|\boldsymbol{\lambda}_{k+1} - \boldsymbol{\lambda}_k\|^2 - \frac{\beta}{2} \|\boldsymbol{y}_k - \boldsymbol{y}_{k+1}\|^2,$$

*and*

$$\langle \hat{\nabla} f_k(\boldsymbol{x}_{k+1}), \boldsymbol{x}_{k+1} - \boldsymbol{x}_k^\star \rangle + \langle \hat{\nabla} g(\boldsymbol{y}_{k+1}), \boldsymbol{y}_{k+1} - \boldsymbol{y}_k^\star \rangle + \langle \boldsymbol{\lambda}_k^\star, \boldsymbol{x}_{k+1} - \boldsymbol{y}_{k+1} \rangle$$
$$\leq \frac{1}{2\beta} \|\boldsymbol{\lambda}_k - \boldsymbol{\lambda}_k^\star\|^2 - \frac{1}{2\beta} \|\boldsymbol{\lambda}_{k+1} - \boldsymbol{\lambda}_k^\star\|^2 + \frac{\beta}{2} \|\boldsymbol{y}_k^\star - \boldsymbol{y}_k\|^2 - \frac{\beta}{2} \|\boldsymbol{y}_k^\star - \boldsymbol{y}_{k+1}\|^2$$
$$- \frac{1}{2\beta} \|\boldsymbol{\lambda}_{k+1} - \boldsymbol{\lambda}_k\|^2 - \frac{\beta}{2} \|\boldsymbol{y}_k - \boldsymbol{y}_{k+1}\|^2.$$

**Lemma B.34.** *For the $\boldsymbol{x}_{k+1}$, $\boldsymbol{y}_{k+1}$, and $\boldsymbol{\lambda}_{k+1}$ iterates of the P-ACVI—Algorithm 2—we have:*

$$f(\boldsymbol{x}_{k+1}) + g(\boldsymbol{y}_{k+1}) - f(\boldsymbol{x}^\star) - g(\boldsymbol{y}^\star) + \langle \boldsymbol{\lambda}^\star, \boldsymbol{x}_{k+1} - \boldsymbol{y}_{k+1} \rangle$$
$$\leq f_k(\boldsymbol{x}_{k+1}) + g(\boldsymbol{y}_{k+1}) - f_k(\boldsymbol{x}^\star) - g(\boldsymbol{y}^\star) + \langle \boldsymbol{\lambda}^\star, \boldsymbol{x}_{k+1} - \boldsymbol{y}_{k+1} \rangle$$
$$\leq \frac{1}{2\beta} \|\boldsymbol{\lambda}_k - \boldsymbol{\lambda}^\star\|^2 - \frac{1}{2\beta} \|\boldsymbol{\lambda}_{k+1} - \boldsymbol{\lambda}^\star\|^2$$
$$+ \frac{\beta}{2} \|-\boldsymbol{y}_k + \boldsymbol{y}^\star\|^2 - \frac{\beta}{2} \|-\boldsymbol{y}_{k+1} + \boldsymbol{y}^\star\|^2$$
$$- \frac{1}{2\beta} \|\boldsymbol{\lambda}_{k+1} - \boldsymbol{\lambda}_k\|^2 - \frac{\beta}{2} \|-\boldsymbol{y}_{k+1} + \boldsymbol{y}_k\|^2 \tag{LB.34}$$

The following theorem upper bounds the analogous quantity but for $f_k(\cdot)$ (instead of $f$), and further asserts that the difference between the $\boldsymbol{x}_{k+1}$ and $\boldsymbol{y}_{k+1}$ iterates of P-ACVI (Algorithm 2) tends to $0$ asymptotically.

**Theorem B.35** (Asymptotic convergence of $(\boldsymbol{x}_{k+1} - \boldsymbol{y}_{k+1})$ of P-ACVI). *For the $\boldsymbol{x}_{k+1}$, $\boldsymbol{y}_{k+1}$, and $\boldsymbol{\lambda}_{k+1}$ iterates of the P-ACVI—Algorithm 2—we have:*

$$f_k(\boldsymbol{x}_{k+1}) - f_k(\boldsymbol{x}_k^\star) \leq \|\boldsymbol{\lambda}_{k+1}\| \|\boldsymbol{x}_{k+1} - \boldsymbol{y}_{k+1}\| + \beta \|\boldsymbol{y}_{k+1} - \boldsymbol{y}_k\| \|\boldsymbol{x}_{k+1} - \boldsymbol{x}_k^\star\| \to 0, \tag{TB.35-$f_k$-UB}$$

*and*

$$\boldsymbol{x}_{k+1} - \boldsymbol{y}_{k+1} \to \boldsymbol{0}, \qquad \text{as } k \to \infty.$$

**Lemma B.36.** *For the $\boldsymbol{x}_{k+1}$, $\boldsymbol{y}_{k+1}$, and $\boldsymbol{\lambda}_{k+1}$ iterates of the P-ACVI—Algorithm 2—we have:*

$$\frac{1}{2\beta} \|\boldsymbol{\lambda}_{k+1} - \boldsymbol{\lambda}_k\|^2 + \frac{\beta}{2} \|-\boldsymbol{y}_{k+1} + \boldsymbol{y}_k\|^2 \leq \frac{1}{2\beta} \|\boldsymbol{\lambda}_k - \boldsymbol{\lambda}_{k-1}\|^2 + \frac{\beta}{2} \|-\boldsymbol{y}_k + \boldsymbol{y}_{k-1}\|^2. \tag{LB.36}$$

**Lemma B.37.** *If $F$ is monotone on $\mathcal{C}_=$, then for Algorithm 2, we have:*

$$f_K\left(\boldsymbol{x}_{K+1}\right) - f_K\left(\boldsymbol{x}_K^\star\right) \leq \frac{\Delta}{K+1} + \left(2\sqrt{\Delta} + \frac{1}{\sqrt{\beta}}\|\boldsymbol{\lambda}^\star\| + \sqrt{\beta}D\right)\sqrt{\frac{\Delta}{K+1}}, \qquad \text{(LB.37-1)}$$

$$\text{and} \qquad \|\boldsymbol{x}_{K+1} - \boldsymbol{y}_{K+1}\| \leq \sqrt{\frac{\Delta}{\beta\left(K+1\right)}}, \qquad \text{(LB.37-2)}$$

*where $\Delta \triangleq \frac{1}{\beta}\|\boldsymbol{\lambda}_0 - \boldsymbol{\lambda}^\star\|^2 + \beta\|\boldsymbol{y}_0 - \boldsymbol{y}^\star\|^2$.*

### B.5.3 PROVING THEOREM. 4.1

We are now ready to prove Theorem 4.1. Here we give a nonasymptotic convergence rate of P-ACVI-Algorithm 2.

**Theorem B.38** (Restatement of Theorem 4.1). *Given an continuous operator $F\colon \mathcal{X} \to \mathbb{R}^n$, assume $F$ is monotone on $\mathcal{C}_=$, as per Def. 2.1. Let $(\boldsymbol{x}_K, \boldsymbol{y}_K, \boldsymbol{\lambda}_K)$ denote the last iterate of Algorithm 3. Then $\forall K \in \mathbb{N}_+$, we have*

$$\mathcal{G}(\boldsymbol{x}_K, \mathcal{C}) \leq \frac{\Delta}{K} + \left(2\sqrt{\Delta} + \frac{1}{\sqrt{\beta}}\|\boldsymbol{\lambda}^\star\| + \sqrt{\beta}D\right)\sqrt{\frac{\Delta}{K}} \qquad \text{(na-lf-Rate)}$$

*and*

$$\left\|\boldsymbol{x}^K - \boldsymbol{y}^K\right\| \leq \sqrt{\frac{\Delta}{\beta K}}, \qquad (87)$$

*where $\Delta \triangleq \frac{1}{\beta}\|\boldsymbol{\lambda}_0 - \boldsymbol{\lambda}^\star\|^2 + \beta\|\boldsymbol{y}_0 - \boldsymbol{y}^\star\|^2$ and $D \triangleq \sup_{\boldsymbol{x},\boldsymbol{y}\in\mathcal{C}}\|\boldsymbol{x} - \boldsymbol{y}\|$, and $M \triangleq \sup_{\boldsymbol{x}\in\mathcal{C}}\|F(\boldsymbol{x})\|$.*

*Proof of Theorem 4.1.* Note that

$$\begin{aligned}
(f_k\text{-Pr-2}) &\Leftrightarrow \min_{\boldsymbol{x}\in\mathcal{C}}\langle F(\boldsymbol{x}_{k+1}), \boldsymbol{x}\rangle \\
&\Leftrightarrow \max_{\boldsymbol{x}\in\mathcal{C}}\langle F(\boldsymbol{x}_{k+1}), \boldsymbol{x}_{k+1} - \boldsymbol{x}\rangle \\
&\Leftrightarrow \mathcal{G}(\boldsymbol{x}_{k+1}, \mathcal{C}),
\end{aligned}$$

from which we deduce

$$\mathcal{G}(\boldsymbol{x}_{k+1}, \mathcal{C}) = \langle F(\boldsymbol{x}_{k+1}), \boldsymbol{x}_{k+1} - \boldsymbol{x}_k^\star\rangle = f_K\left(\boldsymbol{x}_{K+1}\right) - f_K\left(\boldsymbol{x}_K^\star\right), \forall k. \qquad (88)$$

Combining with Lemma B.37, we obtain (na-lf-Rate) and (87).

$$\square$$

## C    IMPLEMENTATION DETAILS

In this section, we provide the details on the implementation of the results presented in § 5 in the main part, as well as those of the additional results presented in App. D. In addition, we provide the source code through the following link: https://github.com/Chavdarova/I-ACVI.

### C.1    IMPLEMENTATION DETAILS FOR THE 2D-BG GAME

Recall that we defined the 2D bilinear game as:

$$\min_{x_1 \in \triangle} \max_{x_2 \in \triangle} x_1 x_2 \quad \text{where } \triangle = \{x \in \mathbb{R} | -0.4 \le x \le 2.4\}. \tag{2D-BG}$$

To avoid confusion in the notation, in the remainder of this section, we rename the players in (2D-BG) as $p_1$ and $p_2$:

$$\min_{p_1 \in \triangle} \max_{p_2 \in \triangle} p_1 p_2 \quad \text{where } \triangle = \{p \in \mathbb{R} | -0.4 \le p \le 2.4\}$$

In the following, we list the I-ACVI and P-ACVI implementations.

**I-ACVI.**    For I-ACVI (Algorithm 1), we use the following Python code and the PyTorch library (Paszke et al., 2017). We set $\beta = 0.5$, $\mu = 3$, $K = 20$, $\ell = 20$, $\delta = 0.5$ and use a learning rate of 0.1. The following implementation uses the standard log-barrier ($\wp_1$).

Listing 1: Implementation of the I-ACVI algorithm (using ($\wp_1$)) on the 2D constrained bilinear game.

```
1 import torch
2 lr = 0.1     # learning rate
3 beta = 0.5   # ACVI beta parameter
4 mu = 3       # ACVI mu parameter
5 K = 20       # ACVI K parameter
6 l = 20       # I-ACVI l parameter
7 delta = 0.5 # ACVI delta parameter: exponential decay of mu
8
9 p1_x = torch.nn.Parameter(torch.tensor(2.0))
10 p1_y = torch.nn.Parameter(torch.tensor(2.0))
11 p1_l = torch.nn.Parameter(torch.tensor(0.0))
12
13 p2_x = torch.nn.Parameter(torch.tensor(2.0))
14 p2_y = torch.nn.Parameter(torch.tensor(2.0))
15 p2_l = torch.nn.Parameter(torch.tensor(0.0))
16
17 while mu > 0.0001:
18
19   for itr in range(K):
20
21     for _ in range(l): # solve x problem (line 8 of algorithm)
22       loss_p1 = 1/beta * p1_x * p2_x + 0.5 * (p1_x - p1_y + p1_l/beta).pow(2)
23       p1_x.grad = None
24       loss_p1.backward()
25       with torch.no_grad():
26         p1_x -= lr * p1_x.grad
27
28       loss_p2 = -1/beta * p1_x * p2_x + 0.5 * (p2_x - p2_y + p2_l/beta).pow(2)
29       p2_x.grad = None
30       loss_p2.backward()
31       with torch.no_grad():
32         p2_x -= lr * p2_x.grad
33
34     for _ in range(l): # solve y problem (line 9 of algorithm)
35       phi_1 = p1_y + 0.4 # -0.4 < p1_y # define all the inequality constraints
36       phi_2 = 2.4 - p1_y # p1_y < 2.4
37       phi_3 = p2_y + 0.4 # -0.4 < p2_y
38       phi_4 = 2.4 - p2_y # p1_y < 2.4
39       log_term = -mu * (phi_1.log() + phi_2.log() + phi_3.log() + phi_4.log())
40       loss = log_term + beta/2 * (p1_y - p1_x - p1_l/beta).pow(2)
41                       + beta/2 * (p2_y - p2_x - p2_l/beta).pow(2)
42       p1_y.grad, p2_y.grad = None, None
43       loss.backward()
44       with torch.no_grad():
```

```
45          p1_y -= lr * p1_y.grad
46          p2_y -= lr * p2_y.grad
47
48      # update the lambdas (line 10 of algorithm)
49      with torch.no_grad():
50        p1_l += beta * (p1_x - p1_y)
51        p2_l += beta * (p2_x - p2_y)
52
53   mu *= delta # decay mu
```

For completeness, we provide the source code below when using the ($\wp_2$) barrier map instead of ($\wp_1$).

Listing 2: Implementation of the I-ACVI algorithm (using ($\wp_2$)) on the 2D constrained bilinear game.

```
1 import torch
2 lr = 0.1      # learning rate
3 beta = 0.5    # ACVI beta parameter
4 mu = 3        # ACVI mu parameter
5 K = 20        # ACVI K parameter
6 l = 20        # I-ACVI l parameter
7 delta = 0.5 # ACVI delta parameter: exponential decay of mu
8 c = torch.tensor([1.0]) # c parameter of the extended barrier
9
10 p1_x = torch.nn.Parameter(torch.tensor(2.0))
11 p1_y = torch.nn.Parameter(torch.tensor(2.0))
12 p1_l = torch.nn.Parameter(torch.tensor(0.0))
13
14 p2_x = torch.nn.Parameter(torch.tensor(2.0))
15 p2_y = torch.nn.Parameter(torch.tensor(2.0))
16 p2_l = torch.nn.Parameter(torch.tensor(0.0))
17
18 while mu > 0.0001:
19
20   for itr in range(K):
21
22     for _ in range(l): # solve x problem (line 8 of algorithm)
23       loss_p1 = 1/beta * p1_x * p2_x + 0.5 * (p1_x - p1_y + p1_l/beta).pow(2)
24       p1_x.grad = None
25       loss_p1.backward()
26       with torch.no_grad():
27         p1_x -= lr * p1_x.grad
28
29       loss_p2 = -1/beta * p1_x * p2_x + 0.5 * (p2_x - p2_y + p2_l/beta).pow(2)
30       p2_x.grad = None
31       loss_p2.backward()
32       with torch.no_grad():
33         p2_x -= lr * p2_x.grad
34
35     for _ in range(l): # solve y problem (line 9 of algorithm)
36       phi_1 = p1_y + 0.4 # -0.4 < p1_y # define all the inequality constraints
37       phi_2 = 2.4 - p1_y # p1_y < 2.4
38       phi_3 = p2_y + 0.4 # -0.4 < p2_y
39       phi_4 = 2.4 - p2_y # p1_y < 2.4
40       log_terms = [phi_1, phi_2, phi_3, phi_4]
41       clip_condition = [-phi <= -torch.exp(-c/mu) for phi in log_terms]
42       new_log_terms = [-mu*torch.log(phi) if condition else
43                         -mu*torch.exp(c/mu)*phi+mu+c for
44                         phi, condition in zip(log_terms, clip_condition)]
45       loss = sum(new_log_terms) + beta/2 * (p1_y - p1_x - p1_l/beta).pow(2)
46                                 + beta/2 * (p2_y - p2_x - p2_l/beta).pow(2)
47
48       p1_y.grad, p2_y.grad = None, None
49       loss.backward()
50       with torch.no_grad():
51         p1_y -= lr * p1_y.grad
52         p2_y -= lr * p2_y.grad
53
54     # update the lambdas (line 10 of algorithm)
55     with torch.no_grad():
56       p1_l += beta * (p1_x - p1_y)
57       p2_l += beta * (p2_x - p2_y)
58
59   mu *= delta # decay mu
```

**PI-ACVI.** For PI-ACVI, we use the following Python code implementing Algorithm 2 using the Pytorch library. We set $\beta = 0.5$, $K = 20$, $\ell = 20$, and use a learning rate of $0.1$.

Listing 3: Implementation of the PI-ACVI algorithm on the 2D constrained bilinear game.

```python
import torch

lr = 0.1    # learning rate
beta = 0.5  # ACVI beta parameter
K = 20      # ACVI K parameter
l = 20      # I-ACVI l parameter

p1_x = torch.nn.Parameter(torch.tensor(2.0))
p1_y = torch.nn.Parameter(torch.tensor(2.0))
p1_l = torch.nn.Parameter(torch.tensor(0.0))

p2_x = torch.nn.Parameter(torch.tensor(2.0))
p2_y = torch.nn.Parameter(torch.tensor(2.0))
p2_l = torch.nn.Parameter(torch.tensor(0.0))

for itr in range(K):

  # solve x problem (line 6 of algorithm)
  for _ in range(l):
    loss_p1 = 1/beta * p1_x * p2_x + 0.5 * (p1_x - p1_y + p1_l/beta).pow(2)
    p1_x.grad = None
    loss_p1.backward()
    with torch.no_grad():
      p1_x -= lr * p1_x.grad

    loss_p2 = -1/beta * p1_x * p2_x + 0.5 * (p2_x - p2_y + p2_l/beta).pow(2)
    p2_x.grad = None
    loss_p2.backward()
    with torch.no_grad():
      p2_x -= lr * p2_x.grad

  # solve y problem using projection (line 7 of algorithm)
  with torch.no_grad():
    p1_y.data = p1_x + p1_l/beta
    p1_y.data = p1_y.clip(-0.4, 2.4)

    p2_y.data = p2_x + p2_l/beta
    p2_y.data = p2_y.clip(-0.4, 2.4)

  # update the lambdas (line 8 of algorithm)
  with torch.no_grad():
    p1_l += beta * (p1_x - p1_y)
    p2_l += beta * (p2_x - p2_y)
```

## C.2 Implementation details for the HBG game

**Solution and relative error.** The solution of (HBG) is $\boldsymbol{x}^\star = \frac{1}{500}\boldsymbol{e}$, with $\boldsymbol{e} \in \mathbb{R}^{1000}$. As a metric of the experiments on this problem, we use the relative error: $\varepsilon_r(\boldsymbol{x}_k) = \frac{\|\boldsymbol{x}_k - \boldsymbol{x}^\star\|}{\|\boldsymbol{x}^\star\|}$.

**Experiments of Fig.4.a showing CPU time to reach a fixed relative error.** The target relative error is $0.02$. We set the step size of GDA, EG, and OGDA to $0.3$ and use $k = 5$ and $\alpha = 0.5$ for LA-GDA. For I-ACVI, we set $\beta = 0.5$, $\mu_{-1} = 10^{-6}$, $\delta = 0.8$, $\boldsymbol{\lambda}_0 = \mathbf{0}$, $K = 10$, $\ell = 10$ and the step size is $0.05$.

**Experiments of Fig.4.b showing the number of iterations to reach a fixed relative error.** Hyperparameters are the same as for Fig.4.a. We vary the rotation "strength" $(1 - \eta)$, with $\eta \in (0, 1)$.

**Experiments of Fig.4.c showing the impact of $K_0$.** For this experiment, we depict, for various pairs $(K_0, K_+)$, how many iterations are required to reach a relative error smaller than $10^{-4}$. We set $\beta = 0.5$, $\mu = 1e - 6$, $\delta = 0.8$, $T = 5000$ and $0.05$ as learning rate. We experiment with $K_0 \in \{5, 10, 20, 30, 40, 50, 60, 70, 80, 90, 100, 110, 120, 130\}$ and $K_+ \in \{1, 5, 10, 20, 30, 40, 50, 60, 70\}$.

The following Python code snippet shows an implementation of I-ACVI (Algorithm 1) on the (HBG) game:

Listing 4: Implementation of the I-ACVI algorithm on the HBG game.

```python
import numpy as np
import time

eps = .02  # target relative error
dim = 500  # dim(x1) == dim(x2) == dim
x_opt = np.ones((2*dim,1))/dim # solution

# I-ACVI parameters
beta, mu, delta, K, l, T, lr = 0.5, 1e-6, .8, 10, 10, 100, 0.05

# Building HBG matrices
eta = 0.05
A1 = np.concatenate((eta*np.identity(dim), (1-eta)*np.identity(dim)), axis=1)
A2 = np.concatenate((-(1-eta)*np.identity(dim), eta*np.identity(dim)), axis=1)
A  = np.concatenate((A1, A2), axis=0)

# Build projection matrix Pc
temp1 = np.concatenate((np.ones((1,dim)), np.zeros((1,dim))), axis=1)
temp2 = np.concatenate((np.zeros((1,dim)), np.ones((1,dim))), axis=1)
C = np.concatenate((temp1,temp2), axis=0)
d = np.ones((2,1))
temp = np.linalg.inv(np.dot(C, C.T))
temp = np.dot(C.T, temp)
dc = np.dot(temp, d)
Pc = np.identity(2*dim) - np.dot(temp,C)

# Initialize players
init = np.random.rand(2*dim, 1)
init[:dim] = init[:dim] / np.sum(init[:dim]) # ensuring it is part of the simplex
init[dim:] = init[dim:] / np.sum(init[dim:])
z_x = np.copy(init)
z_y = np.copy(init)
z_lmd = np.zeros(init.shape)

finished, cnt, t0 = False, 0, time.time()

for _ in range(T):
  mu *= delta
  for _ in range(K):
    cnt += 1
    # Solve approximately the X problem (line 8 of algorithm)
    for _ in range(l):
      g = z_x + 1/beta * np.dot(Pc, np.dot(A,z_x)) - np.dot(Pc, z_y) + 1/beta * np.dot(Pc,
      z_lmd) - dc
      z_x -= lr * g

    if np.linalg.norm(z_x-x_opt)/np.linalg.norm(x_opt) <= eps:
      finished = True
      print(f"Reached a relative error of {eps} after {cnt} iterations in
      {time.time()-t0:.2f} sec.")
      break

    # Solve approximately the Y problem (line 9 of algorithm)
    for _ in range(l):
      assert all(z_y > 0) # ensuring the log terms are positive
      g = - mu * 1/z_y + beta*(z_y - z_x - z_lmd/beta)
      z_y -= lr * g

    # Update lambdas (line 10 of algorithm)
    z_lmd += beta*(z_x-z_y)

  if finished:
    break
```

## C.3 IMPLEMENTATION DETAILS FOR THE C-GAN GAME

For the experiments on the MNIST dataset, we use the source code of Chavdarova et al. (2021) for the baselines, and we build on it to implement PI-ACVI (Algorithm 2). For completeness, we provide an overview of the implementation.

**Models.** We used the DCGAN architectures (Radford et al., 2016), listed in Table 2, and the parameters of the models are initialized using PyTorch default initialization. For experiments on this

| Generator | Discriminator |
|---|---|
| *Input:* $\boldsymbol{z} \in \mathbb{R}^{128} \sim \mathcal{N}(0, I)$ | *Input:* $\boldsymbol{x} \in \mathbb{R}^{1 \times 28 \times 28}$ |
| transposed conv. (ker: $3 \times 3$, $128 \to 512$; stride: 1) | conv. (ker: $4 \times 4$, $1 \to 64$; stride: 2; pad:1) |
| Batch Normalization | LeakyReLU (negative slope: 0.2) |
| ReLU | conv. (ker: $4 \times 4$, $64 \to 128$; stride: 2; pad:1) |
| transposed conv. (ker: $4 \times 4$, $512 \to 256$, stride: 2) | Batch Normalization |
| Batch Normalization | LeakyReLU (negative slope: 0.2) |
| ReLU | conv. (ker: $4 \times 4$, $128 \to 256$; stride: 2; pad:1) |
| transposed conv. (ker: $4 \times 4$, $256 \to 128$, stride: 2) | Batch Normalization |
| Batch Normalization | LeakyReLU (negative slope: 0.2) |
| ReLU | conv. (ker: $3 \times 3$, $256 \to 1$; stride: 1) |
| transposed conv. (ker: $4 \times 4$, $128 \to 1$, stride: 2, pad: 1) | $Sigmoid(\cdot)$ |
| $Tanh(\cdot)$ | |

Table 2: DCGAN architectures (Radford et al., 2016) used for experiments on **MNIST**. With "conv." we denote a convolutional layer and "transposed conv" a transposed convolution layer (Radford et al., 2016). We use *ker* and *pad* to denote *kernel* and *padding* for the (transposed) convolution layers, respectively. With $h \times w$, we denote the kernel size. With $c_{in} \to y_{out}$ we denote the number of channels of the input and output, for (transposed) convolution layers. The models use Batch Normalization (Ioffe & Szegedy, 2015) layers.

dataset, we used the *non-saturating* GAN loss as proposed in (Goodfellow et al., 2014):

$$\mathcal{L}_D = \mathbb{E}_{\tilde{\boldsymbol{x}}_d \sim p_d} \log\big(D(\tilde{\boldsymbol{x}}_d)\big) + \mathbb{E}_{\tilde{\boldsymbol{z}} \sim p_z} \log\Big(1 - D\big(G(\tilde{\boldsymbol{z}})\big)\Big) \tag{L-D}$$

$$\mathcal{L}_G = \mathbb{E}_{\tilde{\boldsymbol{z}} \sim p_z} \log\Big(D\big(G(\tilde{\boldsymbol{z}})\big)\Big), \tag{L-G}$$

where $G(\cdot), D(\cdot)$ denote the generator and discriminator, resp., and $p_d$ and $p_z$ denote the data and the latent distributions (the latter predefined as normal distribution).

**Details on the PI-ACVI implementation.** When implementing PI-ACVI on MNIST, we set $\beta = 0.5$, and $K = 5000$, we use $\ell_+ = 20$ and $\ell_0 \in \{100, 500\}$. We consider only inequality constraints (and there are no equality constraints), therefore, the matrices $\boldsymbol{P}_c$ and $\boldsymbol{d}_c$ are identity and zero, respectively. As inequality constraints, we use 100 randomly generated linear inequalities for the Generator and 100 for the Discriminator.

**Projection details.** Suppose the linear inequality constraints for the Generator are $\boldsymbol{A\theta} \leq \boldsymbol{b}$, where $\boldsymbol{\theta} \in \mathbb{R}^n$ is the vector of all parameters of the Generator, $\boldsymbol{A} = (\boldsymbol{a}_1^{\mathsf{T}}, \dots, \boldsymbol{a}_{100}^{\mathsf{T}})^{\mathsf{T}} \in \mathbb{R}^{100 \times n}$, $\boldsymbol{b} = (b_1, \dots, b_{100}) \in \mathbb{R}^{100}$. We use the *greedy projection algorithm* described in (Beck, 2017). A greedy projection algorithm is essentially a projected gradient method; it is easy to implement in high-dimension problems and has a convergence rate of $O(1/\sqrt{K})$. See Chapter 8.2.3 in (Beck, 2017) for more details. Since the dimension $n$ is very large, at each step of the projection, one could only project $\boldsymbol{\theta}$ to one hyperplane $\boldsymbol{a}_i^{\mathsf{T}} \boldsymbol{x} = b_i$ for some $i \in \mathcal{I}(\boldsymbol{\theta})$, where

$$\mathcal{I}(\boldsymbol{\theta}) \triangleq \{j | \boldsymbol{a}_j^{\mathsf{T}} \boldsymbol{\theta} > b_j\}.$$

For every $j \in \{1, 2, \dots, 100\}$, let

$$\mathcal{S}_j \triangleq \{\boldsymbol{x} | \boldsymbol{a}_j^{\mathsf{T}} \boldsymbol{x} \leq b_j\}.$$

The greedy projection method chooses $i$ so that $i \in \arg\max\{dist(\boldsymbol{\theta}, \mathcal{S}_i)\}$. Note that as long as $\boldsymbol{\theta}$ is not in the constraint set $C_{\leq} = \{\boldsymbol{x} | \boldsymbol{Ax} \leq \boldsymbol{b}\}$, $i$ would be in $\mathcal{I}(\boldsymbol{\theta})$. Algorithm 4 gives the details of the greedy projection method we use for the baseline, written for the Generator only for simplicity; the same projection method is used for the Discriminator as well.

**Metrics.** We describe the metrics for the MNIST experiments. We use the two standard GAN metrics, Inception Score (IS, Salimans et al., 2016) and Fréchet Inception Distance (FID, Heusel et al., 2017). Both FID and IS rely on a pre-trained classifier and take a finite set of $\tilde{m}$ samples from the generator to compute these. Since **MNIST** has greyscale images, we used a classifier trained on this dataset and used $\tilde{m} = 5000$.

---

**Algorithm 4** Greedy projection method for the baseline.

---
1: **Input:** $\boldsymbol{\theta} \in \mathbb{R}^n$, $\boldsymbol{A} = (\boldsymbol{a}_1^\mathsf{T}, \ldots, \boldsymbol{a}_{100}^\mathsf{T})^\mathsf{T} \in \mathbb{R}^{100 \times n}$, $\boldsymbol{b} = (b_1, \ldots, b_{100}) \in \mathbb{R}^{100}$, $\varepsilon > 0$
2: **while** True **do**
3: $\quad \mathcal{I}(\boldsymbol{\theta}) \triangleq \{j | \boldsymbol{a}_j^\mathsf{T} \boldsymbol{\theta} > b_j\}$
4: $\quad$ **if** $\mathcal{I}(\boldsymbol{\theta}) = \emptyset$ or $\max_{j \in \mathcal{I}(\boldsymbol{\theta})} \frac{|\boldsymbol{a}_j^\mathsf{T} \boldsymbol{\theta} - b_j|}{\|\boldsymbol{a}_j\|} < \varepsilon$ **then**
5: $\quad\quad$ break
6: $\quad$ **end if**
7: $\quad$ choose $i \in \arg\max_{j \in \mathcal{I}(\boldsymbol{\theta})} \frac{|\boldsymbol{a}_j^\mathsf{T} \boldsymbol{\theta} - b_j|}{\|\boldsymbol{a}_j\|}$
8: $\quad \boldsymbol{\theta} \leftarrow \boldsymbol{\theta} - \frac{|\boldsymbol{a}_i^\mathsf{T} \boldsymbol{\theta} - b_i|}{\|\boldsymbol{a}_i\|^2} \boldsymbol{a}_i$
9: **end while**
10: **Return:** $\boldsymbol{\theta}$

---

**Metrics: IS.** Given a sample from the generator $\tilde{\boldsymbol{x}}_g \sim p_g$—where $p_g$ denotes the data distribution of the generator—IS uses the softmax output of the pre-trained network $p(\tilde{\boldsymbol{y}}|\tilde{\boldsymbol{x}}_g)$ which represents the probability that $\tilde{\boldsymbol{x}}_g$ is of class $c_i, i \in 1 \ldots C$, i.e., $p(\tilde{\boldsymbol{y}}|\tilde{\boldsymbol{x}}_g) \in [0, 1]^C$. It then computes the marginal class distribution $p(\tilde{\boldsymbol{y}}) = \int_{\tilde{\boldsymbol{x}}} p(\tilde{\boldsymbol{y}}|\tilde{\boldsymbol{x}}_g) p_g(\tilde{\boldsymbol{x}}_g)$. IS measures the Kullback–Leibler divergence $\mathbb{D}_{KL}$ between the predicted conditional label distribution $p(\tilde{\boldsymbol{y}}|\tilde{\boldsymbol{x}}_g)$ and the marginal class distribution $p(\tilde{\boldsymbol{y}})$. More precisely, it is computed as follows:

$$IS(G) = \exp\left( \underset{\tilde{\boldsymbol{x}}_g \sim p_g}{\mathbb{E}} \Big[ \mathbb{D}_{KL}\big( p(\tilde{\boldsymbol{y}}|\tilde{\boldsymbol{x}}_g) \| p(\tilde{\boldsymbol{y}}) \big) \Big] \right) = \exp\left( \frac{1}{\tilde{m}} \sum_{i=1}^{\tilde{m}} \sum_{c=1}^{C} p(y_c|\tilde{\boldsymbol{x}}_i) \log \frac{p(y_c|\tilde{\boldsymbol{x}}_i)}{p(y_c)} \right). \quad \text{(IS)}$$

It aims at estimating (i) if the samples look realistic i.e., $p(\tilde{\boldsymbol{y}}|\tilde{\boldsymbol{x}}_g)$ should have low entropy, and (ii) if the samples are diverse (from different ImageNet classes), i.e., $p(\tilde{\boldsymbol{y}})$ should have high entropy. As these are combined using the Kullback–Leibler divergence, the higher the score is, the better the performance.

**Metrics: FID.** Contrary to IS, FID compares the synthetic samples $\tilde{\boldsymbol{x}}_g \sim p_g$ with those of the training dataset $\tilde{\boldsymbol{x}}_d \sim p_d$ in a feature space. The samples are embedded using the first several layers of a pretrained classifier. It assumes $p_g$ and $p_d$ are multivariate normal distributions and estimates the means $\boldsymbol{m}_g$ and $\boldsymbol{m}_d$ and covariances $\boldsymbol{C}_g$ and $\boldsymbol{C}_d$, respectively, for $p_g$ and $p_d$ in that feature space. Finally, FID is computed as:

$$\mathbb{D}_{\text{FID}}(p_d, p_g) \approx \mathscr{D}_2\big((\boldsymbol{m}_d, \boldsymbol{C}_d), (\boldsymbol{m}_g, \boldsymbol{C}_g)\big) = \|\boldsymbol{m}_d - \boldsymbol{m}_g\|_2^2 + Tr\big(\boldsymbol{C}_d + \boldsymbol{C}_g - 2(\boldsymbol{C}_d \boldsymbol{C}_g)^{\frac{1}{2}}\big), \quad \text{(FID)}$$

where $\mathscr{D}_2$ denotes the Fréchet Distance. Note that as this metric is a distance, the lower it is, the better the performance.

**Hardware.** We used the Colab platform (https://colab.research.google.com/) and *Nvidia T4* GPUs.

## D   ADDITIONAL EXPERIMENTS AND ANALYSES

In this section, we provide complementary experiments associated with the three games introduced in the main paper: (2D-BG), (HBG), and (C-GAN). We also provide an additional study of the robustness of I-ACVI to bad conditioning by introducing a version of the (HBG) game, see § D.3 for more details.

### D.1   ADDITIONAL RESULTS FOR I-ACVI ON THE 2D-BG GAME

For completeness, in Fig. 5 we show the trajectories for the $x$ iterates—complementary to the $y$-iterates' trajectories depicted in Fig. 1 of the main part.

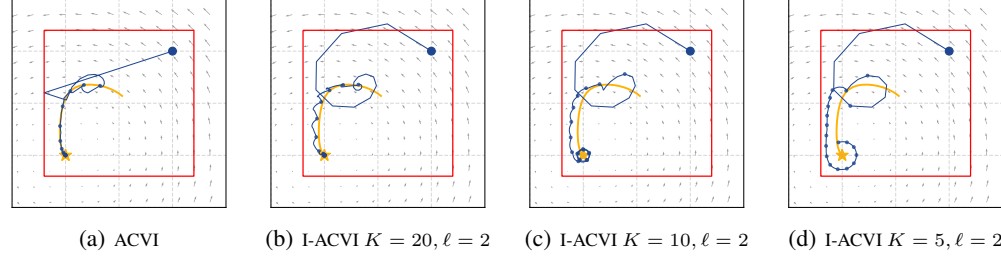

| (a) ACVI | (b) I-ACVI $K = 20, \ell = 2$ | (c) I-ACVI $K = 10, \ell = 2$ | (d) I-ACVI $K = 5, \ell = 2$ |

Figure 5: **Complementary illustrations to those in Fig. 1 of the main part: depicting here the trajectories of the $x$ iterates.** We compare the convergence of ACVI and I-ACVI with different parameters on the (2D-BG) problem while also depicting the central path (shown in yellow). Each subsequent bullet on the trajectory depicts the (exact or approximate) solution at the end of the inner loop (when $k \equiv K - 1$). The Nash equilibrium (NE) of the game is represented by a yellow star, and the constraint set is the interior of the red square.

**Comparison between ($\wp_1$) and ($\wp_2$).**   In Fig. 6 and Fig. 7 we show the trajectories of respectively the $y$ and $x$ iterates as we increase the learning rate. Increasing the learning rate increases the chance of crossing the standard log barrier, which makes the ($\wp_1$) undefined for such input, as the log function is not defined on the entire space. In contrast, the newly proposed barrier ($\wp_2$) is defined everywhere; thus, the $y$ iterates crossing the boundary of the constrained set does not make the ($\wp_2$) unstable and allows for convergence to the solution.

### D.2   ADDITIONAL RESULTS FOR PI-ACVI ON THE 2D-BG GAME

In this section, we provide complementary visualization to Fig. 2 in the main paper. We (i) compare with other methods in Fig. 8,9 and (ii) show PI-ACVI trajectories for various hyperparameters in Fig. 10.

**PI-ACVI vs. baselines.**   In Fig. 8 and 9, we can observe the behavior of projected gradient descent ascent, projected extragradient, projected lookahead, projected proximal point, mirror descent, and mirror prox on the simple 2D constrained bilinear game (2D-BG), we use the same learning rate of 0.2 for all methods except for mirror prox which is using a learning rate of 0.4. In Fig. 10 we show trajectories for PI-ACVI for $\ell \in \{1, 4, 10, 100\}$, $\beta = 0.5$, $K = 150$ and a learning rate 0.2.

### D.3   ADDITIONAL RESULTS ON THE HBG GAME

In this section, we (i) provide complementary experiments to Fig. 4 from the main paper, as well as (ii) analyze the robustness of I-ACVI against bad conditioning.

**CPU time to reach a given relative error.**   In Fig. 11 we extend the x-axis of Fig. 4.a from the main paper for I-ACVI. Unlike baselines, I-ACVI remains fast even when the target relative error is very small. This is due to the fact that I-ACVI uses cheaper approximate steps for lines 8 and 9 of Algorithm 1.

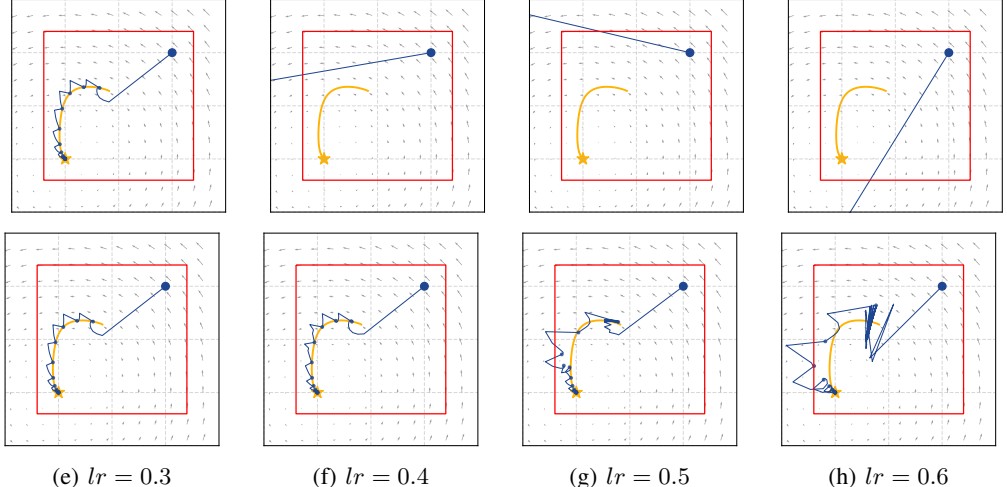

|  |  |  |  |
|---|---|---|---|
| (e) $lr = 0.3$ | (f) $lr = 0.4$ | (g) $lr = 0.5$ | (h) $lr = 0.6$ |

Figure 6: **I-ACVI trjectories for the $y$ iterates for different choices of learning rates $lr$. Top row:** Trajectories for the I-ACVI implementation using the standard barrier function ($\wp_1$). As the learning rate increases, the $y$ iterates cross the log barrier, breaking the convergence. **Bottom row:** Trajectories for the I-ACVI implementation using the new smooth barrier function defined over the entire domain ($\wp_2$). The extended barrier function we proposed is defined everywhere; thus, even if the iterates cross the standard barrier, the method converges, allowing for the use of larger step sizes. We can reduce the constant $c$ to improve the stability; we used $c = \{10, 1, 0.2, 0\}$ and $\gamma = \{0.3, 0.4, 0.5, 0.6\}$ for the learning rate.

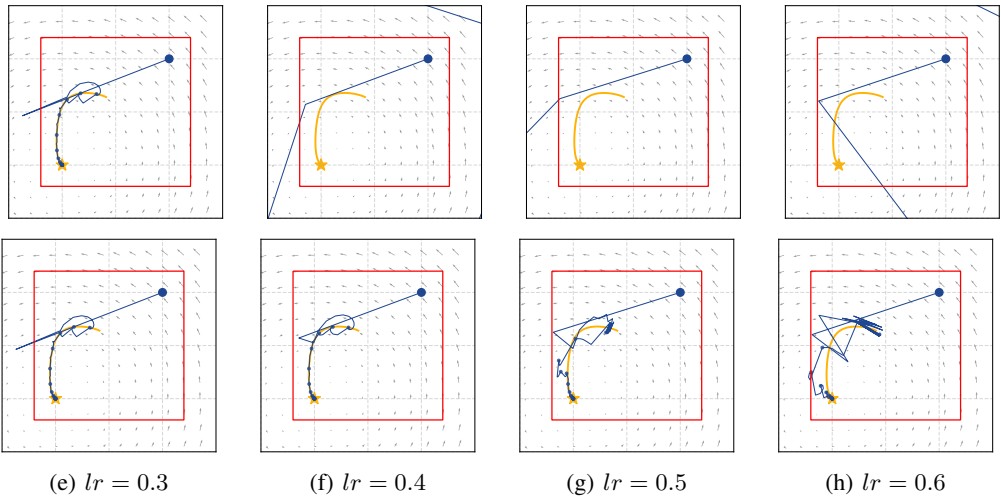

|  |  |  |  |
|---|---|---|---|
| (e) $lr = 0.3$ | (f) $lr = 0.4$ | (g) $lr = 0.5$ | (h) $lr = 0.6$ |

Figure 7: Complementary to Fig. 6, **I-ACVI trjectories for the $x$ iterates for different choices of learning rates $lr$. Top row:** Trajectories for the I-ACVI implementation using the standard barrier function ($\wp_1$). As the learning rate increases, the $y$ iterates (see Fig. 6) are crossing the log barrier, which breaks the optimization. **Bottom row:** Trajectories for the I-ACVI implementation using the new smooth barrier function defined over the entire domain ($\wp_2$). The iterates are allowed to cross the standard log barrier, which allows the $y$ iterates to recover from large steps. We can reduce the constant $c$ to improve the stability, we used $c = \{10, 1, 0.2, 0\}$, and $\gamma = \{0.3, 0.4, 0.5, 0.6\}$ for the learning rate.

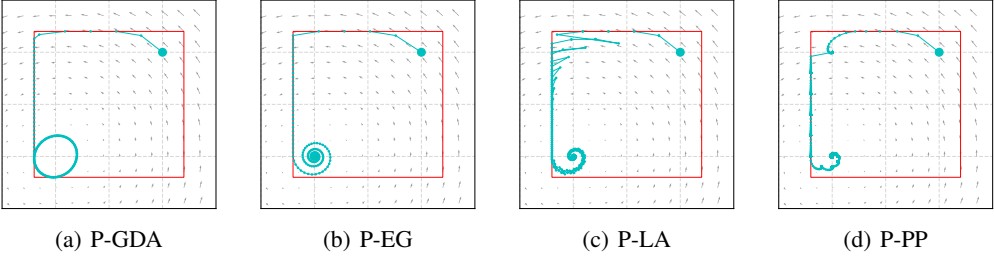

| (a) P-GDA | (b) P-EG | (c) P-LA | (d) P-PP |

Figure 8: **Comparison of Projected Gradient Descent Ascent (P-GDA), extragradient (P-EG) (Korpelevich, 1976), Lookahead (P-LA) (Chavdarova et al., 2021) and Proximal-Point (P-PP)** on the (2D-BG) game. For P-PP, we solve the inner proximal problem through multiple steps of GDA and use warm-start (the last PP solution is used as a starting point of the next proximal problem). All those methods progress slowly when hitting the constraint. Those trajectories can be contrasted with PI-ACVI in Fig. 10.

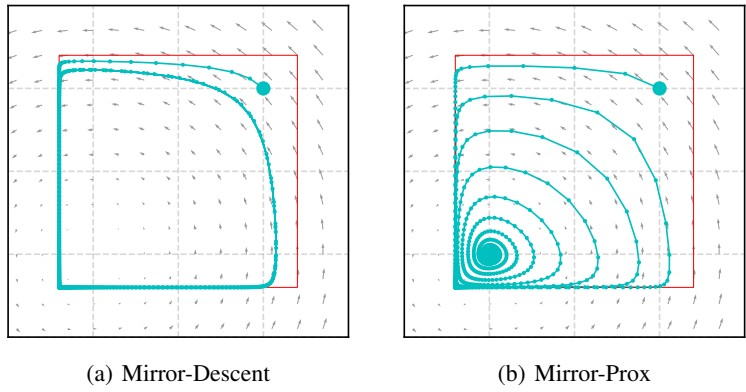

| (a) Mirror-Descent | (b) Mirror-Prox |

Figure 9: **Comparison of Mirror-Descent (MD) and Mirror-Prox (MP) on the (2D-BG) game.** Mirror-descent cycles around the solution without converging. Mirror-prox is converging to the solution. Both methods have been implemented using simultaneous updates and with a Bregman divergence $D_\Psi(x,y)$ with $\Psi(x) = -\frac{x+0.4}{2.8}\log(\frac{x+0.4}{2.8}) - (1 - \frac{x+0.4}{2.8})\log(1 - \frac{x+0.4}{2.8})$.

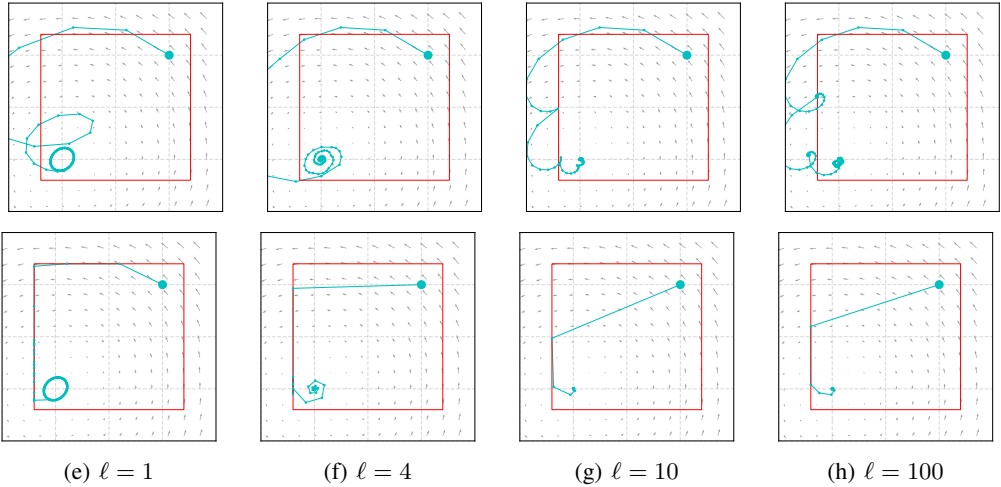

| (e) $\ell = 1$ | (f) $\ell = 4$ | (g) $\ell = 10$ | (h) $\ell = 100$ |

Figure 10: **PI-ACVI (Algorithm 2) for different choices of $\ell$. Top row:** Trajectories for the $x$ iterates. **Bottom row:** Trajectories for the $y$ iterates. For $\ell = 1$, the trajectory for the $y$ iterates is similar to the one of P-GDA (see Fig. 8), as we increase $\ell$ we observe how relatively few iterations are required for convergence compared to baselines.

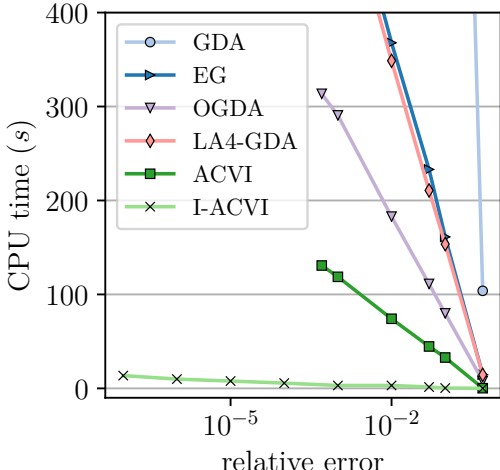

Figure 11: **Comparison between I-ACVI and other baselines used in § 5** of the main part. CPU time (in seconds; y-axis) to reach a given relative error (x-axis); while the rotational intensity is fixed to $\eta = 0.05$ in (HBG) for all methods. I-ACVI is much faster to converge than other methods, including ACVI.

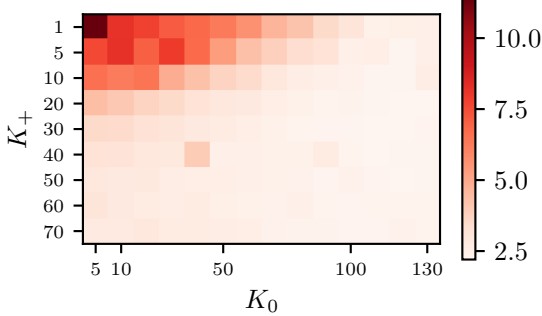

Figure 12: **Impact of $K_0$:** joint impact of the number of inner-loop iterations $K_0$ at $t = 0$, and different choices of inner-loop iterations for $K_+$ at any $t > 0$, on the CPU-time needed to reach a fixed relative error of $10^{-4}$. A large enough $K_0$ can compensate for a small $K_+$.

**Impact of $K_0$.** In Fig. 12 we show, for each $(K_0, K_+)$ the CPU time required to reach a relative error of $10^{-4}$. Those times are highly correlated with the number of iterations shown in Fig. 4.c of the main paper.

**Comparison with mirror-descent and mirror-prox.** In Fig. 13 extend the experiments of Fig. 4.b of the main paper to include the mirror-descent (MD) and mirror-prox (MP) methods described in App. A.5.

**Impact of conditioning.** We modify the (HBG) game to study the impact of conditioning. Hence, we propose the following version:

$$\min_{\boldsymbol{x}_1 \in \triangle} \max_{\boldsymbol{x}_2 \in \triangle} \boldsymbol{x}_1^\intercal \boldsymbol{D} \boldsymbol{x}_2 \,, \qquad\qquad \text{(HBG-v2)}$$

$$\triangle = \{\boldsymbol{x}_i \in \mathbb{R}^{500} | \boldsymbol{x}_i \geq \boldsymbol{0}, \text{ and }, \boldsymbol{e}^\intercal \boldsymbol{x}_i = 1\}, \text{ and } \boldsymbol{D} = \text{diag}(\alpha_1, \dots, \alpha_{500}) \,.$$

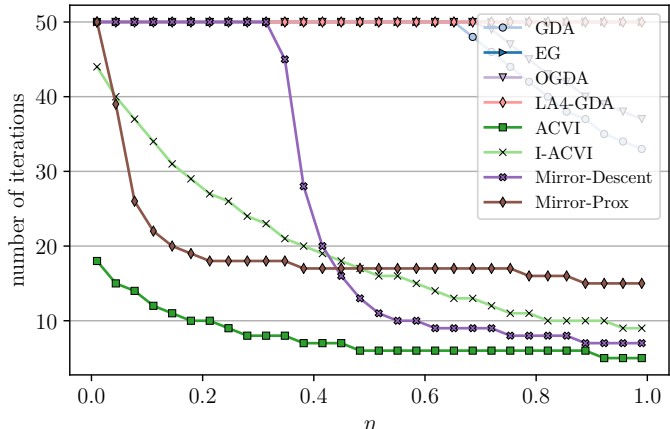

Figure 13: Number of iterations to reach a relative error of 0.02 for varying values of the rotational intensity $\eta$ ($x$-axis). We fixed the maximum number of iterations to 50. For mirror-descent and mirror-prox, we used the KL-divergence as $D_\Psi(x, y)$ and use large step sizes of respectively $\gamma = 500$ and $\gamma = 280$. When the rotational intensity is strong (small $\eta$), mirror-descent fails to converge within the 50 iterations budget. However, when $\eta$ is large, mirror descent converges much faster than GDA, EG, OGDA, and LA4-GDA. Mirror-prox is better than mirror-descent at handling strong rotational intensities but is slowed down when the game is mostly potential. In comparison, ACVI converges after a small number of steps regardless of $\eta$.

The solution of this game depends on the $\{\alpha_i\}_{i=1}^{500}$:

$$
\boldsymbol{x}_1^\star = \boldsymbol{x}_2^\star = \frac{1}{\sum_{i=1}^{500} 1/\alpha_i} \begin{pmatrix} 1/\alpha_1 \\ 1/\alpha_2 \\ \vdots \\ 1/\alpha_{500} \end{pmatrix}
$$

We define the conditioning $\kappa$ as the ratio between the largest and smallest $\alpha_i$: $\kappa \triangleq \frac{\alpha_{\min}}{\alpha_{\max}}$. In our experiments we select $\alpha_i$ linearly interpolated between 1 and $\alpha_{\max}$ (e.g. using the `np.linspace(1,a_max,500)` NumPy function). We set $\alpha_{\min} = 1$ and vary $\alpha_{\max} \in \{1, 2, 3, 4, 5, 6, 7, 8, 9, 10\}$. We compare projected extragradient (P-EG) with I-ACVI. For P-EG, we obtained better results when using smaller learning rates $\gamma$ for larger $\alpha_{\max}$: $\gamma = 0.3 \times 0.9^{\alpha_{\max}}$. For I-ACVI we set $\beta = 0.5$, $\mu = 10^{-5}$, $\delta = 0.5$, $\gamma = 0.003$, $K = 100$ and $T = 200$. We vary $\ell$ depending on $\alpha_{\max}$: $\ell = 20$ for $\alpha_{\max} \in \{1, 2, 3\}$, $\ell = 50$ for $\alpha_{\max} \in \{4, 5, 6\}$, and $\ell = 100$ for $\alpha_{\max} \in \{7, 8, 9, 10\}$. We compare the CPU times required to reach a relative error of 0.02 in Fig. 14. We observe that I-ACVI is more robust to bad conditioning than P-EG. As $\kappa \to 0$, P-EG fails to converge in an appropriate time despite reducing the learning rate. For I-ACVI, keeping the same learning rate and only increasing $\ell$ is enough to compensate for smaller $\kappa$ values. One can speculate that I-ACVI is more robust thanks to (i) the $\boldsymbol{y}$-problem (line 9 in Algorithm 1) not depending on $F(x)$, hence being relatively robust to the problem itself, and (ii) the $\boldsymbol{x}$-problem (line 8 in Algorithm 1) being "regularized" by $\boldsymbol{y}_k$ and $\boldsymbol{\lambda}_k$.

## D.4 Additional results on the C-GAN game

This section shows complementary results to our constrained GAN MNIST experiments. In Fig. 15, we further show the impact of $\ell_0$ on the convergence speed by training different PI-ACVI models with $\ell_0 \in \{20, 50, 100, 200, 400, 600, 800, 1000\}$, all other hyperparameters being equal — setting $\ell_+ = 10$. We compare in Fig. 16 the obtained curves for $\ell_0 = 400$ with projected-GDA (P-GDA), and verify that — similarly to Fig. 3 of the main paper for which $\ell_+ = 20$ — PI-ACVI is here as well outperforming significantly P-GDA. This shows that PI-ACVI is relatively unaffected by $\ell_+$ as opposed to $\ell_0$.

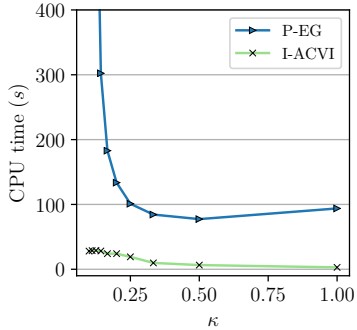

Figure 14: **Experiment on conditioning:** CPU time to reach a relative error of $0.02$ on the (HBG-v2) game, for different conditioning values $\kappa$. While P-EG struggles to converge when the conditioning is bad (small $\kappa$), I-ACVI, on the other hand, can cope relatively well.

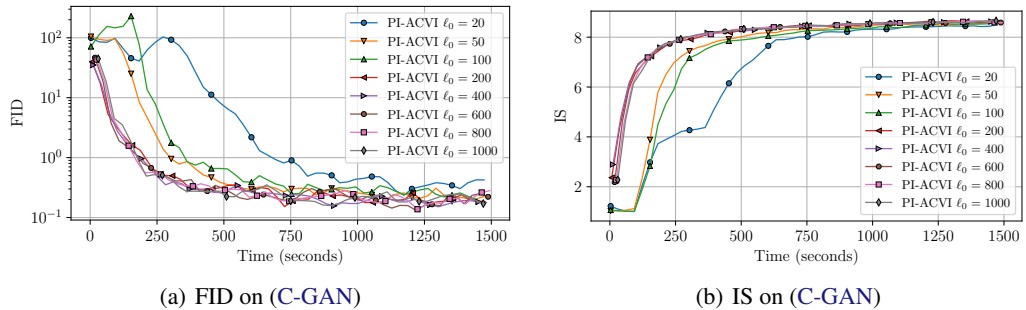

(a) FID on (C-GAN)          (b) IS on (C-GAN)

Figure 15: **Effect of $\ell_0$ on FID and IS:** On the MNIST datasets, comparison of various runs of PI-ACVI for different $\ell_0$. All other hyperparameters are equal: $\ell_+ = 10$, $\beta = 0.5$, see § C for more details. **(a) and (b):** we observe the importance of $\ell_0$, despite $\ell_+ = 10$ being relatively small we still converge fast to a solution — in terms of both FID ($\downarrow$) and IS ($\uparrow$) — given $\ell_0$ large enough. All curves are obtained by averaging over two seeds.

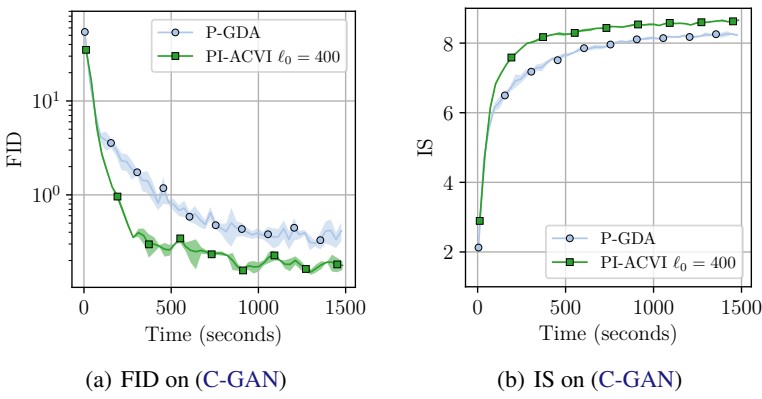

(a) FID on (C-GAN)          (b) IS on (C-GAN)

Figure 16: **PI-ACVI vs. P-GDA on** (C-GAN) **MNIST:** On the MNIST datasets, comparison of P-GDA and PI-ACVI. For PI-ACVI, we set $\ell_0 = 400$ and $\ell_+ = 10$. **(a) and (b):** in both FID ($\downarrow$) and IS ($\uparrow$), PI-ACVI converges faster than P-GDA. The difference with Fig. 3 from the main paper is that we use $\ell_+ = 10$ instead of $\ell_+ = 20$. This shows that PI-ACVI is relatively robust to different values of $\ell_+$.

