# OpenReview forum: "A Primal-Dual Approach to Solving Variational Inequalities with General Constraints"
_ICLR.cc/2024/Conference — ICLR 2024 poster_

### Official Review · Reviewer_uGBX · 2023-10-30

**Soundness:** 3 good
**Presentation:** 3 good
**Contribution:** 3 good
**Rating:** 6
**Confidence:** 2

**Summary:**

Previous research has shown that first-order methods can be used to solve general variational inequality (VI) problems under the assumption that analytic solutions for the subproblems are available. This paper circumvents this assumption by using a warm-starting technique. The authors prove the convergence rate of the gap function for the last iteration of both the proposed exact ACVI and inexact ACVI methods. Additionally, the authors discuss scenarios where the inequality constraints are quick to project onto and introduce an alternative version of ACVI, confirming its convergence under identical conditions.

**Strengths:**

S1: The authors prove that the gap function of the last iterate of ACVI decreases at a rate of $O(1/\sqrt{K})$ for monotone VIs, without depending on the assumption that the operator is L-Lipschitz.

S2: The authors establish the convergence rate for both the exact ACVI and the inexact ACVI methods.

S3: The authors discuss the special case of inequality constraints that are quick to project onto.

S4: Experiments have been conducted to demonstrate that inexact ACVI can outperform other methods, such as the lookahead-Minmax algorithm, by advantageously solving linear systems implicitly beforehand (as shown in Steps 3 and 4 of Algorithm 1).

**Weaknesses:**

W1. The assumption that F(x) is monotone or strictly monotone may be overly stringent. This condition is not met in applications involving Generative Adversarial Networks.

W2. The actual efficiency of the algorithm heavily depends on the solution approach for the x-subproblem, which is challenging to solve. The authors do not address the resolution of this general nonlinear system.

W3. It is preferred to provide an explanation for assuming the "monotonicity" of $F$ on $\mathcal{C}_{=}$ and the "strictly monotonicity" of $F$ on $\mathcal{C}$.

W4. The barrier map is undefined when it first appears in Algorithm 1.

**Questions:**

See above.

---

> ### Author Response · Authors · 2023-11-20
> **Response to Reviewer uGBX**
>
> Thanks for the time you took to review our paper and provide feedback.
>
> > W1. The assumption that $F(x)$ is monotone or strictly monotone may be overly stringent. This condition is not met in applications involving Generative Adversarial Networks.
>
> Monotone variational inequalities (VIs) are crucial in the VI literature and are considered a relatively broad problem class, surpassing min-max problems, for example. Their significance mirrors that of convex optimization problems in minimization literature, where optimization methods are initially validated on convex problems due to the complexity of studying non-convex problems.
> Hence, it is essential and customary to initially examine the convergence properties of new algorithms on monotone VI problems, as evidenced by existing works (see [1-4], for instance). Extending our analysis to a broader class is left for future research.  Additionally, our experiments with GANs (Figure 3) suggest that ACVI performs effectively even in non-monotone problems.
>
> > W2. The actual efficiency of the algorithm heavily depends on the solution approach for the x-subproblem, which is challenging to solve. The authors do not address the resolution of this general nonlinear system.
>
> In terms of our theoretical contributions, our convergence rate does take into account the complexity of solving the $x$-subproblem. In particular, if either EG or GD are used for the $x$-subproblem, the convergence is $O(1/\sqrt{K})$. It's worth noting that if the original problem is monotone, the $x$ problem becomes strongly monotone, making it easier to solve; you can find more details in our discussion on page 51.
> Additionally, if general-form constraints are associated with the problem, there is no other alternative as a first-order method, as we elaborate in the introduction.
>
> From a practical standpoint, solving the $x$-subproblem approximately using unconstrained GDA or ED methods is straightforward—this is reflected in our experiments, which include wall-clock time and gradient query comparisons. Setting the iteration number for solving $x$-subproblem (i.e., $l_x^{(t)}$ in line 8 of Alg. 1)  to be a small constant yields good results. Refer to Figures 2-4, 11, and App. E for more details.
>
> Please let us know if further clarification on the theoretical or practical aspects is needed.
>
> > W3. It is preferred to provide an explanation for assuming the "monotonicity" of $F$ on $C_=$  and the "strictly monotonicity" of $F$ on $C$.
>
>
> The monotonicity of $F$ on $C_=$ guarantees the existence and uniqueness of the solution of the non-linear system for the $x$-subproblem. More details can be found in Theorem B.3 in our paper or Theorem 1 in Yang et al., 2023.
>
> We require either strict monotonicity of $F$ or strict convexity of one of $\varphi_i$. This condition is necessary to ensure the "central path" exists—refer to Appendix B.3 for a more detailed discussion. It's worth noting that Remark 3 in Yang et al. explains that this assumption is relatively weak.
>
> Please let us know if more clarifications are needed.
>
>
> > W4. The barrier map is undefined when it first appears in Algorithm 1.
>
>
> We've postponed the definition of the barrier maps to the convergence subsection to ensure clarity that the theorem specifically applies to those barrier maps (which are clickable in the algorithm). Our revised paper also refers to their equations when introducing Algorithm 1 in Section 3.  Thanks for bringing this to our attention.
>
> ---
>
> [1] Korpelevich. The extragradient method for finding saddle points and other problems. Matecon, 1976.
>
> [2] Mokhtari et al. Convergence Rate of O(1/K) for Optimistic Gradient and Extragradient Methods in Smooth Convex-Concave Saddle Point Problems. SIAM Journal on Optimization, 2020.
>
> [3] Mokhtari et al. A Unified Analysis of Extra-gradient and Optimistic Gradient
> Methods for Saddle Point Problems: Proximal Point Approach. AISTATS, 2020.
>
> [4] Daskalakis et al. The limit points of (optimistic) gradient descent in min-max optimization. NeurIPS, 2018.

---

### Official Review · Reviewer_MAK9 · 2023-11-04

**Soundness:** 3 good
**Presentation:** 3 good
**Contribution:** 2 fair
**Rating:** 6
**Confidence:** 4

**Summary:**

The present paper considers the problem of solving a variational inequality with constraints. Recall that a solution $x_\star$ to variational inequality given by $F:\mathcal{C}\to\mathbb{R}^n$ (with $\mathcal{X}\subset \mathbb{R}^n$) satisfies
$$\forall x\in \mathcal{C}\,:\,\langle F(x_\star),x-x_\star\rangle\geq 0.$$
The paper considers monotone VIs, which include convex optimization ($F=\nabla f$ for $f$ convex) and minimal convex/concave games as special cases. The set $\mathcal{C}$ is given by linear equality and convex inequality constraints.

A standard first-order approach to this problem, such as Korpelevich's extragradient method, would require lots of projection steps to $\mathcal{C}$. Since these may be quite expensive, Yang, Jordan and Chavdarova (ICLR 2023) considered a first-order interior point method. Their approach is based on a rewriting of the constrained VI problem using Lagrangian duality, which leads to an alternating direction method of multipliers (ADMM). Those authors show that their method works (with $O(1/\sqrt{K})$ last-iterate error after $K$ iterations) only under assumptions that are significantly stronger than mere monotonicity (for instance, $F$ is assumed t be Lipschitz).

The present paper strengthens Yang et al's analysis in different ways. First, the algorithm by Yang et al. is shown to converge at the same rate previously shown under the sole assumption of monotonicity (plus strict monotonicity *or* strict convexity of one of the constraints). They also show that, if $F$ is Lipschitz, one may use *approximate* solvers for the optimization problems in each inner loop of the interior point method, while keeping the same convergence rate. Finally, when the constraints are "simple", a projection based method for the inner loop may also be used, with similar assumptions and results. The paper also presents a short empirical study suggesting that the methods given are accurate and faster than the "competition".

**Strengths:**

The paper greatly increases the scope of application of first order methods in the solution of constrained VIs. In particular, it shows that such methods can be fast, while achieving similar or better theoretical guarantees than the "competition"

**Weaknesses:**

The main issue I had in reading the paper was the comparison with Yang et al. I tried to follow the long proofs in that paper and to compare them with the present proofs. Many of the steps were similar, but I was still left wondering where the main improvements took place.

Given this, I have been somewhat conservative in my "contribution" score, but I am perfectly willing to reconsider it in the light of the answer to my question (see below).

___

A small typo: in the first two displays of page 23, the norms in the RHS are missing their squares (the mistake is fixed in the next display).

**Questions:**

In the proof of Theorem 3.1 (including the preceding Lemmas), can you pinpoint exactly where you manage to circumvent the strict/strong monotonicity assumption of Yang et al? It seems to me that equation (14) is one of the crucial steps: it avoids Lemma 1 in Yang et al, where the gap is bounded as a function of the distance to the optimal VI solution. However, I struggle to see where/how exactly the improvements were achieved.

---

> ### Author Response · Authors · 2023-11-20
> **Response to Reviewer MAK9**
>
> Thank you for your attentive review of the proofs and for providing valuable feedback – it is greatly appreciated. We've corrected the mentioned typo in our revised paper; thanks for catching that.
>
> We managed to circumvent the assumptions in Yang et al. by a simple yet not immediately apparent insight into understanding the point implicitly tracked by ACVI at each iteration. Indeed, (14) is one of the crucial steps -- we added more technical discussion below Lemma C.10 (paragraph *discussion*, page 28) in our revised paper,  where we describe our new analysis technique and compare them with the approach of Yang et al., 2023. Please let us know if additional clarification could be helpful. Thanks.
>
> Concerning the remaining proofs of the theorems, (i) addressing the main challenge of Thm. 3.2 involves handling the inherent approximation errors in our inexact algorithm. Meanwhile, (ii) the overall approach for proving Thm. 4.1. differs from the previous one for Thm. 3.1. since there are no central path points in Algorithm 2.

---

### Official Review · Reviewer_VTND · 2023-11-05

**Soundness:** 4 excellent
**Presentation:** 3 good
**Contribution:** 3 good
**Rating:** 8
**Confidence:** 3

**Summary:**

Variational inequality problems are a class of numerical problems that generalize constrained optimization problems. The rough idea is to use the first-order optimality characteization, and replace the gradient $\nabla f(x)$ with some other more general function $F(x)$. These problems arise naturally when considering equilibrium computation or robust modifications of ordinary constrained optimization problems. The authors study the ACVI algorithm, an ADMM-type algorithm for this setting, and show that it converges under weaker assumptions compared to prior work, specifically under generalization of convexity to the variational inequality setting without Lipschitz-smoothness-type assumptions used previously. They also show this works even if certain steps within the respective algorithm are performed approximately rather than exactly, and use this to study how warm-start techniques affect those steps. The argument is reminiscent of Lyapunov analysis, and works by exhibiting a certain gap function which is shown to decrease at every iteration, revealing part of why ACVI behaves this way. The developed theory is illustrated on somewhat standard experiments.

**Strengths:**

This is a good paper. This review is going to be relatively short because the contribution is good, the paper is well-written, and everything makes sense.
* Understanding the behavior of optimization algorithms and their generalizations in cases where smoothness assumptions are relaxed is a clear and well-motivated class of research questions, and many papers in convex optimization have been written about this, which overtime have significantly improved understanding of the field and led to advances. It makes a lot of sense to also do this for more general settings.
* The paper as a whole and introduction in particular is clear, well-written, and makes sense. The authors do an excellent job at drawing attention to the aspects that distinguish variational inequalities from ordinary optimization, including limit-cycle-type behavior often found when solving for equilibria and similar problems. that monotonicity plays the role of convexity, and similar points.that someone who is familiar with optimization but not with variational inequalities - perhaps a significant fraction of readers  would want to know.
* There is not enough work on algorithms for equilibrium-solving. The fact that algorithms for doing so are underdeveloped compared to gradient-based optimization is one of the big reasons why GANs have declined in favor of diffusion models and other approaches. Having better tools could help unlock technical tools for problems like this, and for getting multi-agent reinforcement learning algorithms to work more reliably, which is an important goal for machine learning.
* The algebra in Appendix C, which is used to prove the results, is rather heavy: makes me glad that the authors have taken the effort to prove the necessary results, as it means that somebody else that wants to use the algorithm or a similar-enough variant doesn't have to. I'd imagine someone working on optimization would find it instructive to go through the appendix to see what kind of tricks are used, especially in comparison with more ordinary analysis of convex optimization algorithms.

**Weaknesses:**

* Experiments are somewhat limited. The only examples are synthetic bilinear games, and GANs. It would be nice to also have an example motivated by computational game theory, or by multi-agent reinforcement learning, or applications to economics, where equilibrium-finding algorithms that are a special case of the variational inequality framework are important.
* Figure 1 needs a legend, because right now it is confusing. I understand the red square is the constraint set and the yellow star is the Nash from the caption: what's the difference between the yellow and blue line? Which one is the initial point? It would save my time and improve clarity to have this labeled in full on the actual plot rather than partially explained in a manner that is buried inside the caption which has a lot more details that I may or may not be ready to delve into. The vector fields are also way too small, I can't tell on my display which direction any of the arrows are pointing.
* Please label all equations, not just the ones you think are important. When discussing technical papers with other people over Zoom, not labeling everything makes it needlessly difficult to communicate precise equation lines. Often, I end up saying "the 3rd equation on page 4" and then my collaborator is confused because they think I am counting by multi-line equation rather than per-line, or they think I am counting from the top rather than the bottom. This waste of time is extremely annoying and the outdated convention of "not labeling important equations to reduce clutter (actually, to reduce ink and paper costs, which no longer exist in the digital world)" needs to go away.
* Font size in figures is a bit small, making it bigger would help see what is happening.
* Notation is not always completely consistent, for instance min and argmin are sometimes italicized, sometimes not.

**Questions:**

Can you comment a bit further on the role of the barrier function in ACVI, and in the theory of Section 3.2? The barrier function is already a part of ACVI, and therefore not a new component of inexact ACVI - is this correct? If so, why talk about it in detail before Theorem 3.2, rather than in the background section? Does Theorem 3.2 specifically require one of these two barrier functions, and would not work with others? If so, this should be stated more explicitly - right now it's buried in the definition of Algorithm 1, and one has to unroll that definition - including the line above saying "Choosing among these is denoted with $\wp$" - to realize this, which isn't great because it puts a key part of the logic outside of both the text which defines Theorem 3.2 and the text that defines Algorithm 1.

Appendix typo: "subproblems and definitions" -> "Subproblems and definitions" (missing caps)

---

> ### Author Response · Authors · 2023-11-21
> **Authors' response to Reviewer VTND**
>
> Thank you for reviewing our paper. We appreciate your unique and insightful summary.
>
> > Additional experiments
>
> Thanks; integrating those experiments could indeed help highlight the value of our contributions. However, due to space limitations, we opt for the standard experiments commonly found in related works. To complement our numerical simulations, we include experiments on GANs—a less conventional practice in theoretical papers.
>
>
> It's worth noting that the bilinear games we employ have been extensively explored in computational game theory, reinforcement learning, and economics, as indicated by references [1-5]. The high-dimensional bilinear games, coupled with the probability simplex constraints (problem (HBG) in Section 5), are reminiscent of two-player zero-sum games [5]. In each turn, players select actions from predetermined probability distributions, akin to defining their policies. The effectiveness of our methods in addressing this particular problem subtly implies that they may perform well in the mentioned areas as well.
>
> > W2: Fig. 1
>
> We've edited Fig. 1 in the uploaded revised version; thanks for raising that. The yellow curve is the central path, or in other words, the trajectory that the second-order interior point methods would follow. The dark blue trajectory is that of the corresponding method (in the subfigure's caption), where all methods are initialized at the same point depicted with a dark blue circle.
>
>
> > W3: labeling all equations  \& W4: The figures' font size
>
> Thanks for pointing that out; we will improve these for the final version.
>
> > W5: min and argmin
>
> We've corrected that; thanks.
>
> > Whether the barrier function is a new component of inexact ACVI
>
> In short, $\wp_1(\cdot)$ is the same as in Yang et al., and $\wp_2(\cdot)$ is new for I-ACVI.
>
>
> In Yang et al., it is assumed that the $y$ subproblem can be solved exactly analytically.  In contrast, in I-ACVI, where the solution is obtained iteratively through gradient-based methods, we introduced $\wp_2$ to make the objective nicer. Specifically, $\wp_1(\cdot)$ is non-smooth, and while technically functional, there are no efficient methods to solve it.  We design $\wp_2(\cdot)$ to make the objective smooth;  this barrier term could potentially be helpful also for constrained minimization.  Please see App. C.4 for a more detailed discussion.
>
>
> > Does Theorem 3.2 specifically require one of these two barrier functions?
>
> Yes. Theorem 3.2 requires one of these two barrier functions and would not work with others; in addition to Alg. 1, we noted it in Theorem 3.2, too.
>
>
> > subproblems and definitions caps
>
> Fixed in the updated version.
>
> ---
>
> [1] Hwang et al. Strategic decompositions of normal form games: Zero-sum games and potential games. Games and Economic Behavior, 2020 - Elsevier.
>
> [2] Kannan et al. Games of fixed rank: A hierarchy of bimatrix games. Economic Theory, 2010 - Springer.
>
> [3] Campana et al. Nash equilibria and bargaining solutions of differential bilinear games. Dynamic Games and Applications. 2021 - Springer.
>
> [4] Sherali et al. A new reformulation-linearization technique for bilinear programming problems. Journal of Global optimization, 1992 - Springer.
>
> [5] Cen et al. Fast Policy Extragradient Methods for Competitive Games with Entropy Regularization. NeurIPS 2021.

---

> ### Comment · Reviewer_VTND · 2023-11-22
>
> Thanks very much for your rebuttal! I have no further questions. My main concern is that the font size in the figures is still too small. Please make it at least as big as the font size in the document itself.

---

### Official Review · Reviewer_mjvt · 2023-11-09

**Soundness:** 2 fair
**Presentation:** 3 good
**Contribution:** 2 fair
**Rating:** 6
**Confidence:** 4

**Summary:**

The paper studies how to use first-order methods to solve the variational inequality with general constraints. The authors first show the last iterate convergence of the ACVI method in Yang et al. (2023) without assuming the Lipschitzness of the operator. Then, the authors show similar convergence of an inexact version of ACVI if the errors of subproblem solvers decrease at proper rates. When the inequality constraints are fast to project, the authors show a simplified ACVI and its convergence. Experiments are provided to show the convergence and less computational time.

**Strengths:**

- The paper is well written, and all results are supported by proofs.

- The authors prove the same performance guarantee for ACVI under less restrictive conditions. A new analysis technique is developed to remove the Lipschitz assumption.

- It is also important to consider inexact ACVI that takes warm-start technique, which provides a more practical algorithm than the original. The authors provide the same convergence rate by conditioning the quality of approximation solutions, which is a useful guide.

-  Some experiments on standard bilinear games and GANs are provided to show convergence.

**Weaknesses:**

- The last iterate convergence is characterized by the gap function and the difference of two primal iterates. This metric is different from the optimality gap used in Yang et al. (2023). I am not sure if they are comparable.

- The algorithm design has a great similarity with ACVI in Yang et al. (2023). I would expect much more on highlights on analysis techniques, which seem to be not formally presented in the main paper.

- It is less discussed why removing the Lipschitz assumption is important in theory and practice.

- The method is limited to monotone operators.

**Questions:**

- Since this paper focuses on improving existing analysis, can the authors more formally state new analysis techniques compared with Yang et al. (2023)?

- How can the errors of solving subproblems be controlled in practice?

- What is the computational complexity of inexact ACVI? How can this be compared with the one for ACVI in your experiments?

- Any examples that lack Lipschitzness? and experiments?

- Can the authors generalize inexact ACVI to more general operators? What would be failure if not?

- Have you compared this work with last-iterate or non-ergodic convergence results in the constrained optimization literature?

**Details Of Ethics Concerns:**

none.

---

> ### Author Response · Authors · 2023-11-20
> **Response to Reviewer mjvt (1/2)**
>
> Thanks a lot for your time reviewing our paper and for your feedback.
>
> > W1: convergence metric relative to Yang et al. (2023)
>
> We use the standard gap function as in Yang et al. (2023); see our def. 2.2, and def. 2 in Yang et al., as well as item 3 in Thm.2 and Thm.3 of Yang et al.(2023).
>
> > W2: more highlights on analysis techniques relative to Yang et al. (2023)
>
> The relaxed assumptions relative to Yang et al. (Thm. 3.1) are owning to a simple but helpful insight of identifying a relationship between the reference point of the gap function and a KKT point that ACVI implicitly targets, as we mention in the last paragraph on page 2. We also added a more formal discussion on page 27 to discuss this further; thanks.
> The remaining theorems' proofs differ from Yang et al. (2023) as Thm. 3.2 deals with approximation errors of our inexact algorithm and the proof of Thm. 4.1 differs in that we do not have central path points.
>
> > W3: Why removing the Lipschitz assumption is important
>
> Practical scenarios often entail operators lacking Lipschitz continuity, such as in GANs and multi-agent reinforcement learning. For example, if using relu (or a lot of other activations), the successive multiplication of coefficient matrices leads to a derivative that is not Lipschitz continuous. Relaxing this assumption broadens the scope for which the convergence guarantee holds. It's worth noting that a substantial body of work in both VI and minimization literature focuses on eliminating the Lipschitzness assumption; references [1-5], for instance.
>
> > W4. The method is limited to monotone operators.
>
> Monotone variational inequalities (VIs) occupy a pivotal position in the VI literature and are considered a relatively broad problem class, surpassing min-max problems, for example. Their significance is comparable to that of convex optimization problems in the realm of minimization literature, where, due to the complexity of non-convex problems, optimization methods are initially validated on convex problems.
> Hence, it is essential and customary to initially examine the convergence properties of new algorithms on monotone VI problems, as evidenced by existing works (see [6-9], for instance). Extending our analysis to a broader class is left for future research.
>
> Our experiments with GANs (Figure 3) indicate that ACVI performs effectively even in non-monotone problems.
> > Q1. Since this paper focuses on improving existing analysis, can the authors more formally state new analysis techniques compared with Yang et al. (2023)?
>
> Please refer to our earlier answer to W2.
> In addition to Thm. 3.1, which improves the mentioned analysis, we introduce two algorithms and study their convergence. This is in response to the limitation of the alg. in Yang et al. (2023) in handling the general case when the subproblem cannot be analytically solved and efficiently managing structured inequality constraints.
>
> > Q2. How can the errors of solving subproblems be in practice?
>
> In our experiments, we found that the errors of solving subproblems can be controlled simply by fixing $l_x^{(t)}$ and $l_y^{(t)}$ to be a small constant number $l$ in lines $8$ and $9$ of Alg. 1. This entails running $l$ iterations of $\mathcal{A}_x$ and $\mathcal{A}_y$ each time to solve the subproblems. To provide specific examples, in the 2D-BG games in Figures 1 and 2, we set $l$ to be $2$ and $20$, respectively. In the C-GAN experiments (Figure 3, Section 5.3), we use a large $l$ for the first iteration ($l_0$) (the warm-start technique) and then fix $l=20$. See App. C.4, where we elaborate on this.
>
> > Q3. What is the computational complexity of inexact ACVI? How can this be compared with the one for ACVI in your experiments?
>
> To give an $\varepsilon$-accurate solution, the overall complexity of inexact ACVI is still $\mathcal{O}(1/\varepsilon^2)$ up to a log factor, as we discussed at the end of App. C.4. In our experiments, the complexity is also $\mathcal{O}(1/\varepsilon^2)$ since we fix the number of iterations to solve the subproblems to be a small constant number, which only adds constant to the complexity.

---

> > ### Author Response · Authors · 2023-11-20
> > **Response to Reviewer mjvt (2/2)**
> >
> > > Q4. Any examples that lack Lipschitzness? and experiments?
> >
> > For example, the operators of Von Newmann's ratio game and Frasaken game study in Yang et al., 2023 (see Figure 2 and App. D therein) are not Lipschitz continuous.
> > Operators lacking Lipschitzness are ubiquitous when using neural nets, such as in GANs and multi-agent reinforcement learning (albeit non-monotone); please see Figure 3 and Section 5.3 for our experiments on GANs.
> >
> > > Q5. Can the authors generalize inexact ACVI to more general operators? What would be failure if not?
> >
> > For general operators (without structural assumptions), finding the optimal solutions is NP-hard. Studying the convergence of our methods on some broader or different structured non-monotone operators is beyond the scope of this work. Please also see our above response on W4.
> >
> > > Q6. Have you compared this work with last-iterate or non-ergodic convergence results in the constrained optimization literature?
> >
> > Certainly, we first compare our work with both first-order and second-order methods in the constrained optimization literature from a theoretical aspect in the introduction. In short, (i) prior first-order methods could only handle problems with simple constraints and lack last-iterate convergence analysis for general monotone VIs, whereas (ii) second-order methods can handle general constraints but are computationally intractable for high-dimensional problems. Regarding the former, our convergence rates match the known lower bound.
> >
> > In our experiments, we compare our methods with commonly used first-order methods, such as projected GDA/EG on the C-GAN and HBG games, see Figures 3 and 4. Here, we did not compare with the second-order methods because these problems are high-dimensional, and second-order methods are computationally intractable. We also compare our methods with the projected proximal-point (PP) method and Mirror-Descent (MD) and Mirror-Prox (MP) in App. E, see Figure 8-10, Figure 13 for example (albeit these methods cannot be used for general-form constraints).
> >
> >
> > ---
> >
> > [1] Denisov et al. Convergence of the Modified Extragradient Method for Variational Inequalities with Non-Lipschitz Operators.  Cybernetics and Systems Analysis, 2015.
> >
> > [2] Thong et al. Weak and strong convergence theorems for solving pseudo-monotone variational inequalities with non-Lipschitz mappings. Springer, 2019.
> >
> > [3] Alfredo et al. An iterative algorithm for the variational inequality problem. Computational and Applied Mathematics, 1994.
> >
> > [4] Solodov et al. A New Projection Method for Variational Inequality Problems. SIAM Journal on Control and Optimization, 1999.
> >
> > [5] Anh et al. Generalized projection method for non-Lipschitz multivalued monotone variational inequalities. Acta Mathematica Vietnamica, 2009.
> >
> > [6] Korpelevich. The extragradient method for finding saddle points and other problems. Matecon, 1976.
> >
> > [7] Mokhtari et al. Convergence Rate of O(1/K) for Optimistic Gradient and Extragradient Methods in Smooth Convex-Concave Saddle Point Problems. SIAM Journal on Optimization, 2020.
> >
> > [8] Mokhtari et al. A Unified Analysis of Extra-gradient and Optimistic Gradient
> > Methods for Saddle Point Problems: Proximal Point Approach. AISTATS, 2020.
> >
> > [9] Daskalakis, et al. The limit points of (optimistic) gradient descent in min-max optimization. NeurIPS, 2018.

---

> ### Comment · Reviewer_mjvt · 2023-11-22
>
> Thank you for the response. I don't have futher questions. My overall suggestion is moving the comparison of proof techniques to the main paper, higlighting examples like GAN or multi-agent RL that lack lipscthizness, and emphasizing more on the proof challenges when stating theorems.

---

### Meta-Review · Area_Chair_mtgQ · 2023-12-07

**Metareview:**

This paper heavily related to the existing work Yang et al. (2023). In Yang et al. (2023), an ADMM-based interior point method was proposed for constrained VI (ACVI). The current paper has two improvements over Yang et al. (2023): (i) it proposes an inexact version of the ACVI method where the subproblems can be solved inexactly now; (ii) it relaxes the smoothness assumptions in Yang et al. (2023). Overall, it makes some contributions to the study of constrained VI. The authors are advised to include some more important references on ADMM-based interior point method -- the Yang et al. (2023) paper did a good job citing some important references on this method.

**Justification For Why Not Higher Score:**

Heavily related to existing work Yang et al. (2023).

**Justification For Why Not Lower Score:**

Some nice justifiable contributions. Can be accepted.

---

### Decision · Program_Chairs · 2024-01-16

Accept (poster)